



# Quantification of solid fuel combustion and aqueous chemistry
# contributions to secondary organic aerosol during wintertime
# haze events in Beijing
**Yandong Tong[1], Veronika Pospisilova[1,a], Lu Qi[1], Jing Duan[2], Yifang Gu[2], Varun Kumar[1],**
**Pragati Rai[1], Giulia Stefenelli[1], Liwei Wang[1], Ying Wang[2], Haobin Zhong[2], Urs Baltensperger[1],**
**Junji Cao[2], Ru-Jin. Huang[2], Andre Stephan Henry Prevot[1], and Jay Gates Slowik[1]**
[1]Laboratory of Atmospheric Chemistry, Paul Scherrer Institute (PSI), 5232 Villigen, Switzerland
[2]Key Lab of Aerosol Chemistry & Physics, Institute of Earth Environment, Chinese Academy of
Sciences, Xi'an, China
[a]now at: Tofwerk AG, Uttigenstrasse 22, 3600 Thun, Switzerland
**Correspondence: J. G. Slowik (jay.slowik@psi.ch)**
**Abstract:** In recent years, intense haze events in megacities such as Beijing have received significant
study. Although secondary organic aerosol (SOA) has been identified as a major contributor to such
events, knowledge of its sources and formation mechanisms remains uncertain. We investigate this
question through the first field deployment of the extractive electrospray ionisation time-of-flight
mass spectrometer (EESI-TOF-MS) in Beijing, together with an Aerodyne long time-of-flight aerosol
mass spectrometer (L-TOF AMS). Measurements were performed during autumn and winter 2017,
capturing the transition from non-heating to heating seasons. Source apportionment resolved four
factors related to primary organic aerosols (traffic, cooking, biomass burning, and coal combustion),
as well as four related to secondary organic aerosol (SOA). Of the SOA factors, two were related to
solid fuel combustion (SFC), one to SOA generated from aqueous chemistry, and one to
mixed/indeterminate sources. The SFC factors were identified from spectral signatures corresponding
to aromatic oxidation products, while the aqueous SOA factor was characterised by signatures of
small organic acids and diacids, and unusually low $CO^+/CO_2^+$ fragment ratios measured by the AMS.
Solid fuel combustion was the dominant source of SOA during the heating season. However, a
comparably intense haze event was also observed in the non-heating season, and was dominated by
the aqueous SOA factor. Aqueous chemistry was promoted by the combination of high relative
humidity and air masses passing over high $NO_x$ regions to the south and east of Beijing, leading to
high particulate nitrate. The resulting high liquid water content was highly correlated with the
concentration of the aqueous SOA factor. These results highlight the strong compositional variability
between different haze events, indicating the need to consider multiple formation pathways and
precursor sources to describe SOA during intense haze events in Beijing.
## 1. Introduction
Atmospheric aerosols negatively affect human health (Beelen et al., 2014; Laden et al., 2006; Pope et
al., 2002), visibility (Chow et al., 2002), and urban air quality (Fenger, 1999; Mayer, 1999) on local
and regional scales. Aerosols are also linked to the most important uncertainties with regards to global
radiation balance and climate change (Myhre et al., 2014; Penner et al., 2011; Forster et al., 2007;
Lohmann and Feichter, 2005). Organic aerosol (OA) is a major component of atmospheric aerosol
and contributes significantly to the total aerosol mass (Jimenez et al., 2009). OA sources are typically
classified as either primary organic aerosol (POA), which is directly emitted from sources such as
fossil fuel combustion, industrial emissions, biomass burning and cooking emissions, or secondary
organic aerosol (SOA), which is produced by atmospheric oxidation of volatile organic compounds
(VOCs), yielding lower-volatility products that can subsequently partition to the particle phase.



Overall, SOA accounts for approximately from 20 % to 90 % of total OA (Jimenez et al., 2009). The
health effects resulting from primary and secondary aerosols have drawn public attention in recent
years. For example, exposure to primary wood burning aerosol is associated with increased risk of
respiratory disease and may cause adverse effects to airway epithelia (Liu et al., 2017a; Krapf et al.,
2017). SOA has the potential for inducing reactive oxygen species (ROS); these can be linked to
inflammation and oxidative stress at high concentration (Reuter et al., 2010; Li et al., 2003), which in
turn can cause oxidative damage to proteins and DNA within cells (Halliwell and Cross, 1994).
Therefore, understanding the aerosol sources becomes essential. Previous studies have been relatively
successful in quantitatively linking POA to its sources, However, quantification of SOA sources
and/or formation pathways is much more challenging, because SOA consists of thousands of
multifunctional, highly oxygenated compounds, including high molecular weight species and
oligomers, which are difficult to measure using traditional instrumentation. Therefore, the effects of
individual SOA sources on health and climate remain poorly constrained.
Air pollution problems caused by high concentration of fine particles are of great concern in many
Chinese cities, including Beijing. In 2013, the Chinese government introduced the "Atmospheric
Pollution Prevention and Control Action Plan", a five-year plan to aggressively control anthropogenic
emissions and $PM_{2.5}$ concentration. Although the annual mean concentration of $PM_{2.5}$ in Beijing
decreased from 90 µg m$^{-3}$ in 2013 to 58 µg m$^{-3}$ in 2017, it remains much higher than the Chinese
National Ambient Air Quality Standard (CNAAQS, 35 µg m$^{-3}$ as an annual mean $PM_{2.5}$ concentration)
and the World Health Organisation (WHO) guidelines (10 µg m$^{-3}$ as an annual mean $PM_{2.5}$
concentration). Also, extreme haze events are still very frequent in northern China (An et al., 2019).
Extensive studies conducted in Beijing in recent years investigated the chemical composition and
sources of fine particles by different online and offline analytical methods (Duan et al., 2020; Duan et
al., 2019; Xu et al., 2019; Zhao et al., 2019; Äijälä et al., 2017; Elser et al., 2016; Hu et al., 2016; Sun
et al., 2016a; Huang et al., 2014; Zhang et al., 2014; Sun et al., 2013). Among these studies, aerosol
mass spectrometers (AMS) equipped with a $PM_1$ aerodynamic lens are widely used due to their robust
quantification ability. In these studies, POA factors are separated and well-understood. Coal
combustion aerosol was found to be one of the most important POA sources in winter, with mean
contributions ranging from 10 % to 30 %. Primary biomass burning emissions contributed
approximately 9 % to 18 % of OA. Traffic and cooking were also consistently identified as significant
POA sources, comprising 9 % to 18 % and 12 % to 20 %, respectively, of OA. SOA comprises about
35 % to 70 % of total OA in Beijing, but apportionment to specific sources and/or formation
processes has not been achieved yet in Beijing. This is typical of AMS measurements, which in most
studies report either a single SOA factor (denoted oxygenated organic aerosol, OOA), or two factors
distinguished by the extent of oxygenation (less oxygenated OOA, LO-OOA, and more oxygenated
OOA, MO-OOA) (Xu et al., 2019; Elser et al., 2016; Sun et al., 2016a; Sun et al., 2013). These
factors have alternatively been described in terms of volatility as semi-volatile OOA (SV-OOA) and
low-volatility OOA (LV-OOA), respectively (Zhao et al., 2019; Zhang et al., 2014; Hu et al., 2013).
The main exceptions to this descriptive framework for SOA include SOA from oxidation of isoprene
epoxydiols (Budisulistiorini et al., 2013) and long-term measurements in Europe by online ACSM
and/or offline AMS that show systematic differences in SOA factors retrieved during different
seasons. These differences have been attributed to the relative importance of biogenic SOA in summer
vs. SOA from domestic wood combustion in winter, although this attribution rests on temporal
correlations with external tracer measurements and knowledge of source seasonality rather than
specific AMS spectral markers (Bozzetti et al., 2017; Daellenbach et al., 2017). As a result, the main
contributors to SOA in Beijing (and other megacities) remain a source of significant uncertainty,
which continues to hinder the development of effective mitigation strategies.
OA composition is measured either by filter sampling followed by offline laboratory analyses or by
online, real-time instrumentation. Offline filter analysis has some advantages, including 1) possibility
to apply a wide variety of analytical techniques, which can maximise the chemical information



retrieved for the analysed fraction; and 2) low cost and maintenance requirements for filter sampling,
which in turn facilitates 3) practicality of wide spatial and temporal coverage. However, it also has
some drawbacks, including 1) low time resolution incapable of capturing characteristic timescales of
certain OA sources and/or ageing and formation processes; 2) artefacts due to adsorption, evaporation,
and chemical reactions during sample collection, storage, and/or transfer, (Ge et al., 2012; Huang et
al., 2010; Hildebrandt et al., 2010; Hallquist et al., 2009); 3) the analysable OA faction may vary
significantly between different techniques. Current online real-time aerosol speciation techniques
provide higher time resolution, but have typically required trade-offs between chemical resolution,
time resolution, and/or detectable mass fraction. For example, the Aerodyne aerosol mass
spectrometer (AMS) provides quantitative NR-PM$_{2.5}$ chemical composition (including sulphate,
nitrate, ammonium, chloride and organic) with time resolution on the order of 1 min. These
quantitative and highly time-resolved measurements are facilitated by high temperature vaporisation
(600 $^{\circ}$C) and high energy ionisation (electron ionisation at 70 eV) (DeCarlo et al., 2006), which induce
significant thermal decomposition and ionisation-induced fragmentation. These processes destroy
chemical information of molecules beneficial for source apportionment, particularly for the
multifunctional and highly oxygenated molecules of which SOA is comprised. The CHARON PTR-
MS uses a lower temperature vaporisation scheme to avoid thermal decomposition, while maintaining
high time resolution, but the proton transfer reaction ionisation scheme is sufficiently energetic to
cause extensive fragmentation of typical SOA molecules (Muller et al., 2017; Eichler et al., 2015). To
reduce ionisation-induced fragmentation, several semi-continuous measurement techniques have also
been developed, e.g. Filter Inlet for Gases and AEROsols chemical ionisation time-of-flight mass
spectrometer (FIGAERO-CIMS) by Lopez-Hilfiker et al. (2014), and Thermal Desorption Aerosol
GC/MS-FID (TAG) by Williams et al. (2006). Although these instruments have lower thermal
decomposition and better chemical resolution, like offline filter sampling they are subject to
reaction/vaporisation processes on the collection substrate and decreased time resolution. Therefore,
to better quantify the SOA sources and/or formation processes, an instrument that can resolve aerosol
chemical composition at the molecular level with higher time resolution is required. The extractive
electrospray ionisation time-of-flight mass spectrometer (EESI-TOF-MS), developed at Paul Scherrer
Institut (PSI), provides online, near-molecular-level measurement (i.e., molecular formulae) of
organic aerosol composition with high time resolution of seconds without thermal decomposition or
ionisation-induced fragmentation (Lopez-Hilfiker et al., 2019). Two recent source apportionment
studies in Zurich using an EESI-TOF, together with an AMS, successfully resolved several SOA
factors and quantified the processes governing SOA concentrations (Qi et al., 2019; Stefenelli et al.,
2019). These studies confirm that the combination of EESI-TOF and AMS is highly complementary,
with the AMS providing robust quantification but limited chemical resolution, and the EESI-TOF
providing a linear but hard-to-quantify response with high chemical resolution. The combined
measurements, therefore, has the potential to provide quantitative, real-time measurement of organic
aerosol composition with high chemical resolution.
Here we present AMS and EESI-TOF measurements in Beijing from late September to mid-
December 2017. This campaign captures distinct characteristics of the non-heating season and heating
season, which begins on 15 November. An integrated source apportionment analysis of AMS and
EESI-TOF data is performed to characterise the sources and physicochemical processes governing
SOA composition.

## 2. Methodologies

### 2.1 Measurement campaign

Measurements were conducted at the National Centre for Nanoscience and Technology in Beijing
(40.00º N, 116.38º E). Beijing is the capital city of China PR and one of the most populated cities in





the world, with more than 20 million inhabitants. It is located at the northwestern end of the North
China Plain and surrounded by the Yan Mountains from the southwest to the northeast. The
measurement site is located on the roof of the South Building of the National Centre for Nanoscience
and Technology (~20 m above ground level) surrounded by smaller buildings, with the exception of
an 18-floor building approximately 30 m to the north, which may interfere and even block the wind
from this direction. The northern part of the fourth ring highway is situated about 200 m way to the
south of the site, however, between the highway and the site, there are several buildings, which reduce
the influence from highway traffic. This location is not affected by major emissions from industries.
The measurements took place from late September to mid-December, 2017. Here we focus on OA
measurements from late October to mid-December, 2017, conducted by an extractive electrospray
ionisation long time-of-flight mass spectrometer (EESI L-TOF MS) and a long time-of-flight aerosol
mass spectrometer (L-TOF AMS). A scanning mobility particle sizer (SMPS) was additionally
deployed at the site to measure particle size distribution. Ambient air was sampled through a $PM_{2.5}$
cyclone to remove coarse particles (~ 50 cm above the roof of the measurement site building). The air
passed through a stainless steel (~ 6 mm outer diameter and ~ 4 mm inner diameter) tube into the
EESI L-TOF MS, L-TOF AMS, and SMPS, installed on the same line and in close proximity.
**2.2 Instrumentation**
2.2.1  Extractive electrospray ionisation long time-of-flight mass spectrometer (EESI-
TOF)
The EESI-TOF provides online, highly time-resolved measurements of the organic aerosol molecular
ions without thermal decomposition or ionisation-induced fragmentation. A detailed description can
be found elsewhere (Lopez-Hilfiker et al., 2019). The system used in this campaign consists of a
recently developed EESI source integrated with a commercial long-time-of-flight (LTOF) mass
spectrometer (Tofwerk AG, Thun, Switzerland), which in this campaign achieved mass resolution of
~8000 Th $Th^{-1}$ at mass to charge ratios $m/z$ higher than 170. The EESI-TOF continuously sampled at
~0.8 L $min^{-1}$, alternating between direct ambient sampling (15 min) and sampling through a particle
filter (5 min) to obtain a measurement of the instrument background. The ambient spectrum ($M_{total}$)
minus the average of the immediately adjacent background spectra (before and after) ($M_{filter}$) yields a
difference spectrum taken as the ambient aerosol composition ($M_{diff}$). In both modes, the sampled air
passes through a multi-channel extruded carbon denuder (with diameter of 4 mm and length of 3 to 4
cm) which eliminates negative artefacts from semi-volatile species desorbing from the particle filter
and positive artefacts when the particle filter acts as a sink of semi-volatile species, and also improves
detection limits by reducing the gas-phase background. Particles then intersect a spray of charged
droplets generated by a conventional electrospray probe and the soluble fraction is extracted into the
solvent. Then the droplets pass through a heated stainless-steel capillary (~250 ◦C), wherein the
electrospray solvent evaporates and ions are ejected into the mass spectrometer. Due to short
residence time (~1 ms) in the capillary, no thermal decomposition is observed. Finally, the ions are
analysed by a portable high resolution time-of-flight mass spectrometer with an atmospheric pressure
interface (API-TOF) (Junninen et al., 2010). In this campaign, the electrospray consisted of a 1:1
water/acetonitrile mixture doped with 100 ppm NaI, and the mass spectrometer was configured to
detect positive ions. Ions are detected in the form of [M]Na$^+$ (where M is the analyte) and other
ionisation pathways are mostly suppressed, yielding a linear response to mass (without significant
matrix effects) and simplifying spectral interpretation (Lopez-Hilfiker et al., 2019).
The high pollution levels experienced during this campaign presented several operational and
analytical challenges for the EESI-TOF, specifically: (1) denuder break-through, which increased
background signal, led to the detection of spurious signals in the particle phase, and increased the time
required to achieve a stable signal following a filter switch between $M_{total}$ and $M_{filter}$; (2) prevalence of
large particles during haze events; and (3) increase in the required frequency of cleaning (unclogging)



and realigning the electrospray capillary. These issues are discussed below in conjunction with the
operational and data analysis protocols.
The EESI-TOF achieved ~ 90 % data coverage during the sampling period, with the 10 % missing
data including solution changes, signal loss due to electrospray capillary clogging, interruption by
periodic maintenance (e.g. to clean the ESI and capillary into the TOF and to regenerate the denuder)
and calibration by nebulising levoglucosan aerosol to quantify the mass concentration with an SMPS
after each haze event (typically three to four days). Although the calibration by levoglucosan could
indicate the instrument's linear response to mass concentration, the sensitivity to levoglucosan was
found to be different in between different haze events because of the interruption. Therefore, a
diagnostic species that can be measured with higher time resolution is utilised to monitor the
sensitivity throughout the campaign. Intercomparison of inorganic nitrate species between AMS and
EESI-TOF yieded strong correlations during periods in which the instrument operation was stable
(i.e., not affected by major clogging or cleaning/realignment of the electrospray capillary). Note that
these issues are expected to result in changes in EESI-TOF sensitivity that uniformly affect all
measured ions (i.e., without compound-dependent effects). Therefore, we correct for these effects by
normalizing the EESI-TOF signal using a comparison of the nitrate signal ($[NaNO_3]Na^+$) from the
EESI-TOF and the nitrate concentration ($NO_3^-$) from the AMS (Fig. S1). The whole campaign was
divided into different periods, and the slope of linear fit between EESI-TOF nitrate signal and AMS
nitrate concentration in each period was taken as the sensitivity of EESI-TOF to nitrate in this period
as shown in Eq. (1a). The time period from 3 to 7 November was selected as a reference period and
the sensitivity determined in other periods ($k_q$, with $q$ denoting the individual periods) was normalised
to the sensitivity of reference period ($k_{ref}$). Finally, the data collected from EESI-TOF was normalised
according to Eq. (1b)

$$k_q = \left( \frac{I_{[NaNO_3]Na^+, EESI-TOF}}{I_{NO_3^-, AMS}} \right)_q \tag{1a}$$

$$I'_{i,j,q} = I_{i,j,q} \times \frac{k_{ref}}{k_q} \tag{1b}$$

Here $I_{[NaNO_3]Na^+, EESI-TOF}$ and $I_{NO_3^-, AMS}$ are the signal of $[NaNO_3]Na^+$ and $NO_3^-$ collected by EESI-
TOF and AMS, respectively in Eq. (1a). In Eq. (1b), $I'_{i,j,q}$ and $I_{i,j,q}$ indicate the signal of the $i$th ion at
time point $j$ in $p$th period after and before normalisation, respectively, and $k_{ref}$ is the reference
sensitivity. After normalisation, time-dependent changes in sensitivity are eliminated for nitrate and,
we assume, for organics.
In the conventional EESI-TOF setup, the denuder is positioned less than 1 cm away from the
electrospray probe (Lopez-Hilfiker et al., 2019). In this campaign, we found that this configuration led
to significant losses of large particles. This was inferred from investigation of the ratio of the EESI-
TOF particle signal ($M_{diff}$) at $m/z$ 185 ($C_6H_{10}O_5Na^+$, corresponding to levoglucosan and its isomers) to
that of the AMS $C_2H_4O_2^+$ ($m/z$ 60) signal, a characteristic fragment of anhydrosugars such as
levoglucosan. During a haze period (characterised by large particles with a vacuum aerodynamic
diameter ($d_{va}$) mass distribution centred around 665 nm, Fig. S2), the EESI-TOF:AMS ratio was ~2.
In contrast, during a clean period (characterised by smaller particles with a $d_{va}$ mass distribution
centred around 302 nm, Fig S2) the EESI-TOF:AMS ratio was ~13 in the AMS. By changing the
position of the denuder from 1 cm to 9 cm away from the ESI probe, we increased the ambient signal
of $m/z$ 185 ($M_{diff}$) by a factor of 6 under haze conditions, recovering the EESI-TOF:AMS ratio
observed for small particles. Therefore, we positioned the denuder in this campaign about 9 cm away
from the probe to avoid size-dependent transmission artefacts. We suggest that the effect of the
denuder position on large particle collection is due to the axial velocity profile, which is independent
of radial position at the exit of the honeycomb denuder (and nearly so at the 1 cm position) but closer





1 to a laminar flow profile at 9 cm. For the 1 cm position, the increased momentum of large particles in
2 the outer regions of the particle flow likely prevents their efficient intersection with the spray and/or
3 subsequent collection into the MS capillary inlet.

4 Data analysis was performed using Tofware version 2.5.7 (Tofwerk AG, Thun, Switzerland). Before
5 high-resolution peak fitting, data were averaged to 2 min. Although EESI-TOF ion signal normally
6 takes only a few seconds to stabilize after a filter switch, in this campaign denuder breakthrough
7 yielded stabilization times from several seconds to several minutes, depending on the ion. Therefore,
8 only the stabilised part of the averaged time series was used for further analysis, corresponding to the
9 last 4 min in the 15 min period of ambient sampling, and the last 2 min in the 5 min filter sampling
10 period, while the remaining time is classified transitional measurements and discarded from further
11 analysis. Adjacent filter sampling periods were linearly interpolated to obtain an estimated $M_{filter}$
12 corresponding to each $M_{total}$; the difference of $M_{total}$ minus the interpolated $M_{filter}$ yields the $M_{diff}$
13 reported here. In total, 2824 ions were identified ranging from $m/z$ 64 to $m/z$ 400. All ions were
14 detected as adducts with Na[+]. To facilitate comparison with bulk mass measurements, EESI-TOF
15 signals were converted from counts per second (cps) to the mass flux of ions to the microchannel plate
16 detector (ag s[-1]), as follows:

$$M_x = I_x \times (\mathrm{MW}_x - \mathrm{MW}_{CC}) \tag{2}$$

18 where $M_x$ and $I_x$ are respectively the mass flux of ions in attogrammes per second and the ion flux
19 (cps) reaching the detector for a given ion of identity $x$ . $\mathrm{MW}_x$ and $\mathrm{MW}_{CC}$ represent the molecular
20 weight of the ion and the charge carrier (e.g. Na[+]), respectively (Lopez-Hilfiker et al., 2019; Qi et al.,
21 2019; Stefenelli et al., 2019). This measured mass flux can in principle be converted to ambient
22 concentration by the instrument flow rate, EESI collection efficiency (the probability that the analyte-
23 laden droplet enters the inlet capillary), EESI extraction efficiency (the probability that a molecule
24 dissolves in the spray), ionisation efficiency (the probability that an ion forms and survives
25 declustering forces induced by evaporation and electric fields), and ion transmission efficiency (the
26 probability that a generated ion is transmitted to the detector). However, since several of these
27 parameters are compound-dependent and remain uncharacterised, mass concentration at this stage
28 cannot be determined (Lopez-Hilfiker et al., 2019).

29 The high background signals resulting from denuder breakthrough compromised high-resolution peak
30 fitting of the spectral region containing particle-phase signals in Tofware. Because particle-phase
31 signals tend to be less oxygenated (lower mass defect) than the background ions, a custom peak fitting
32 algorithm (outside of Tofware) was used in which the relative weight of this spectral region was
33 increased, as described in the supplement (see Sect. S1, Fig. S3 and Fig. S4). Further, denuder
34 breakthrough rendered non-trivial the classification of ions as arising primarily from the particle
35 phase, working solution and its impurities vs. gaseous molecules transmitted to the ion source via
36 denuder breakthrough. As only particle-phase ions are desired for further analysis, three criteria were
37 applied for their selection: 1) the ratio of signal to uncertainties, 2) ratio of signal to background and
38 3) estimated saturation vapour mass concentration ($C_0$) (see Text S2). For criterion (3), only ions
39 having a lower $C_0$ than levoglucosan were retained. Note that this biases our measurements towards
40 the exclusion of small acids characteristic of aqueous processes.

41 After application of the criteria in Text S2, 401 ions are retained for further analysis. As discussed in
42 Sect. 2.3, source apportionment was conducted on the EESI-TOF data by positive matrix factorization
43 (PMF), which requires as input the mass spectral time series and corresponding uncertainties.  The
44 input data matrix $M_{diff}(i, j)$ is calculated according to Eq. (3):

45
$$M_{diff}(i.j) = \mathrm{M}_{total}(i,j) - M_{filter}(i,j) \tag{3}$$





where $M_{total}(i,j)$ denotes the signal of spectra measured in total sampling period,
$M_{filter,estimate}(i,j)$ denotes signal of estimated background spectra after interpolation of the filter
sampling period, and $M_{diff}(i,j)$ denotes signal of the difference spectra between total sampling
period and estimated background and consists of 401 (ions) $\times$ 1239 (time points). The error matrix
corresponding to $M_{diff}$ is estimated by adding in quadrature the uncertainty of total sampling
measurement $\sigma_{total}(i,j)$ and filter sampling measurement $\sigma_{filter,estimate}(i,j)$ , which are based on
ion counting statistics and detector variability (Allan et al., 2003b), shown in Eq. (4):

$$\sigma_{diff}(i,j) = \sqrt{\sigma^2_{total}(i,j) + \sigma^2_{filter,estimate}(i,j)} \tag{4}$$

10          2.2.2    Long time-of-flight aerosol mass spectrometer (L-TOF AMS)

A long time-of-flight aerosol mass spectrometer (L-TOF AMS, Aerodyne Research Inc.) equipped
with a $PM_{2.5}$ aerodynamic lens was deployed to monitor the non-refractory (NR) particle composition
with a time resolution of 2 min. The instrument is described in detail elsewhere (Canagaratna et al.,
2007). Briefly, particles are sampled continuously at ~0.1 L min$^{-1}$ into a 100 µm critical orifice and
then a $PM_{2.5}$ aerodynamic lens, which focuses the particles into a narrow beam and accelerates them to
a velocity inversely related to their vacuum aerodynamic diameter (Williams et al., 2013). The
particle beam impacts on a heated tungsten surface ~ 600 ˚C, and ~ $10^{-7}$ Torr) and the NR
components flash vaporise. The resulting gases are ionised by electron ionisation (EI, ~ 70 eV) and
measured by a TOF mass spectrometer. The instrument was calibrated for ionisation efficiency (IE) at
the beginning, middle and end of the campaign by a mass-based method using 350 nm $NH_4NO_3$
particles. To eliminate the influence from relative humidity (RH) on collection efficiency (CE), a
Polytube Dryer Gas Sample Dryer (Perma Pure LLC) was mounted in front of the AMS inlet. A
composition-dependent collection efficiency (CDCE) was applied to correct the measured aerosol
mass (Middlebrook et al., 2012). Data analysis was performed in Igor Pro 6.39 (Wavemetrics, Inc.)
using SQUIRREL 1.57 and PIKA 1.16 ((Donna Sueper, ToF-AMS high-resolution analysis software).
In conventional AMS data analysis, the signal from $CO^+$ cannot be directly determined due to
interference from $N_2^+$, and is instead assumed to be equal to that of $CO_2^+$. However, the increased
mass resolution provided by the L-TOF detector was sufficient in this study to allow direct peak
fitting of $CO^+$, which is reported herein. As shown by Pieber et al. (2016), $CO_2^+$ signal in the AMS
derives not only from OA and gaseous $CO_2$, but is also generated directly from the vaporiser in the
presence of some inorganic aerosols, notably $NH_4NO_3$. This effect was corrected using 350 nm
$NH_4NO_3$ aerosol according to the method recommended by Pieber et al. (2016); the nitrate fraction
was not high enough to require the composition-dependent method of Freney et al. (2019). The $CO_2^+$
signal resulting from nitrate was found to be 4.4 % of the total $CO_2^+$ signal. In principle, spurious $CO^+$
signal can be generated by the same process, either through fragmentation of $CO_2$ or directly via
related oxidation reactions. However, the $CO^+$ signal was below detection limit for the $NH_4NO_3$ test
aerosol. We therefore assumed a value of 0.4 % of total $CO^+$ signal, which corresponds to 10 % of
$CO_2^+$ as given by the 70 eV EI reference mass spectrum of $CO_2$ according to the NIST Standard
Reference Simulation Website (Shen et al., 2017).
Source apportionment (see Sect. 2.3) was performed on the AMS OA data and requires as inputs the
OA mass spectral time series and corresponding uncertainties. The data matrix was constructed by
including both (1) ions with known molecular formula for $m/z \leq 120$ and (2) the integrated signal
across each integer $m/z$ for $m/z$ 121 to $m/z$ 300. This allows inclusion of chemical information at $m/z$
where the number of possible ions and AMS resolution are insufficient for robust identification and
quantification of individual ions. Of particular note for the current dataset, inclusion of the high $m/z$
data allows inclusion of polycyclic aromatic hydrocarbons (PAHs) in the PMF analysis. Uncertainties


were calculated according to the method of Allan et al. (2003a), and account for electronic noise, ion-
to-ion variability at the detector, and ion counting statistics, with a minimum error enforced according
to the method of (Ulbrich et al., 2009). As recommended by Paatero and Hopke (2003), variables with
weak SNR (0.2<SNR<2) were down-weighted by a factor of 2 and variable with low SNR (SNR<0.2)
were removed from input matrices.
Ions that were not independently fit but calculated as a constant ratio of $CO_2^+$ i.e. $O^+$, $HO^+$ and $H_2O^+$,
were removed from PMF analysis to avoid overweighting the contribution of $CO_2^+$. After obtaining
the PMF solutions, the contribution of these ions was recalculated and reinserted into the factor
profile. The resulting factor profiles were re-normalised, likewise the total mass. Note that although
typical AMS source apportionment studies likewise remove $CO^+$, the increased mass resolution of the
LTOF detector allows an independent measurement of $CO^+$ and this ion is therefore retained for PMF.
Isotopes were removed prior to PMF analysis (to avoid overweighting the parent ions) and reinserted
afterwards

### 2.3 Source Apportionment Technique

Source apportionment was performed using the positive matrix factorisation (PMF) model,
implemented within the multilinear engine (ME-2). AMS and EESI-TOF measurements are highly
complementary, with the AMS providing robust quantification but limited chemical resolution, and
the EESI-TOF providing a linear but hard-to-quantify response with high chemical resolution. As a
result, integrating these two instruments in single source apportionment model represents a promising
strategy for improved source apportionment, especially of the SOA fraction. Conceptually, this can be
executed in three ways: (1) PMF analysis on a single dataset containing both AMS and EESI-TOF
data; (2) PMF analysis of EESI-TOF-only data to identify factors and determine their time series,
followed by PMF on AMS-only data with factor time series constrained according to EESI-TOF
results; or (3) PMF on AMS-only data to determine factor time series, followed by PMF on EESI-
TOF-only data with constrained factor time series to facilitate chemical interpretation of the AMS-
determined factors. For the present analysis, we selected method (3) because of EESI-TOF data
quality issues related to denuder breakthrough (see Sect. 2.2.1) and the appearance of several
interesting-but-unexplained factors in preliminary AMS PMF analysis.
For the AMS PMF analysis, one factor related to traffic and one factor related to cooking activities
were constrained using the *a* value approach for the HOA spectra from Mohr et al. (2012) and the
COA spectra from Crippa et al. (2013). Based on the result from PMF analysis on AMS data, PMF
was then performed for the EESI-TOF dataset, by constraining all factor time series retrieved from the
first step source apportionment except for the HOA time series (because the hydrocarbon-like species
dominating HOA are undetectable by the EESI-TOF extraction/ionization scheme used here). This is
conceptually similar to chemical mass balance (CMB), except that here the factor time series are
constrained instead of factor profiles. This allows AMS-resolved factors, notably those related to
SOA, to be described in terms of the higher chemical resolution achievable by the EESI-TOF. To
explore the robustness and uncertainties of each step in our integrated source apportionment,
bootstrap analysis was conducted individually on the first step PMF solution from the AMS and the
second step "CMB-analogue" result from the EESI-TOF.
Determine of the proper number of factors to obtain the most interpretable PMF solution is partly
subjective. In this paper, criteria to identify and interpret the factors implemented include to compare
the correlation between factor time series or profiles with external references, and to investigate the
factor's distinctive chemical signatures. In addition, z-score analysis is introduced to interpret
retrieved factor profiles, and demonstrates its advantage in identifying unique ions in each factor.

46         2.3.1    Positive matrix factorisation (PMF)





Positive matrix factorisation (PMF) was implemented using the Multilinear Engine (ME-2) (Paatero,
1997), with model configuration and post-analysis performed with the Source Finder interface (SoFi,
version 6.8b) (Canonaco et al., 2013), programmed in Igor Pro 6.39 (Wavemetrics, Inc.). PMF is a
bilinear receptor model which describes the input data matrix (here the mass spectral time series) as a
linear combination of static factor profiles (in this case characteristic mass spectra, representing
specific sources or/and atmospheric processes) and their corresponding time-dependent source
contributions, as described in Eq. (5):
$$\mathbf{X} = \mathbf{G} \times \mathbf{F} + \mathbf{E} \tag{5}$$
Here $\mathbf{X}$ is the input data matrix with dimensions of $m{\times}n$, representing $m$ measurements of $n$ variables
(here ions or $m/z$), $\mathbf{G}$ and $\mathbf{F}$ are respectively the static factor time series with the dimension of $m{\times}p$,
and factor profiles with the dimension of $p{\times}n$, where $p$ is the number of factors in the PMF solution,
and is determined by the user. $\mathbf{E}$ is the residual matrix. $\mathbf{G}$ and $\mathbf{F}$ in Eq. (5) are solved by a least-
squares algorithm that iteratively minimises the quantity $Q$, which is defined in Eq. (6) as the sum of
the squares of the uncertainty-weighted residuals:
$$Q = \sum_i \sum_j \left(\frac{e_{ij}}{\sigma_{ij}}\right)^2 \tag{6}$$
Here $e_{ij}$ is an element in the residual matrix $\mathbf{E}$, and $\sigma_{ij}$ is the corresponding element in the
measurement uncertainty matrix, where $i$ and $j$ are the indices representing measurement time and ion
(or integer $m/z$), respectively.
PMF is subject to rotational ambiguity, in that different combinations of the $\mathbf{G}$ and $\mathbf{F}$ matrices may
yield solutions with the same or similar $Q$. In practice, this often leads to mixed or unresolvable
factors. Here we explore a subset of the possible PMF solutions, directed towards environmentally
meaningful rotations. This is achieved via the $a$-value approach, wherein one or more factor profiles
and/or factor time series are constrained using reference factors profiles or/and time series, with the
scalar $a$ ($0{\leq} a {\leq}1$) determining the tightness of constraint. This approach has been shown to improve
solution quality relative to unconstrained PMF (Crippa et al., 2014; Canonaco et al., 2013). The $a$-
value approach determines the extent to which the resolved factor profiles $(g_{i,k})_{solution}$ and time
series $(f_{k,j})_{solution}$ may differ from the input values ($g_{i,k}$ or $f_{k,j}$), as shown in Eq. (7a) and Eq. (7b):
$$(g_{i,k})_{solution} = g_{i,k} \pm a \times g_{i,k} \tag{7a}$$
$$(f_{k,j})_{solution} = f_{k,j} \pm a \times f_{k,j} \tag{7b}$$
Note that the final value of $(g_{i,k})_{solution}$ and $(f_{k,j})_{solution}$ may slightly exceed the prescribed limits
due to post-PMF renormalisation of the $\mathbf{G}$ and $\mathbf{F}$ matrices. Here the $a$-value approach was used for
both the AMS and EESI-TOF datasets. Sensitivity tests to determine an appropriate range of $a$-values
were performed in combination with bootstrap analysis, as described in the following section.
2.3.2   Bootstrap Analysis
Bootstrap analysis (Davison and Hinkley, 1997) was performed to characterise solution stability and
estimate uncertainties. Bootstrapping creates a set of new input and error matrices by random
resampling of rows from the original input data and error matrices. This resampling preserves the
original dimension of input data constant for every single resampling, but randomly duplicates some
time points from the original input matrices while excluding others (Paatero et al., 2014). For the
AMS dataset, we performed 1000 bootstrap runs on an eight-factor solution, with HOA and COA
factors constrained. For each factor, a random $a$-value was selected for each bootstrap run, ranging
from 0 to 0.5 with a step size of 0.1. For the EESI-TOF dataset, 1000 bootstrap runs were performed



on a 7-factor solution. Each EESI-TOF factor was constrained by a factor from the AMS 8-factor
solution, with AMS HOA excluded because it is not detectable in the EESI-TOF due to low solubility
and ionization efficiency. For the EESI-TOF bootstrapping, each factor was constrained with a
randomly selected *a*-value ranging from 0 to 0.6 with a step size of 0.1.
Conceptually, each bootstrap solution can be classified in three ways: (1) qualitatively similar to the
base case; (2) qualitatively similar to the base case, but with 2 or more factors mixed; (3)
fundamentally different from the base case, e.g. one or more factors has appeared and/or disappeared.
For characterising uncertainties in the factor profiles and/or time series, only solutions of type (1) are
considered. We therefore use the solution classification methods of Stefenelli et al. (2019), which are
based on determining whether each factor profile and/or time series from the base case is with
statistical significance more similar to one and only one factor in a given bootstrapped solution (with
no duplication). This method is implemented in three steps: 1) creation of a base case, 2) Spearman
correlation between the time series of each factor from the base case and every bootstrap solution is
calculated to sort the bootstrap factors, yielding a correlation matrix with the highest correlation
values on the diagonal, 3) each correlation coefficient on the diagonal is compared to values on the
row and column to evaluate whether this coefficient is statistically significant higher than other values
on the same row or column, by *t*-test analysis. The bootstrap solutions that fail to meet this criterion
are classified as "mixed".
The definition of a mixed solution therefore depends on the selected confidence level $p$, which is
evaluated here by a sensitivity test of $p$ ranging from 0.05 to 0.95 with a step of 0.05; the number of
solutions classified as "mixed" rises as $p$ increases (Fig. S5). This enables identification of the
solutions most likely to be classified as "mixed" for each increment of $p$. These solutions are
manually inspected to confirm that they do in fact appear mixed, and the final $p$ is selected once this
no longer holds true. Using this method, a final $p$ of 0.40 for AMS was chosen, yielding 918 accepted
bootstrap runs. For EESI-TOF bootstrap analysis, since the time series of all factors are constrained,
all runs are considered as good runs and utilised to explore the variability of factor profiles.
30            2.3.3    z-score

The dynamic range of EESI-TOF and AMS ion signal concentrations spans several orders of
magnitude. It can be that key chemical information is contained in low-intensity ions, which are not
readily evident from the factor profile. To assist in identifying such spectral features, we calculate the
z-score of each ion across the factor profile matrix as follows:

$$z_{p,g} = (x_{p,g} - \mu_p)/\sigma_p \tag{8}$$

Here $z_{p,g}$ and $x_{p,g}$ is the z-score and the relative intensity of ion $p$ in factor profile $g$, respectively,
and $\mu_p$ and $\sigma_p$ is the mean and standard deviation of relative intensity of ion $p$ in all PMF factors. The
Z-score is a signed, dimensionless quantity whose absolute value is to describe the distance between
an observation $x$ and population mean $\mu$ in the unit of standard deviation $\sigma$ (Larsen and Marx, 2018).
It therefore highlights ions whose contribution to a factor profile is unexpectedly high (or low),
independent of absolute signal magnitude. In this study, z-score is used to identify key ions that are
unique to a specific factor or small subset of factors, as discussed in Sect. 3.3.

**3.    Results**
45         **3.1 Campaign overview**

During the measurement period, nine haze episodes were observed in total (Fig. 1) from 31 October to
5 December. Of these, four haze episodes occurred during the non-heating season, four during the



heating season, and one episode bridged the transition date. Consistent with previous studies (Duan et
al., 2020; Duan et al., 2019; Zhao et al., 2019; Xu et al., 2019; Sun et al., 2016a; Sun et al., 2016b),
alternating haze episodes and clean periods corresponded systematically to changing meteorological
conditions. Haze build-up was associated with stagnant air masses with slow wind speed ($< 1.5$ m s$^{-1}$)
mainly from the south or southwest, and terminated by air masses with high wind speed ($> 3$ m s$^{-1}$)
from the north or northwest (Fig. 1b and 1c). Different from previous studies in Beijing in 2014 and
2015, where haze events lasting more than five days were observed (Zhao et al., 2019; Xu et al., 2019;
Sun et al., 2016b), all haze events in this campaign lasted for two to four days. The maximum
concentration of NR-PM$_{2.5}$ measured by the L-TOF AMS exceeded 100 µg m$^{-3}$ in only one haze event
(4 to 7 November), and the mean NR-PM$_{2.5}$ concentration in the haze episodes was $36.6 \pm 22.7$ µg m$^{-3}$
, which is even lower than mean concentrations of NR-PM$_1$ observed in Beijing winter from 2013
($89.3 \pm 85.6$ µg m$^{-3}$) to 2016 ($64 \pm 59$ µg m$^{-3}$) (Zhao et al., 2019; Xu et al., 2019; Sun et al., 2016a;
Zhang et al., 2014).
Aerosol bulk composition differs between the non-heating and heating seasons, indicating changes in
sources and/or chemical processes. Organic aerosol (OA) is the major fraction of NR-PM$_{2.5}$
throughout the campaign period, with a mean contribution of 54 %, consistent with previous winter
studies in Beijing (Zhao et al., 2019; Xu et al., 2019; Elser et al., 2016). The temporal evolution of
OA shows that the contribution in haze episodes increased from 41 % during the non-heating season
to 54 % during the heating season. This contrasts with nitrate, which is the second largest contributor
to NR-PM$_{2.5}$ in this study and contributes 37 % of NR-PM$_{2.5}$ in non-heating season haze events (37 %)
but decreases to 23 % during heating season haze events. Of particular note is the non-heating season
haze event from 4 to 7 November, where nitrate comprises more than 50 % of NR-PM$_{2.5}$, exceeding
OA contribution to total mass in this event. This event is discussed in detail in Sect. 3.3.4 and Sect. 4.
It is also worth noticing that the nitrate concentration and its contribution was lower than sulphate
during every clean period, but higher during every haze episode. The mean nitrate/sulphate ratio in the
present study is 2.8±2.4, a significant increase compared to observations in 2014 (0.7±0.6) and 2016
(1.4±0.9) from Xu et al. (2019). In addition, the nitrate/sulphate ratio exceeded 1 for 63 % of
measurements in the present study, compared with only 24 % in 2014. It is clear that the contribution
of nitrate in haze events gradually exceeded the contribution of sulphate from 2014 to 2017,
indicating nitrate is playing an increasingly important role relative to sulphate in haze formation,
mainly due to large reduction in SO$_2$ emissions from coal fired power plants in Beijing and
surrounding areas.

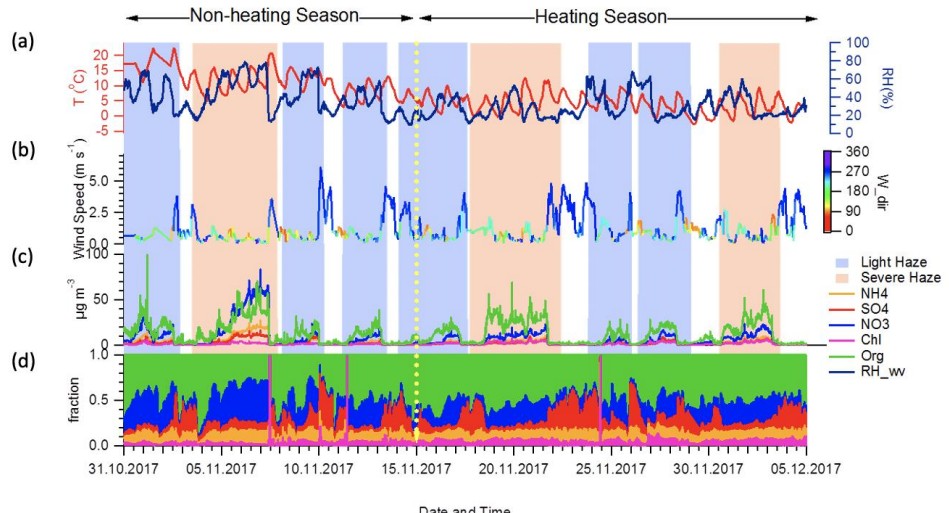

Figure 1. Time series of meteorological variables and NR-PM$_{2.5}$ composition. (a) temperature ($T$) and
relative humidity (RH), (b) wind speed and wind direction, (c) mass concentrations of NR-PM$_{2.5}$
species measured by the AMS, and (d) mass fractions of the species shown in Fig. 1c. Shaded area
indicates haze episodes: light haze episodes are defined as having NR-PM$_{2.5}$ concentrations from 20 to
150 µg m$^{-3}$ (light blue), while severe haze episodes are defined having NR-PM$_{2.5}$ concentrations above
150 µg m$^{-3}$ (light red).
**3.2 AMS source apportionment**
With the combination of HR ions and UMR sticks in the PMF input matrix, eight factors were
resolved, including four primary and four secondary organic factors. Figure 2 shows the averaged MS
profiles of the selected eight-factor solution and corresponding relative contribution of each ion (i.e.,
fraction of signal from a given ion apportioned to each factor), while Fig. 3 shows the factor time
series in terms of both absolute concentration and OA mass fraction. Diurnal patterns are shown in
Fig. 3c. The four primary organic factors consist of a traffic related factor (hydrocarbon-like OA,
HOA), a cooking related factor (COA), and two solid fuel combustion-related factors (biomass
burning OA, BBOA and coal combustion OA, CCOA). The four primary factors retrieved in this
solution (HOA, COA, BBOA, and CCOA) have been resolved in several previous winter studies in
Beijing (Huang et al., 2014; Elser et al., 2016; Hu et al., 2016; Sun et al., 2016a). However, the SOA
factor resolution is unusual. AMS source apportionment studies typically report one or two
oxygenated organic aerosol (OOA) factors attributed to SOA, which are distinguished by the extent of
oxygenation, which is in turn typically linked to volatility, age, or season. Here, we report four
secondary factors, consisting of two more-oxygenated OOAs (MO-OOAs) and two less-oxygenated
OOAs (LO-OOAs). For reasons described below and in Sect. 3.3, the MO-OOA factors are attributed
to aqueous phase chemistry (MO-OOA$_{aq}$) and solid fuel combustion (MO-OOA$_{SFC}$), while the two
LO-OOA factors are attributed to solid fuel combustion (LO-OOA$_{SFC}$), and a non-source-specific
factor denoted as (LO-OOA$_{ns}$).
In selecting the PMF solution that best represents the AMS dataset, we considered both mathematical
diagnostics (e.g. $Q/Q_{exp}$) as a function of the number and the interpretability of the retrieved factors.
Evaluation of factor interpretability includes: 1) correlation of the time series with external data, 2)
comparison of factor diurnal cycles with known source activity and previous measurements in
Beijing; 3) identification of source-specific spectral features; and 4) differences in factor trends
between heating/non-heating and/or haze/non-haze periods. Solutions from five to ten factors were
explored (Fig. S7 to Fig. S12), with an eight-factor solution selected as the best representation of the
data according to the above criteria. Solutions with less than six factors showed evidence of mixed
primary sources. The seven- and eight-factor solutions resolve additional OOA factors, which have
clear temporal and compositional differences that support their separation and interpretation.
Compared to the eight-factor solution, higher-factor number solutions lead to additional splitting of
OOA factors, which cannot be interpreted by previous criteria. Therefore, the eight-factor solution is
retained for further analysis.
**HOA** -- The HOA spectrum is characterised by alkyl fragments, especially $C_nH_{2n-1}^+$ and $C_nH_{2n+1}^+$.
Major ions include $C_3H_7^+$, $C_4H_9^+$, $C_5H_{11}^+$ (Zhao et al., 2019; Xu et al., 2019; Sun et al., 2016a; Elser et
al., 2016; Zhang et al., 2014; Ng et al., 2011). It also shows good correlation with CO (Fig. S13),
which is a tracer for traffic emissions (Sun et al., 2016a; Zhang et al., 2014; Chan et al., 2011).
Concentrations of this factor peak from 06:00 to 09:00 corresponding to the morning rush hour, and
from 17:00 to 21:00 in the evening corresponding to evening-night rush hour. The averaged
concentration during the evening peak (0.50 µg m$^{-3}$) is almost twice as high as the morning peak (0.26
µg m$^{-3}$), due to the low planetary boundary layer height and accumulation of vehicle emissions at
night (Sun et al., 2016a; Han et al., 2009). This diurnal pattern is consistent with other winter studies
in Beijing (Sun et al., 2016a; Zhang et al., 2014). However, the averaged relative contribution of HOA
factor to total mass (~3 %) is significantly lower than previous studies (~10 %) (Elser et al., 2016; Hu
et al., 2016; Sun et al., 2016a; Zhang et al., 2014; Huang et al., 2010), this indicates that primary
traffic emissions comprise a minor fraction of OA during both non-heating and heating periods.
**COA** -- The COA spectrum contains both alkyl fragments and slightly oxygenated ions, consistent
with aliphatic acids from cooking oils (Hu et al., 2016). It is typically characterised by a ratio of
$C_3H_3O^+$ to $C_3H_5O^+$ which greater than 2 and is 3.4 in this study (Xu et al., 2019; Zhao et al., 2019;
Sun et al., 2016a; Sun et al., 2016b; Crippa et al., 2013; Mohr et al., 2012). The time series of the
COA factor strongly correlates with $C_6H_{10}O^+$ (*m/z* 98), a good tracer for cooking activities reported by
many studies (Xu et al., 2019; Zhao et al., 2019; Elser et al., 2016; Hu et al., 2016; Sun et al., 2016a;
Sun et al., 2016b; Mohr et al., 2012; Sun et al., 2011), with $r^2 = 0.96$ and 60.1 % of the mass of this
ion being apportioned to COA. The diurnal cycle shows three peaks: from 07:00 to 09:00 at breakfast
and from 12:00 to 13:00 at lunch time and a larger peak from 18:00 to 21:00 during dinner. This
three-peak diurnal pattern agrees with the diurnal cycle observed by Sun et al. (2016a), but differs
from many other studies at different sites in winter Beijing where only two peaks are evident and the
morning peak from 07:00 to 09:00 is missing, suggesting a dependence on the proximity to local
emissions (Xu et al., 2019; Elser et al., 2016; Hu et al., 2016; Zhang et al., 2014). The ratio of dinner
peak to lunch peak is about 2, similar to the values of ~2 and 2.3 observed by Elser et al. (2016) and
Hu et al. (2016), respectively, whereas Sun et al. (2016a) report a ratio of 1.29. Overall, the COA
factor is a non-negligible contributor to total OA, with a relative contribution of 6 %, lower than 18 %
in 2013 (Sun et al., 2016a), 25 % in 2014 and 16 % in 2016 wintertime (Xu et al., 2019). The mean
concentration is 0.30 µg m$^{-3}$, lower than previous studies (Xu et al., 2019; Zhao et al., 2019; Elser et
al., 2016; Hu et al., 2016; Sun et al., 2016a; Sun et al., 2016b; Mohr et al., 2012; Sun et al., 2011).
**BBOA** – Consistent with other studies in Beijing (Zhao et al., 2019; Elser et al., 2016; Hu et al., 2016;
Sun et al., 2016a), a BBOA factor was resolved. Typically, the BBOA factor spectrum is
characterised by increased contributions from $C_2H_4O_2^+$ at *m/z* 60 and $C_3H_5O_2^+$ at m/z at 73, which is
typical of anhydrosugars such as levoglucosan (Alfarra et al., 2007; Lanz et al., 2007; Sun et al.,
2011). However, although the contribution of the BBOA factor to $C_2H_4O_2^+$ is the highest (28.6 %)
among those factors and its correlation is also high, with $r^2 = 0.62$, other primary sources like CCOA
and COA also contribute significant fractions of $C_2H_4O_2^+$ signal. BBOA also correlates strongly with
$C_3H_5O_2^+$ ($r^2 = 0.71$) and $C_6H_6O_2^+$ ($r^2 = 0.81$), which are also typical of biomass burning activities
(Lanz et al., 2007; Sun et al., 2011). The O:C ratio and N:C ratio of this factor is 0.29 and 0.22,



respectively, agreeing quite well with the values found in other studies (Xu et al., 2019; Zhao et al.,
2019; Hu et al., 2016).
The BBOA time series is event-driven, with both concentrations and relative contributions increasing
during haze events, especially the haze event from 18 to 22 November (68.7 % of total OA). Apart
from this event, the BBOA concentration increase during other haze events is also clear, regardless of
non-heating and heating season. Overall, the average BBOA concentration for the haze events was 1.9
$\mu g\ m^{-3}$, with a maximum of 19.1 $\mu g\ m^{-3}$ for the event from 18 to 22 November, and 0.13 $\mu g\ m^{-3}$ for
the clean periods, both lower than the study in mid-winter from 2013 to 2014 (Sun et al., 2016a) and
the studies covering the same time period of the early winter in 2014 and 2016 (Xu et al., 2019). Its
relative contribution to total OA is 15.4 % for haze periods and 8.24 % for the clean period,
respectively, consistent to observations of Elser et al. (2016), who report 13.9 % and 8.9 % for haze
and clean periods in wintertime in Beijing, respectively.
**CCOA** – apart from alkyl fragments $C_nH_{2n-1}^+$ and $C_nH_{2n+1}^+$, the main feature of the CCOA profile is
the high contribution from PAHs (approximately $m/z$ 175 to 300), especially in the high $m/z$ range,
consistent with studies from Elser et al. (2016), Zhang et al. (2008) and Xu et al. (2006). In the high
mass range, PAHs contribute an increasingly higher fraction at higher $m/z$ (Fig. 2b). Strong aromatic
and PAH signatures found in the factor profile are at $m/z$ 115 ($C_9H_7^+$), 128, 139, 152, 165, 178, 189,
202, 215, 226, 239 and 252 in this study. Moreover, the time series of this factor and these signatures
correlate quite well with $r^2$ of 0.812 ($C_9H_7^+$), 0.801 ($m/z$ 128), 0.834 ($m/z$ 139), 0.903 ($m/z$ 152), 0.906
($m/z$ 165), 0.926 ($m/z$ 178), 0.938 ($m/z$ 189), 0.972 ($m/z$ 202), 0.967 ($m/z$ 215), 0.982 ($m/z$ 226), 0.962
($m/z$ 239) and 0.984 ($m/z$ 252), respectively, consistent with observations from Dzepina et al. (2007),
Hu et al. (2013), Hu et al. (2016) and Sun et al. (2016a).
Coal is used widely for domestic heating in northern China including the greater Beijing area and
surrounding provinces (Zhang et al., 2008), but is not permitted for residential use in the downtown
area. Instead, beginning on 15 November, power plants using natural gas provide heating to every
household in the Beijing downtown area, and municipal coal combustion begins to provide heating to
the surrounding area. Interestingly, the time series of the CCOA factor reflects this seasonal transition,
as the mean daily maximum concentration increased from 2.9 $\mu g\ m^{-3}$ before 15 November to 5.9 $\mu g$
$m^{-3}$ after. Similar to other studies (Elser et al., 2016; Hu et al., 2016; Sun et al., 2016a; Zhang et al.,
2014), the diurnal concentration peaks at night between 21:00 and 06:00 with an average contribution
of 15.5 % to total OA, and decreases during the day from 07:00 to 20:00 with an average contribution
of 7.4 %, consistent with domestic heating. Overall, the mean contribution to total OA is 11.4 %, with
7.1 % in the non-heating period and 14.7 % in the heating season. The latter number agrees with
observations conducted in the heating period in Beijing during winter, ranging from 10 % to 30 %
(Elser et al., 2016; Hu et al., 2016; Zhang et al., 2014; Sun et al., 2013).
**OOAs** – As noted above, the OOA factors resolved here differ from previous AMS studies in Beijing,
where only one or two OOA factors were resolved and classified based on the volatility (semi-volatile
OOA and low-volatility OOA) or oxidation state (more-oxygenated OOA and less-oxygenated OOA).
In this study, two more-oxygenated OOAs (MO-OOA) and two less-oxygenated OOA (LO-OOA)
were resolved. The OOA factors are characterised by higher signal from $CO_2^+$ than found in the POA
factors. In this study, $CO_2^+$ comprises approximately 15 % of the two MO-OOA factors. For the two
LO-OOAs, the $CO_2^+$ contribution to the total signal is only 3.78 % in LO-OOA$_{SFC}$ and 5.41 % in LO-
OOA$_{ns}$, while the ratio of $CO_2^+$ to $C_2H_3O^+$ is still higher than it for the POAs. Moreover, a higher
contribution of the $C_xH_y$ group is observed in the LO-OOA factors than in the MO-OOA factors. The
four OOA factors have significantly different time series, corresponding to specific haze events and/or
seasonal changes, providing a first suggestion that their separation may be meaningful.





Among the MO-OOA factors, one factor (influenced by aqueous phase chemistry, defined as MO-
OOA$_{aq}$) has high absolute and relative concentrations during a single haze event from 4 to 7
November (maximum 16.2 µg m$^{-3}$, > 60 % of the total OA mass), but is a minor component
throughout the rest of the campaign. In contrast, the other MO-OOA factor (oxygenated from solid
fuel combustion, defined as MO-OOA$_{SFC}$) is a minor component before 15 November, but both its
mass and relative contribution steadily increase during the heating season, especially during haze
periods. This is consistent with the temporal pattern of CCOA, suggesting this factor may be linked to
coal combustion activities. The temporal evolution of the two LO-OOA factors are also
distinguishable. The concentration of one factor (LO-OOA$_{SFC}$) increases in every haze episode under
stagnant conditions and is correlated with the total OA time series ($r^2$ =0.91), whereas the other factor
(LO-OOA$_{ns}$) exhibits a clear diurnal pattern in the non-heating season, but this diurnal cycle is absent
during the heating season. Interestingly, the contribution of the LO-OOA$_{ns}$ factor to total OA is higher
during the clean days, suggesting this factor may be more influenced by regional processes. The
chemical characteristics and sources/processes governing these OOA factors are discussed in detail in
the next section, in conjunction with the EESI-TOF analysis.

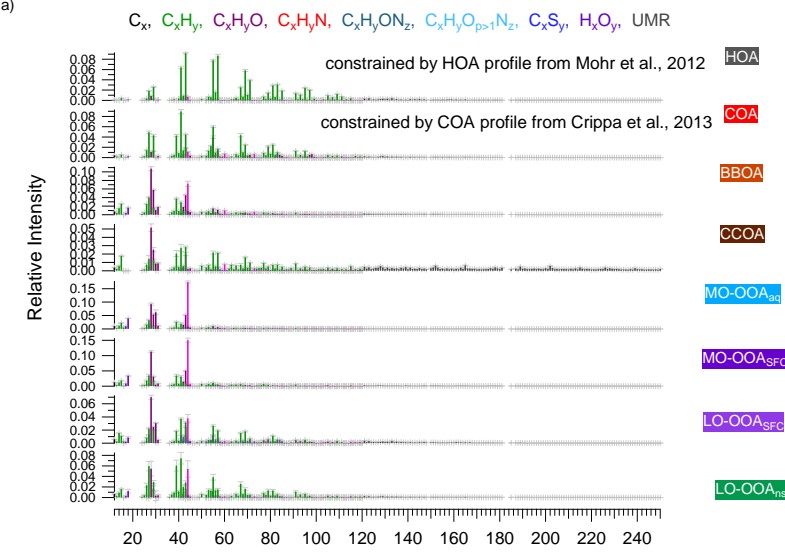





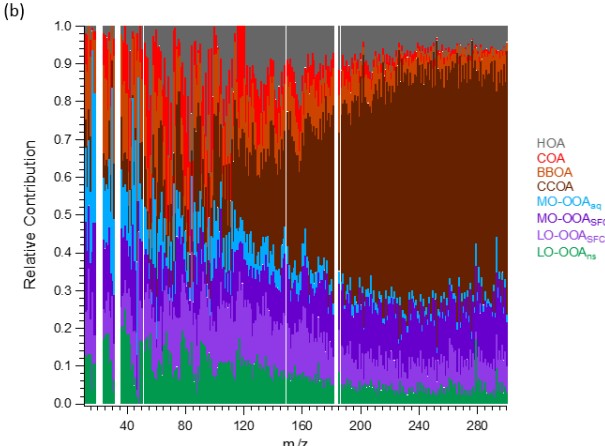

Figure 2. Averaged mass spectra (a) and relative contributions (b) of the 8-factor solution from the
AMS PMF bootstrap result. The mass spectra consist of HR ions from $m/z$ 12 to 120, and integrated
integer $m/z$ (denoted UMR) from $m/z$ 121 to 300. In (a), error bars denote standard deviation of each
stick calculated from all accepted bootstrap solutions.

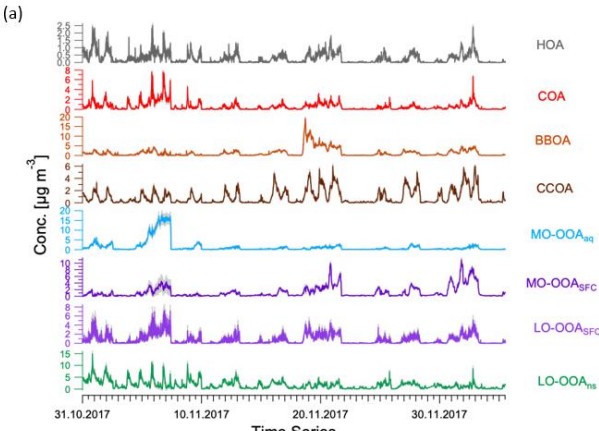

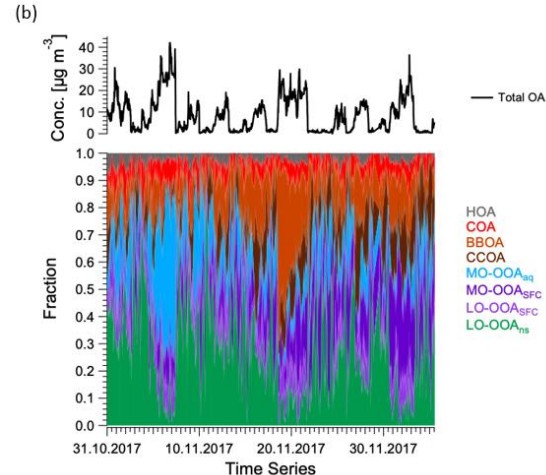

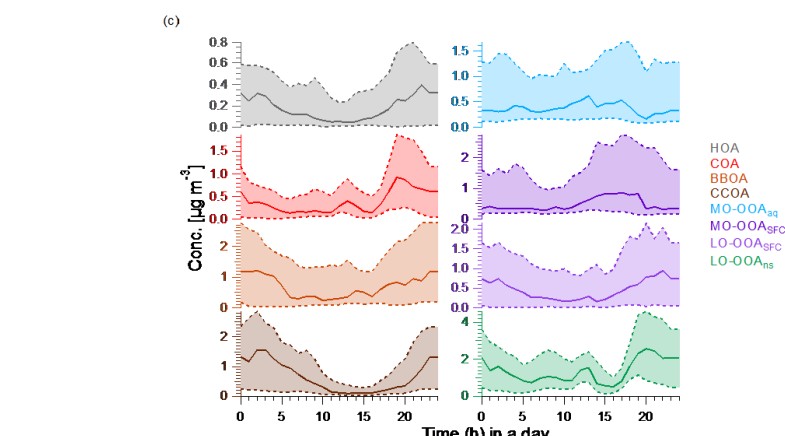

Figure 3. (a) Averaged time series with standard deviations (grey area), (b) averaged total OA concentration and relative contributions and (c) diurnal cycle the accepted AMS PMF bootstrap 8-factor solutions based on the criteria discussed in Sect. 2.3. Lower and upper dashed lines in (c) indicate the 1st and the 3rd quartile.

### 3.3 Investigation of factor composition by EESI-TOF

As discussed in Sect. 2.3, PMF of the EESI-TOF mass spectral time series was conducted on a 7-factor solution where all factor time series were constrained by the seven non-HOA factors retrieved from AMS PMF. This approach enables a more chemically specific interpretation of the retrieved AMS factors, which both supports POA factor identification and provides additional insight into the sources and processes governing SOA. The PMF result of the EESI-TOF time series was used as the base case for bootstrap runs, and all the bootstrap runs were retained for further analysis. EESI-TOF factor profiles (corresponding to AMS-derived factor time series) are interpreted by 1) comparison between these factor profiles and mass spectra retrieved from a chamber study using an EESI-TOF (Bertrand et al., 2020) and/or field studies (Qi et al., 2019; Stefenelli et al., 2019), 2) identification of key ions in the factor profiles by z-score analysis introduced in Sect. 2.3.3. The time series and factor profiles of the seven-factor solution are shown in Fig. 4.



We discuss the three primary factors in Sect. 3.3.1 and the four OOA factors individually in the
subsequent sections. For better interpretation, we present carbon number distribution plots from the
EESI-TOF factor profiles colour-coded by different families in Fig. 5 and Fig. 6 for the three POA
factors, and Figure 5 and Figure 6 for the four OOA factors respectively. In the carbon number
distribution plots, ions are classified first based on carbon numbers ($x$-axis) and ions with same
number of carbons are further divided into different categories based on H:C and O:C ratios (colour
code). Figure 7 shows Van Krevelen plots (atomic H:C vs. O:C ratio) for the four OOA factors based
on AMS factor profiles coloured by number of nitrogen atoms in each fragment, and sized by the
median z-score across all bootstrap runs, with large markers denoting ions having z-score > 1.5.
### 3.3.1  POA factors
**COA** – Consistent with Qi et al. (2019) and Stefenelli et al. (2019), the mass spectrum of this factor is
characterised by having most of the mass at ions with high $m/z$. These ions at high $m/z$ are likely long-
chain fatty acids or/and alcohols related to cooking emission and oils (Liu et al., 2017b). For example,
this factor is characterised by long-chain acids like $C_{18}H_{34}O_2^+$, $C_{19}H_{36}O_2^+$ and $C_{21}H_{38}O_3^+$, which
apportion 87.2 %, 76.2 %, and 92.3 % of their total mass to this factor, and they are also unique ions
in this factor, with z-scores of 2.61, 2.95 and 3.34, respectively.
**BBOA** – The mass spectrum of BBOA is characterised by a strong signal at $C_6H_{10}O_5$, corresponding
to levoglucosan and its isomers. Levoglucosan is a well-established tracer for primary aerosols
formed from pyrolysis of cellulose in biomass burning activities. This ion contributes 6.6 % to the
mass in this factor, about 4.5 times higher than the second strongest ion, consistent with previous field
and laboratory measurements of biomass burning by the EESI-TOF. Both Qi et al. (2019) (winter
measurements in Zurich, Switzerland) and Bertrand et al. (chamber study of wood burning emissions)
showed levoglucosan and its isomers to be the dominant ion in EESI-TOF spectra of primary wood
burning, with contributions of 13 % and 21 % respectively. In addition, the ion series $C_{10}H_{14}O_x$ ($x \geq 4$)
is observed in the BBOA and aged-SFC factors, consistent with Qi et al. (2019).
**CCOA** – as shown in the carbon number distribution plots (Figure  and Figure ), lower H:C and O:C
ratios are observed compared to other factors, especially for species with more than 10 carbons, ,
suggesting increased contributions from aromatic acids. This is consistent with Zhang et al. (2008)
who found that particles generated from industrial boilers typically contain a considerable fraction
from both aromatic acids and aliphatic acids. Note that PAHs, which comprise the unique AMS
spectral marker, are not detectable by the EESI-TOF extraction/ionisation scheme used here.





(a)

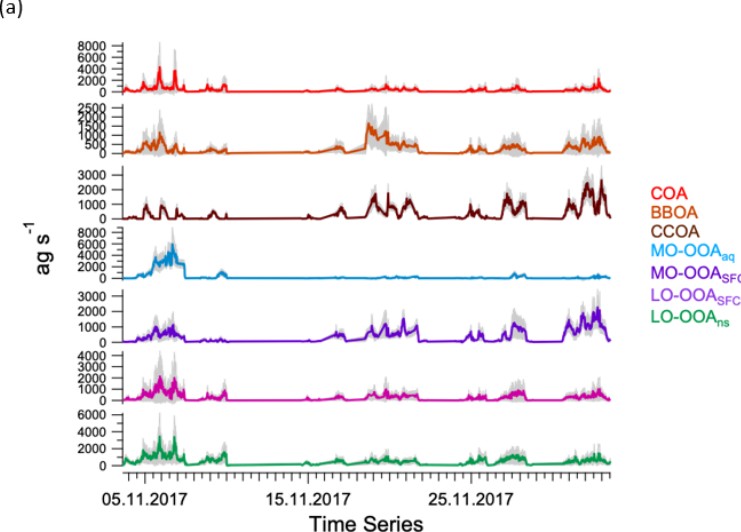

(b)

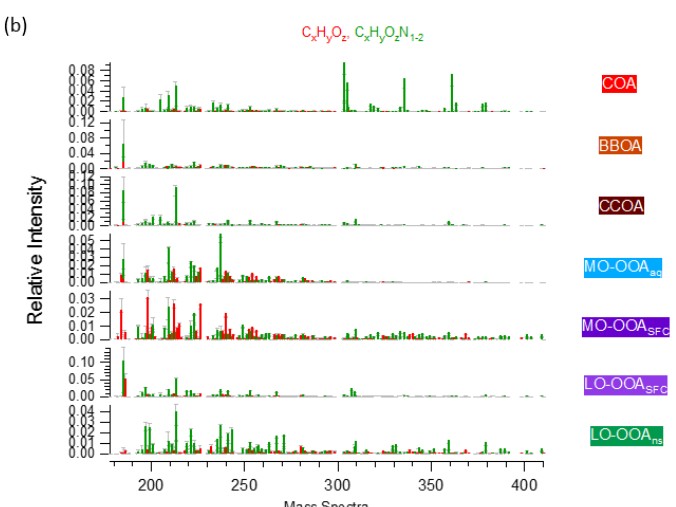

Figure 4. The averaged (a) time series and (b) mass spectra of accepted solutions from combined
bootstrap/$a$-value analysis of the EESI-TOF dataset. EESI-TOF time series are constrained by the 7
non-HOA factors retrieved from AMS PMF analysis. Shaded area in (a) indicates the anchor of
bootstrap/$a$-value analysis as shown in Eq. (7) and in (b) indicate the standard deviation of each stick
calculated from all selected solutions.



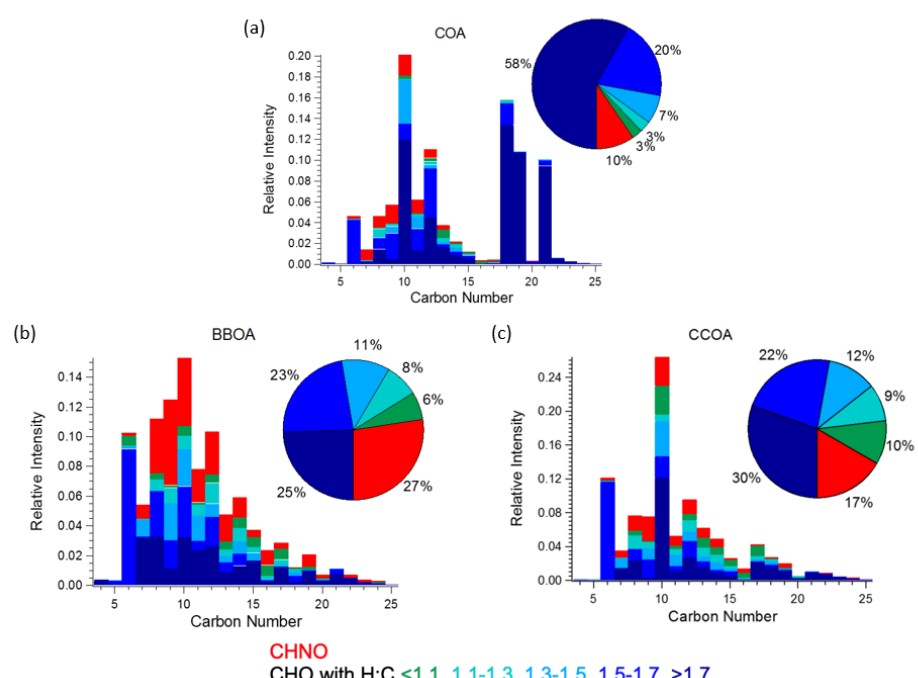

2 Figure 5. Carbon number distribution plots of three primary factors coloured by $C_xH_yO_zN_{1-2}$ and five
3 different $C_xH_yO_z$ categories based on H:C ratio (H:C < 1.1, 1.1 < H:C < 1.3, 1.3 < H:C < 1.5, 1.5 <
4 H:C < 1.7 and H:C > 1.7). The sum of each distribution is normalised to 1.

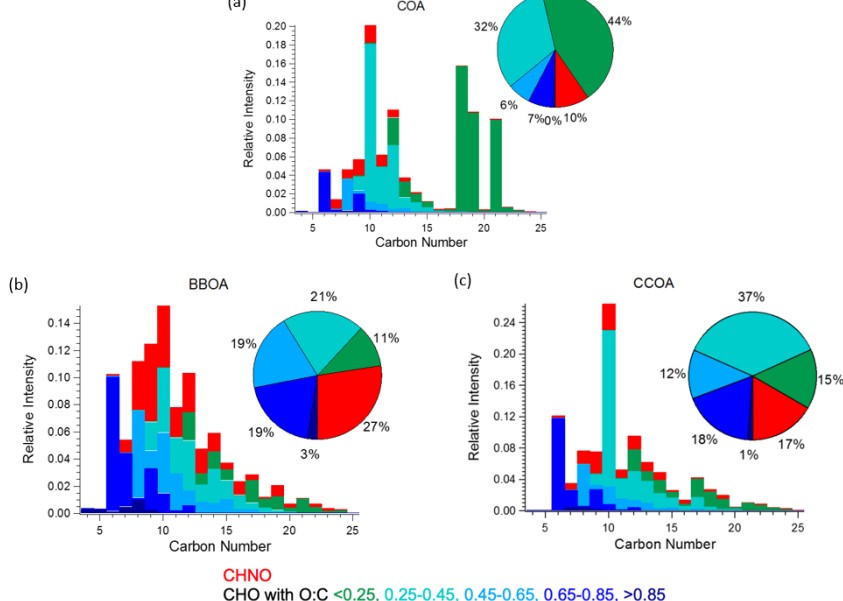



Figure 6. Carbon number distribution plots of three primary factors coloured by $C_xH_yO_zN_{1-2}$ and five
different $C_xH_yO_z$ categories based on O:C ratio (O:C < 0.25, 0.25 < O:C < 0.45, 0.45 < O:C < 0.65,
0.65 < O:C < 0.85 and O:C > 0.85). The sum of each distribution is normalised to 1.
### 3.3.2 MO-OOA$_{SFC}$
As noted in Sect. 3.2, the AMS MO-OOA$_{SFC}$ mass spectrum is consistent with OOA factors
characteristic of SOA, and represents aged, oxygenated emissions from solid fuel combustion. The
carbon number distribution of the EESI-TOF MO-OOA$_{SFC}$ mass spectrum (Fig. 8b) shows several
notable features that provide further insight into its source. First, the contribution of $C_xH_yO_z$ ions with
low H:C is significantly higher than for the other OOA factors. Specifically, $(C_xH_yO_z)_{H:C\leq1.3}$ comprises
12 % of the total signal and 22 % of $C_xH_yO_z$; for the other non-SFC related OOA factors,
$(C_xH_yO_z)_{H:C\leq1.3}$ comprises a maximum of 8 % of the total signal and 10 % of $C_xH_yO_z$. The high
fraction from low H:C ratio ions is consistent with other field studies using the EESI-TOF, and
suggests a higher contribution from aromatic precursors relative to the other OOA factors. The
$(C_xH_yO_z)_{H:C\leq1.3}$  is consistent with that of aged wood burning factors retrieved during winter in Zurich
(13-14%, Qi et al., 2019) (Fig. S14). Aged wood burning factors were also retrieved from source
apportionment of wintertime EESI-TOF measurements in Magadino, located in a Swiss alpine valley
(Stefenelli et al., *in prep*), where $(C_xH_yO_z)_{H:C\leq1.3}$ comprises 9-23 % of the total signal. Different from
the aged biomass burning factors found in Zurich and Magadino, $C_6H_{10}O_5$ is not observed in MO-
OOA$_{SFC}$, but other ions found in the aged biomass burning factors from Qi et al. (2019) and Stefenelli
et al., *in prep* (2019) including $C_{10}H_{16}O_x$ (x = 3,4,5,6…), are also apportioned to SFC-related factors
in the present study. Still, the $C_xH_yO_z$ distribution in the MO-OOA$_{SFC}$ factor retrieved in Beijing
differs from the previous studies in Switzerland in terms of the overall carbon number distribution.
Specifically, the Swiss measurement in Magadino featured by biomass burning activities (Stefenelli,
2019) showed a peak at $C_6$ and a peak from $C_8$ to $C_{10}$, the chamber study on coal combustion
oxidation (Bertrand et al.) exhibits a peak from $C_6$ to $C_{12}$ whereas in Beijing the signal is spread over a
much larger range (approximately $C_7$ to $C_{19}$).
Also evident from Fig. 8 is the high contribution from $C_xH_yO_zN_{1-2}$ ions, which comprise ~46 % of the
total signal. This is significantly higher than the 18-25 % observed in the Zurich factors by Qi et al.
(2019) but comparable to 35-41 % observed in Magadino. As above, the carbon number distribution
of $C_xH_yO_zN_{1-2}$ differs between Beijing and Switzerland, although the trends are reversed. In Beijing,
the $C_xH_yO_zN_{1-2}$ signal occurs mostly in the $C_6$ to $C_{10}$ range with a contribution of 73 % to total
$C_xH_yO_zN_{1-2}$ signal, whereas for the Swiss measurements it spans $C_8$ to $C_{10}$ with a contribution of 46 %
at most to total $C_xH_yO_zN_{1-2}$ signal and almost evenly distributes into other bins.  High intensity
$C_xH_yO_zN_{1-2}$ ions in Beijing MO-OOA$_{SFC}$ include $C_6H_{11}NO_4$, $C_7H_{13}NO_4$, $C_8H_{15}NO_4$, $C_9H_{17}NO_4$ and
$C_{10}H_{19}NO_4$. The high nitrogen content in MO-OOA$_{SFC}$ likely reflects high $NO_x$ concentrations in the
Beijing region during wintertime. In addition, ions tentatively attributed to nitrocatechol ($C_6H_5NO_4$)
and its homologous series ($C_7H_7NO_4$, $C_8H_9NO_4$) are apportioned predominantly to this factor and
CCOA (see Fig. S16), indicating the influence of oxidised aromatics from coal combustion emissions
(Mohr et al., 2013).
Interestingly, the AMS MO-OOA$_{SFC}$ profile and Van Krevelen plot (Fig. 7) show that the ions for
which MO-OOA$_{SFC}$ has a high z-score (>1.5) predominantly exhibit low H:C ratios. These ions
include $C_7H_2O^+$, $C_7H_3O^+$, $C_7H_4O^+$, $C_7H_5O^+$, $C_8H_4O^+$ and $C_8H_5O^+$. Although these ions are not
addressed in OOA factor separation in most AMS PMF studies due to their low intensities, their high
z-score in the present work suggests they may contain some source-specific information. The
temporal evolution of these ions is consistent with the EESI-TOF ions with low H:C ratio, e.g.
$C_{12}H_{10}O_8$ and $C_{16}H_{14}O_6$ (see Fig. S16). This also suggests an elevated contribution from aromatic
oxidation relative to the non-SFC-derived SOA factors.
### 3.3.3 LO-OOA$_{SFC}$





The LO-OOA$_{SFC}$ factor mass spectrum is also consistent with solid fuel combustion, but is less
oxygenated than MO-OOA$_{SFC}$. The carbon number distribution of the EESI-TOF MO-OOA$_{SFC}$ mass
spectrum (Fig. 8b) shows a contribution of $C_xH_yO_z$ ions with low H:C comparable to that of MO-
OOA$_{SFC}$. Specifically, $(C_xH_yO_z)_{H:C\leq 1.3}$ comprises 11 % of the total LO-OOA$_{SFC}$ signal, compared to
12 % from MO-OOA$_{SFC}$. This is consistent with less-aged biomass burning (LABB) factors retrieved
from source apportionment of wintertime EESI-TOF data in Zurich and Magadino, where
$(C_xH_yO_z)_{H:C\leq 1.3}$ contributed 10-16 %. LO-OOA$_{SFC}$ contains a substantial contribution (8 %) from
$C_6H_{10}O_5$ (levoglucosan and its isomers), which is substantially higher than that of MO-OOA$_{SFC}$ (0 %)
and LO-OOA$_{ns}$ (0 %) but lower than for primary BBOA (9 %) and CCOA (12 %). Interestingly, this
factor has a very high fraction (32 %) from $(C_xH_yO_z)_{H:C\geq 1.7}$, significantly higher than the 12 % to 14 %
observed in Zurich and Magadino. It also has 19 % contribution from $(C_xH_yO_z)_{O:C\geq 0.65}$, half of the
fraction (~40 %) of the LABB factors in Zurich and Magadino. The high H:C (1.66) and low O:C
(0.40) from EESI-TOF result in low averaged carbon oxidation states $\overline{OS}_c$ (-0.87) of this factor
suggests this factor is less oxygenated than the LABB factors in those two studies, with lowest $\overline{OS}_c$ of
15    -0.60.

Regarding nitrogen-containing species, $C_xH_yO_zN_{1-2}$ ions contribute 23 % to the total signal in this
factor, similar to their contributions in the Zurich and Magadino LABB (17 % to 22 %). However, in
Beijing a large fraction (8 %) of the $C_xH_yO_zN_{1-2}$ derives from a single ion ($C_6H_{11}NO_4$). Otherwise, the
carbon number distribution of $C_xH_yO_zN_{1-2}$ ions in Beijing is weighted from $C_7$ to $C_{10}$, consistent with
SOA from wood burning experiments with OH or $NO_3$ (Bertrand et al.) as shown in Fig. S17. Similar
to the primary BBOA and CCOA factors, LO-OOA$_{SFC}$ is elevated overnight, suggesting a
contribution from nighttime chemistry and/or rapid oxidation of primary emissions.
### 3.3.4    MO-OOA$_{aq}$
The MO-OOA$_{aq}$ factor time series is dominated by high absolute and relative concentrations during a
haze event in the non-heating season. Both the atmospheric conditions during this event and the
overall factor composition are consistent with a strong influence from SOA formed by aqueous phase
chemistry.
Figure 10a shows the time series of the $CO_2^+$ and $CO^+$ ions measured by the AMS, with the
corresponding scatter plot shown in Fig. 10b. For most of the data, the ratio of $CO^+$ to $CO_2^+$ is
approximately 1, consistent with the mean $CO^+/CO_2^+$ value for bulk atmospheric OA (Canagaratna et
al., 2015; Aiken et al., 2008) and the assumption in the standard AMS fragmentation table. In
contrast, the $CO^+/CO_2^+$ slope is only 0.5 for the haze event on 4 to 7 November. This relative
enhancement of $CO_2^+$ is characteristic of small acids or diacids, e.g. oxalic acid, malonic acid and
succinic acid (Canagaratna et al., 2015). Such molecules are mainly derived from aqueous phase
chemistry (Tan et al., 2012; Tan et al., 2010; Carlton et al., 2007; Ervens et al., 2004), although minor
contributions might also be possible from soluble species in aerosols formed by gas-phase reactions
(Legrand et al., 2005; Sellegri et al., 2003; Sempere and Kawamura, 1994).
An enhanced contribution from small acids is also suggested by the EESI-TOF MO-OOA$_{aq}$ profile.
As shown in Figs. 7 and 8, MO-OOA$_{aq}$ has enhanced signal from ions with low carbon number
relative to the other OOA factors. Further, Fig. 8 shows that these low-C ions are highly oxygenated
(e.g. $C_6H_6O_5$), which is likewise consistent with small multifunctional acids and polyacids. The EESI-
TOF spectra thus provide further support for the attribution of this factor to SOA generated from
aqueous-phase chemistry. Note that due to the application of the volatility-based filter for
distinguishing particle-phase vs. spurious ions (see section x), the contribution of such small, highly
oxygenated ions presented here represents a lower limit.
As shown in Fig. 3, MO-OOA$_{aq}$ provides a major fraction of 41 % to the total OA during the major
haze event on 4 to 7 November (peak concentration > 40 µg m$^{-3}$). In fact, OA concentrations during





this event are at least as high as those observed during the heating period, despite the likelihood of
reduced concentrations of precursor VOCs due to the mandated reductions in combustion activities
related to domestic heating in rural areas. We therefore investigate the reasons for the high aqueous
SOA production during this specific event. As aqueous phase chemistry is typically associated with
high aerosol liquid water content (LWC), LWC was calculated from ISORROPIA-II (Fountoukis and
Nenes, 2007). The LWC concentration is presented in Fig. 10, together with the time series of MO-
$OOA_{aq}$. The two time series are strongly correlated ($r^2$ =of 0.93), and both are dramatically higher
during the 4 to 7 November event than for the rest of the study. Note that the strong correlation
between MO-$OOA_{aq}$ and LWC is not driven solely by the event on 4 to 7 November; rather, the two
time series are remarkably well correlated throughout the entire campaign. This further supports the
interpretation of MO-$OOA_{aq}$ as characteristic of aqueous SOA production throughout the campaign,
rather than being characteristic of only a single event.
The question arises whether MO-$OOA_{aq}$ reflects the irreversible production of SOA via aqueous
pathways, or instead reversible solvation of volatile and semivolatile organics. To assess this, we look
in detail at the MO-$OOA_{aq}$ and LWC correlations during the 4 to 7 November event. The most
significant disagreement between the time series occurs from 08:00 to 23:00 on 6 November, when
the LWC sharply decreases while MO-$OOA_{aq}$ remains high. If MO-$OOA_{aq}$ were driven by reversible
solvation, this extended decrease in LWC would be expected to drive a corresponding decrease in
MO-$OOA_{aq}$. However, the MO-$OOA_{aq}$ concentrations appear unaffected by the decrease in LWC,
suggesting that the MO-$OOA_{aq}$ does indeed consist of irreversibly-generated SOA via aqueous
chemistry.
The reasons for the high LWC are driven by the combination of high RH and high inorganic fraction
(especially $NH_4NO_3$), which as shown in Fig. 1 are both maximised during this period. The high
$NH_4NO_3$ content during 4 to 7 November is in turn driven by a unique airmass source region. Figure
11a shows 72-h backward trajectories calculated from the HYSPLIT transport model (Rolph et al.,
2017; Stein et al., 2015), and analysed in Zefir v 4.0 (Petit et al., 2017). Trajectories are coloured by
date and time. In the figure, trajectories from 4 to 7 November pass over regions of high $NO_x$
emissions to the east and south of Beijing (Shandong and Henan provinces) before arriving at the
sampling site. The air parcel spends approximately 30 hours over these high-$NO_x$ regions, as shown in
Fig. 11b. As shown in Fig. S18, the period of 4 to 7 November is the only time in the campaign where
the back trajectories pass over this region.  Due to the high $NO_2$ concentration and high RH in this
period, particulate nitrate is produced during this regional transport homogeneously and/or
heterogeneously, resulting in water uptake and high LWC in the aerosol phase. The high LWC in turn
facilitates further heterogeneous formation of nitrate. This positive feedback provides favorable
conditions for efficient aqueous chemistry and thus production of MO-$OOA_{aq}$ (Kuang et al., 2020).
### 3.3.5   LO-$OOA_{ns}$
In Sect. 3.2, this factor has been identified as LO-OOA because of its moderately high $CO_2^+$ signal
and non-negligible contribution from the $C_xH_y$ group. The time series of this factor shows clear
diurnal variation which peaks at around 20:00 in the non-heating season (Fig. 3a and 3c), but this
variation is not clear in the heating season. In addition, the contribution of this factor to total OA is
higher in the clean period than during the haze events (Fig. 3b), indicating this may be related to
regional sources/processes rather than more local SFC emissions. The diurnal cycle of this factor is
similar to COA and LO-$OOA_{SFC}$, but the chemical characteristics of these three factors are different.
Compared to LO-$OOA_{SFC}$, this factor is characterised by ions with high H:C and low O:C and does
not have $C_6H_{10}O_5$, a key ion in SFC-related LO-OOAas identified in both the present and previous
studies. LO-$OOA_{ns}$ also does not have large contributinos from ions with the aromatic feature of low
H:C. Although the spectrum of COA is also characterised by ions with high H:C and low O:C, the
carbon number distribution plots of COA are characterised by significant signal from long-chain acids





at high carbon number, whereas the carbon number distribution of this factor is characterised by high
signal at low carbon number (from $C_8$ to $C_{12}$). Compared to other OOA factors, this factor has the
lowest O:C ratio (0.33) and highest H:C ratio (1.69). Since it is not characterised by any key ions to
our knowledge on previous EESI-TOF studies (e.g. levoglucosan and its isomers), this factor is named
as LO-OOA$_{ns}$, representing non-source-specific LO-OOA.

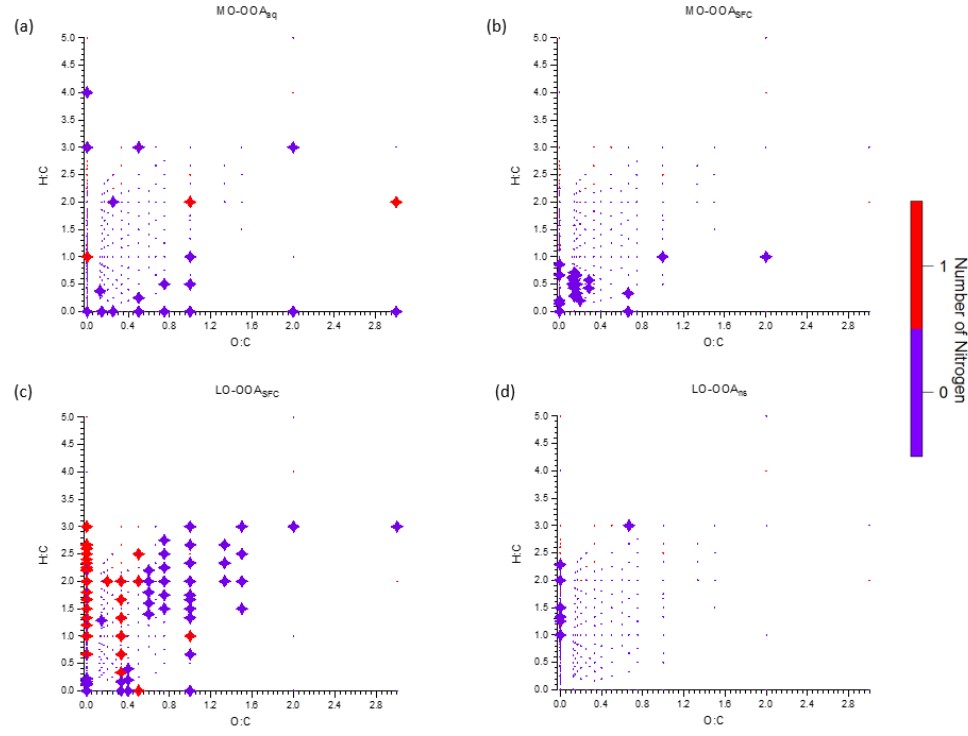

Figure 4. Van Krevelen plot of (a) MO-OOA$_{aq}$, (b) MO-OOA$_{SFC}$, (c) LO-OOA$_{SFC}$ and (d) LO-OOA$_{ns}$,
coloured by the number nitrogen atoms in the AMS fragments. Large symbols denote s with median
z-score > 1.5 for accepted runs from bootstrap analysis.



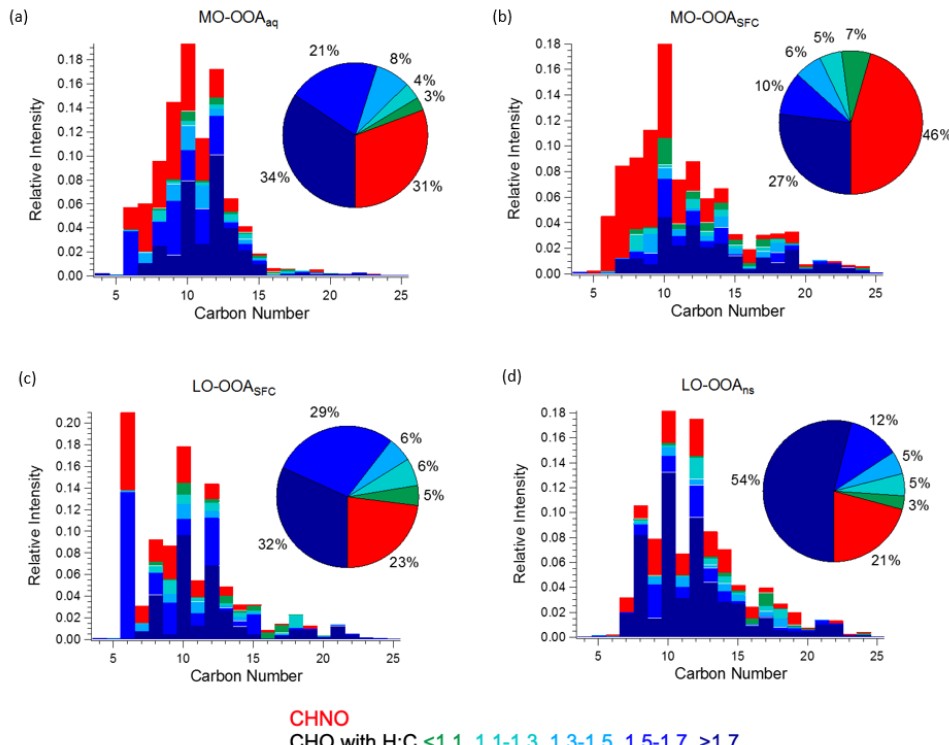

CHNO
CHO with H:C <1.1, 1.1-1.3, 1.3-1.5, 1.5-1.7, >1.7

Figure 5. Carbon number distribution plots of four OOA factors coloured by $C_xH_yO_zN_{1-2}$ (red) and five different $C_xH_yO_z$ categories (green to blue) based on H:C ratio (H:C < 1.1, 1.1 < H:C < 1.3, 1.3 < H:C < 1.5, 1.5 < H:C < 1.7 and H:C > 1.7).





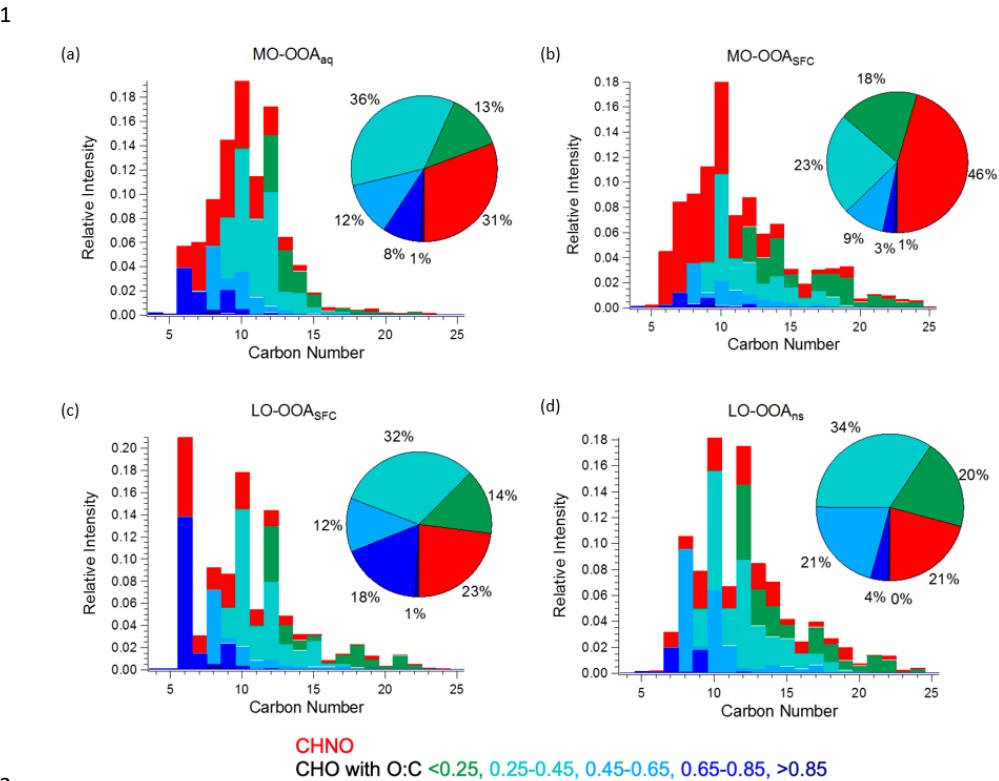

Figure 6. Carbon number distribution plots of four OOA factors coloured by $C_xH_yO_zN_{1-2}$ (red) and five
different $C_xH_yO_z$ categories (green to blue) based on O:C ratio (O:C < 0.25, 0.25 < O:C < 0.45, 0.45 <
O:C < 0.65, 0.65 < O:C < 0.85 and O:C > 0.85).

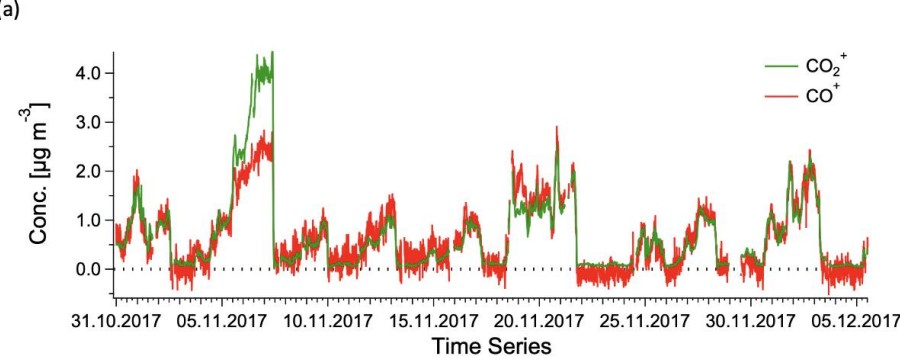

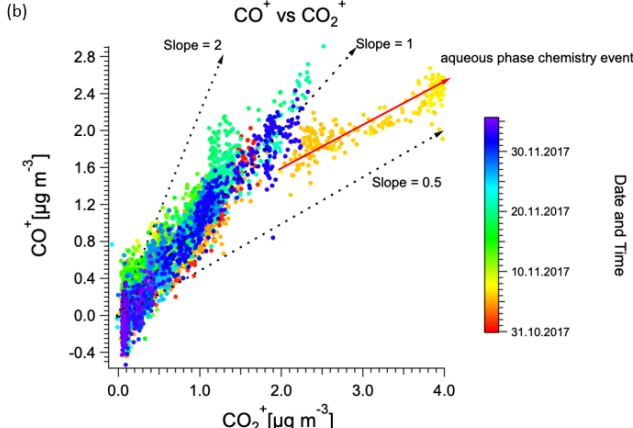

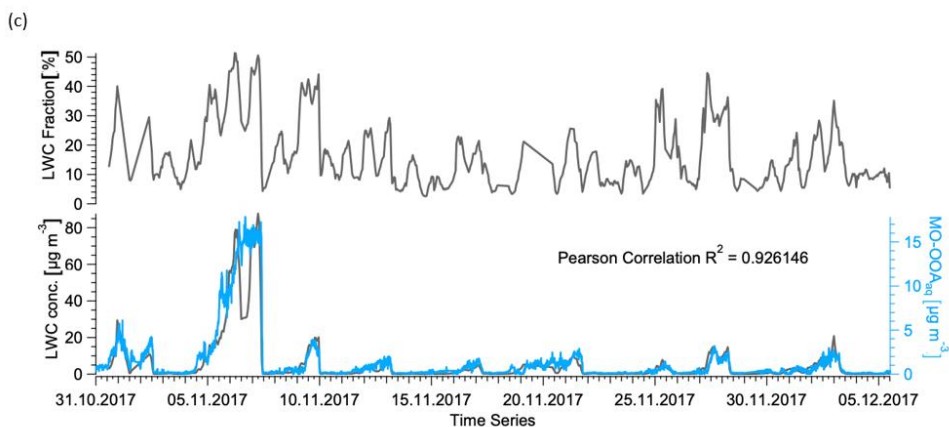

Figure 7. (a) Time series of $CO^+$ and $CO_2^+$ throughout the campaign after correction for $CO_2^+$
contribution from $NH_4NO_3$ to total $CO_2^+$ and (b) scatter plot of $CO^+$ and $CO_2^+$ indicating a different
slope for the haze event between 4 November to 7 November 2017, suggesting aqueous phase
chemistry may happen in this period. (c) Time series of LWC, both in fraction (top) and mass
concentration (bottom), complemented by MO-OOA$_{aq}$, demonstrating the high correlation between
the latter two variables.

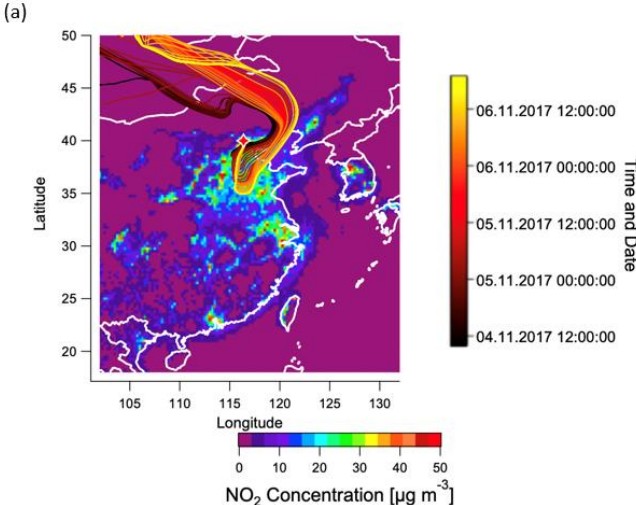

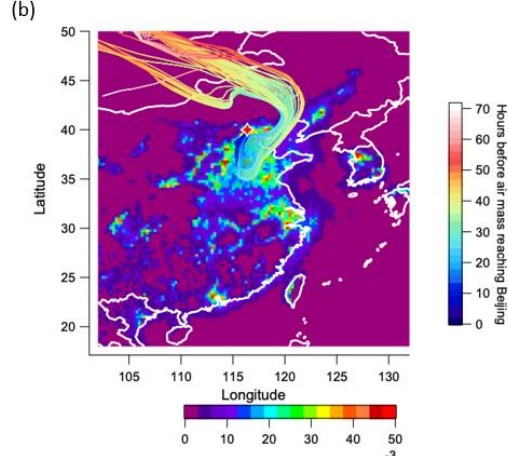

Figure 8. Trajectories analysis. (a) 72-h back-trajectories (HYSPLIT) for the haze event from 4 to 7
November colour-coded by date and time, (b) 72-h back-trajectories for the haze event from 4 to 7
November colour-coded by hours before the air mass reaches Beijing. In both figures, trajectories are
overlaid on a 2015 map of surface $NO_2$ concentrations based on the CHIMERE model and driven by
the 2015 DECSO inventory (Liu et al., 2018).

## 4. Atmospheric implications

As discussed in Sect. 3.1, meteorological conditions are responsible for an alternating occurrence of
haze and clean periods and these effects from meteorology are well-understood (Duan et al., 2020;
Duan et al., 2019; Zhao et al., 2019; Xu et al., 2019; Sun et al., 2016a; Sun et al., 2016b). In addition,



meteorology can also influence air mass trajectories on the regional/mesoscale, which may further influence the aerosol chemical composition. By comparing measurements before and after the start of the heating season (15 November), the effects of heating emissions on clean and haze periods in Beijing can be assessed. Figure 12 shows the time series of total OA and the contribution of different factors to each haze event, suggesting that different seasons are influenced by different sources/processes.

Clean periods in both the non-heating and heating seasons are dominated by SOA, comprising 77 % in the non-heating season and 71 % in the heating season. In both seasons, the single largest component is LO-OOA$_{ns}$ (45% and 33% in the non-heating and heating seasons, respectively), consistent with its identification as regional SOA not specific to a single emissions source. The SFC fraction is higher in the heating season, with CCOA and BBOA jointly comprising 22% (vs. 15% in the non-heating season) and LO-OOA$_{SFC}$ and MO-OOA$_{SFC}$ jointly comprising 25% (vs. 19% non-heating).

Seasonal differences become more pronounced under haze conditions. Three minor haze events (maximum concentrations between x and y) were observed in each season. During the non-heating season, LO-OOA$_{ns}$ remains the single largest component (33 to 43%), although its fraction is slightly reduced. There is no corresponding fractional increase observed in any of the other factors, but rather an across-the-board relative increase in all, which results in a slightly increased POA fraction (29 to 38%, vs. 23% under clean conditions). These changes likely result from an increased role of local emissions and reactivity under the stagnant conditions giving rise to haze. The non-heating minor haze events contrast strongly with the heating minor haze, where there is a larger reduction in the LO-OOA$_{ns}$ fraction (21 to 30%) that corresponds specifically to increased SFC POA (27 to 36%). Interestingly, the SFC SOA fraction is not significantly larger than under clean conditions (21 to 26%).

In general, the minor haze events within a given season are relatively similar to each other. However, significant differences in composition are observed between the minor and major haze events within a given season. The two major haze events occurring within the heating season are also quite different from each other. The conclusions that can be drawn from this observation are limited by the small number of major haze events sampled (1 non-heating, 2 heating), but suggest the potential for unique meteorological/transport phenomena that may affect sources and composition during the most extreme events. For example, the non-heating haze event (4 to 7 Nov.) is dominated by MO-OOA$_{aq}$ from aqueous processes (41% of OA), and as discussed in the previous section corresponds to unique airmass back-trajectories over high-NO$_x$ regions. The event from 18 to 22 Nov. is dominated by SFC, especially BBOA, which comprises 36% of OA (with CCOA contributing an additional 13%), while SFC SOA comprises an only slightly larger fraction (28% of OA) than under clean conditions. In contrast, the major haze event from 30 Nov. to 3 Dec. has a large contribution from both SFC POA (33%) and SFC SOA (40%). Interestingly, the temporal evolution of these two events is also different, with the 18 to 22 Nov. event (high SFC POA) commencing with a sudden concentration increase but remaining relatively stable thereafter, while concentrations during the 30 Nov. to 3 Dec. event (high SFC POA and SOA) increase gradually over multiple days. However, a close inspection of the 18 to 22 Nov. event in Fig. 3b shows a decrease in the BBOA fraction and increase in MO-OOA$_{SFC}$ as the event proceeds, suggesting a generally important role for local SOA formation in a stagnant airmass during the course of a haze event.

As a conclusion, our observation suggests that the sources and processes giving rise to haze events in Beijing are variable and seasonally-dependent. Two salient features are: 1) in the heating season, SOA formation is driven by oxidation of aromatics from solid fuel combustion (with the contribution from 41% to 71% from SFC-related OOA factor to total SOA), 2) under high NO$_x$ and RH conditions, aqueous phase chemistry may have a major contribution to SOA formation (with the contribution of



54% from MO-OOA$_{aq}$ to total SOA). The combination of high inorganic content and aqueous SOA
can yield total mass concentrations comparable to those observed in the heating season, despite
reduced regional VOC emissions in the absence of heating processes.
Our back-trajectory analysis shows that from 4 to 7 November, the air masses passed through a high
NO$_2$ concentration region and stayed for more than 24 hours in this region (Fig. 11), which facilitated
nitrate formation in the aerosol phase and thus water uptake. Therefore, our observation suggests that
meteorology cannot only influence the haze evolution on a local scale, but also can have significant
effect on aerosol chemistry and chemical composition by influencing the origin and pathway of air
mass.
From a technical perspective, a surprising outcome of this source apportionment analysis was the
extent to which the AMS SOA factor profiles contained source-related information corroborating the
chemically more specific measurements of the EESI-TOF. Specifically, the SFC-related factors
exhibited systematic enhancements in ions with low H:C ratios, while the $CO^+/CO_2^+$ ratio clearly
higher than 1 was found to be a clear indicator for aqueous-phase chemical processing. Although the
latter observation requires the improved mass resolution of the L-ToF-AMS and is therefore not
retrievable from most existing AMS datasets, the former suggests that AMS SOA spectra may contain
more source-specific information than is typically recognised. Although these results represent a
single case study and so should not be overinterpreted, we suggest that intensity-independent
statistical tools such as the z-score analysis employed here may be effective in retrieving such
information and in providing additional insight into SOA sources. The combination of quantitative
AMS data with semi-quantitative EESI-TOF measurements is also shown to be promising, and
alternative methods for combining such datasets (e.g. as discussed in the Methods section) should be
pursued.





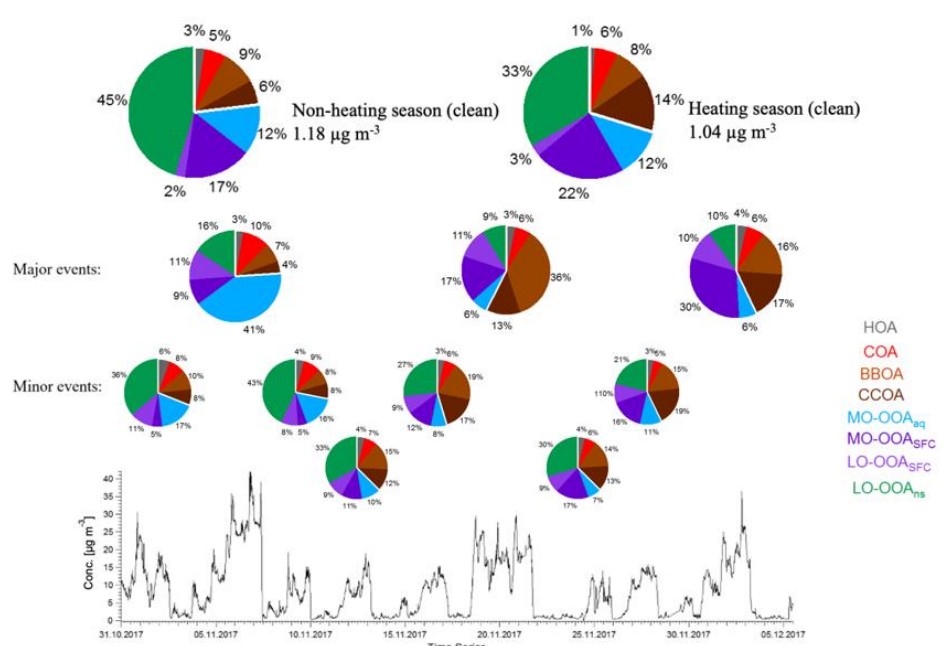

Figure 9. Time series of total OA and the mean contribution of eight AMS factors in each haze event
and clean periods for the non-heating and heating periods. The top two pie charts indicate the
averaged contributions for clean periods in non-heating season and heating season.
**5.  Conclusions**
OA sources were investigated in Beijing during an intensive field deployment of AMS and EESI-TOF
instruments from late September to mid-December 2017, covering the transition from the non-heating
to heating seasons. This represents the first deployment of the EESI-TOF in a heavily polluted city.
The robust quantification of the AMS and high chemical resolution of the EESI-TOF are shown to be
highly complementary, facilitating identification of the sources and processes governing SOA
concentrations. An integrated source apportionment study was conducted, by running PMF on AMS-
only data first to determine factor time series, followed by PMF on EESI-TOF-only data with
constrained factor time series to facilitate chemical interpretation of the AMS-determined factors,
which successfully resolved and interpreted four SOA sources and processes.
The source apportionment analysis yielded four primary factors and four secondary factors. Primary
factors were hydrocarbon-like OA (HOA) characterised by a high fraction of hydrocarbon fragments,
cooking-related OA (COA) characterised by long-chain fatty acids, biomass burning OA (BBOA)
with a high contribution from levoglucosan, and coal combustion OA (CCOA) with a high PAH
signal at high $m/z$ range. The secondary factors consisted of more- and less-oxygenated oxygenated
organic aerosol from solid fuel combustion (MO-OOA$_{SFC}$ and LO-OOA$_{SFC}$), more-oxygenated aerosol
from aqueous-phase chemistry (MO-OOA$_{aq}$), and less-oxygenated OA from mixed or indeterminate
sources (LO-OOA$_{ns}$). The SFC-related factors were characterised by a low H:C ratio in both the
EESI-TOF and AMS spectra and increased concentrations during the heating period. MO-OOA$_{aq}$ was
characterised by an increased contribution from small, highly oxygenated ions and a low AMS



$CO^+/CO_2^+$ ratio; taken together, these observations suggest an enhanced contribution from small acids
and diacids.
The OA composition in Beijing is dominated by organic aerosols, with a significant SOA fraction
($66.4\pm13.5$ %) to total OA throughout the campaign. SOA formation during the heating season
derives mainly from solid fuel combustion. However, even during the non-heating season when solid
fuel combustion was not a major source, an intense haze event was observed with OA concentrations
comparable to the highest concentrations observed during the heating season. These high
concentrations were due to significant SOA production from aqueous phase chemistry, and
corresponded to the passage of air parcels over the high $NO_x$ regions to the east and south of Beijing.
This suggests that aqueous chemistry may provide a major contribution to SOA formation under
certain meteorological conditions, even during periods of intense haze
*Competing interests*. The authors declare that they have no conflict of interest.
*Acknowledgements*. We gratefully acknowledge the contribution from Dr. LIU Fei from NASA for
providing $NO_x$ emission map of China. We also acknowledge the NOAA Air Resources Laboratory
(ARL) for the provision of the HYSPLIT transport and dispersion model and READY website
(https://www.ready.noaa.gov) used in this publication. Logistical support by André TEIXEIRA (Paul
Scherrer Institut) and Dr. WU Yunfei (Institute of Atmospheric Physics, Chinese Academy of
Sciences), and coordination support by Prof. CHEN Chunying (National Center for Nanoscience and
Technology, Chinese Academy of Sciences) are gratefully acknowledged. This study was funded by
the Swiss National Science Foundation starting grant BSSGI0_155846 (IPR-SHOP), the EU Horizon
2020 Framework Programme via the ERA-PLANET project SMURBS (grant agreement no. 689443),
the National Research Program for Key Issues in Air Pollution Control (DQGG0105), the Natural
National Science Foundation (21661132005), the National Natural Science Foundation of China
(NSFC) under grant no. 41925015 and the Chinese Academy of Sciences (no. ZDBS-LY-DQC001).

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
