# Peer review of "Quantification of solid fuel combustion and aqueous"

_Atmospheric Chemistry and Physics, 2020_

## Referee Comment (RC1) · Anonymous Referee #1 · 23 Sep 2020

The manuscript presents analysis of EESI-TOF-MS and AMS measurements carried out in Beijing during autumn and winter 2017. It aims to address important questions surrounding haze formation and is within the scope of ACP.

The authors use positive matrix factorisation (PMF) to identify factors corresponding to primary and secondary organic aerosol measured by AMS and use these factor time series to constrain a PMF analysis of the EESI-TOF-MS observations. Factor composition and time series are compared between heating and non-heating seasons, identifying varying contributions from aerosol components observed by the two instruments.

The authors identify solid fuel combustion and aqueous chemistry to be particularly important in the build-up of winter haze events.

The paper provides extensive technical recommendations for operation of the EESI-TOF-MS in highly polluted field measurement campaigns which will be a valuable contribution to the instrument user community. The analysis gives potentially useful information on sources and formation of organic aerosol, but I have some concerns about the approach taken. In addition, I think that the presentation and discussion of results requires significant improvement for publication in ACP so I would recommend major revisions. Major issues identified are discussed below, followed by detailed comments and suggestions.

General Comments:

PMF of the EESI-TOF-MS has been constrained by the time series derived from PMF of AMS measurements. This has the potential to introduce significant biases in the results, which should be addressed in the manuscript. EESI-TOF-MS detects a similar range of compounds to those detected by FIGAERO-CIMS and previous studies have shown distinct differences in the time series of CIMS factors compared with those derived from AMS (Chen et al. 2020). The additional chemical composition provided by the EESI-TOF-MS should enable more factors to be derived on the basis of chemistry occurring across the measurement period which the AMS does not have the chemical resolution to identify. Therefore the approach taken to constrain the time series of EESI-TOF-MS PMF factors could have influenced the identification of factors. The PMF of the AMS factors forms the basis of the results presented in the manuscript, and the use if EESI-TOF-MS data is supplementary and used simply to show the composition of the EESI which corresponds to the AMS factors. This is different to a true PMF analysis of the chemically insightful EESI-TOF-MS data and should be discussed accordingly. What are the correlations between the EESI-TOF-MS and AMS factor time series? How sensitive are the time series to the constrains. I understand that carrying out a separate PMF of the EESI-TOF-MS without constraining the time series is a

large body of work, however, you should try to discuss how not doing this could have influenced your results and interpretation.

The discussion of ions identified by EESI-TOF-MS is limited and needs to be referenced more extensively. Mass spectra from EESI-TOF-MS should be labelled and a more detailed list of ions provided in the supplement along with their potential sources. Correspondences drawn between AMS and EESI-TOF-MS ions should be better justified through a more thorough evaluation of the ions contributing to each factor. Time series of dominant ions contributing to each factor should be compared with that of the factor profile itself and its corresponding AMS factor in a Figure to show the validity of constraining the EESI-TOF-MS factor profiles by the AMS time series. CIMS and AMS factor time series should also be presented on the same axis for comparison.

Presentation, discussion and interpretation of factors is confusing and hard to follow, particularly with respect to comparison between AMS and EESI-TOF-MS factors due to their identical naming. Additionally, many of the figures need to be improved and should be referenced better throughout the manuscript. It would be useful if the haze events of interest and heating/non-heating periods were labelled in each figure. Figure 9, in particular, needs to be overhauled in favour of something much more easy to interpret. Perhaps stacked bars could be used to compare the contributions of different factors during different haze events. Interpretation of figures also requires some attention, with diurnals interpreted based on the time series' of their standard deviations as opposed to their average values. As presented, the figures hinder the interpretation of the manuscript and should be improved to emphasise conclusions.

Structure and language throughout the manuscript need to be improved. Placeholders for data values and figure numbers remain throughout. In addition, there are figures referenced which are not included, and those which are included are not referenced in the text thoroughly making the paper challenging to follow. Several sections including the introduction and methods and not well-structured and clear. The introduction needs to be streamlined and made clearer and referencing needs to be improved throughout,

particularly in the results section where it would be useful to discuss where ions identified by EESI-TOF-MS have been previously observed.

Specific Comments

Introduction:

The introduction is currently very hard to follow and lacks a coherent structure. The author begins by introducing organic aerosol in the first paragraph in which its key sources are outlined, and then in paragraph 2 they mix health effects with sources and instrumentation. This structure should be streamlined into a narrative which provides the information required to contextualise the results presented in the manuscript.

Lines 8-12 on page 2 are not referenced and mention uncertainties in quantification due to instrumentation, however, instrumentation is not introduced until line 48 of the same page. The few sentences about health impacts of ROS seem out-of-place in this paragraph.

Results from the AMS are mentioned in paragraph 2 on page 2, while several sentences describing it are on page 3 (lines 9 -16). I would suggest that the author introduces instrumentation first before talking about observations with those instruments in Beijing. The introduction should be re-written to better contextualise the more detailed aerosol source characterisation which the EESI provides, this is currently disjointed, with the key AMS factors the EESI is expected to better constrain being described on lined 25 - 42 of page 2, while the EESI itself is not described until page 3.

Consider splitting paragraph 2 on page 2 up with the discussion of Beijing separate from the broad discussion of AMS and ACSM capabilities. This should be made clearer.

Lines 32 - 33 on page 2 are not justified, there are a variety of previous studies in Beijing which study and apportion OA to different sources and formation mechanisms, inc. Bryant et al. (2020) and Wang et al. (2019).

The introduction of EESI on page 3 should explain better how it is able to capture detail which the AMS cannot, and which components of aerosol measured by the AMS are most poorly understood. Line 25 on page 3, include references.

Discussion of instruments on page 3 lacks accurate chronology, e.g. CHARON, FI-GAERO and GC/MS-FID.

Page 3 lines 35 - 38 are contradictory, you first say that the EESI-TOF provides a hard-to-quantify response, then go on to say it has the potential to provide a quantitative measurement of OA.

Methodology: Again the structure of this needs to be corrected, you discuss the exact co-ordinates of the measurement location and then go on to talk about Beijing in the context of the NCP. This needs to be more focused. A better structure would start with Beijing, move to the location of the site and then talk about its surroundings and potential influences.

Page 4 lines 6 - 8, you suggest that highway traffic and industry do not influence measurements at the site. However, given that pollution in Beijing is regional, and it is well known that industrial air masses from the SE affect the air quality in Beijing, this statement needs to be removed or better justified. Surely the site is influenced by both local emissions and regional transport? Page 4 lines 9-10, can you explain why this period from October - December was chosen when measurements were carried out from September. No details are provided on sampling flow rates, these need to be added.

Lines 7-10 page 5 - could the sensitivity differences be related to changes in relative humidity?

Lines 42 -43 page 5 - why is there not expected to be any compound dependant effect? Is this backed up by any literature or an assumption? Why is it valid to make this assumption?

Lines 9 -12 page 5 - here you state a " a diagnostic species" is used, and below this you state "inorganic nitrate species" - these need to be clarified in more detail.

Lines 29 - 30 page 5 - provide more discussion of the assumption that the time-dependant changes in sensitivity are eliminated for organics.

Lines 30 - 33 page 6 - here you state that particle-phase signals tend to be less oxy-genated than the background ions, while in supplement section S1 you state the opposite. This needs to be corrected.

Some of the details provided in this section may not be of interest to the typical reader and I would suggest moving more of the detailed instrument trouble shooting to the supplement as currently the paper is harder to follow as a result. The sections in the supplement are much clearer are will be easier to access for a technical reader who wants to understand the troubleshooting process. This level of detail in the main body of the manuscript itself shifts the focus from the results.

Line 42 page 6 - "as discussed in Sect. 2.3" - this should either be as "will" be discussed or should be removed.

Line 41 page 6 to line 9 page 7 - This discussion of PMF and its error seems to be out of place in this section, it should be moved to section 2.3.

Lines 15 -28 page 8 - explain and justify potential biases in the interpretation of EESI-TOF data by constraining time series' to those of AMS factors, given that EESI-TOF factors, similar to CIMS factors have previously provided factor TS which are distinctly different those from the AMS, e.g. Massolli et al (2018), Yan et al (2016) and Chen et al. (2020).

Check how you wish to refer to EESI through the manuscript, it varies between EESI-TOF and EESI-TOF-MS. Choose one.

Line 41 page 8 - determine should be "determination" Line 42 page 8 - "compare the" should be "comparison of" Line 44 page 8 - z-score is mentioned here, either do not

mention or signpost the reader to where it will be discussed.

Major comment - AMS factors and EESI-TOF or CIMS factors are not expected to be directly related to one another, by constraining each EESI factor to an AMS factor solution is it not possible that you are introducing a bias which limits the extent to which you are describing chemistry in your interpretation and are instead biasing the results towards primary sources.

Throughout "as discussed" refers to past tense, it should be changed to "will be discussed in".

Results:

Page 10 lines 46-47 - criteria used to define "haze episodes" should be defined here rather than in figure caption for Figure 1.

Page 10 line 20 - 37 % is written twice. Needs to be corrected.

Figure 1 - Several corrections are needed. 1) y-ticks are overlapping between (b) and (c ). 2) y-label for (c ) should specify if this is PM2.5 mass, which I assume is the case, and also include units in brackets. 3) y-label for (d) should be changed to fractional factor contribution or something more clear 4) Colorscale of W_dir should be labelled as wind direction, which is what I assume it is? It would also be better not to use rainbow. 5) Legend for light and severe haze should be made clearer.

Page 12 lines 9-10 - it should be specified which HR ions were used and where UMR sticks were used. A table in the supplement would be a good way to record this.

Page 12 line 18 - page 13 line 8- Here you discuss that the apportionment of 4 OOA factors is unusual, and point the reader to section 3.3. for interpretation. Is it the case that 4 factors are chosen in the AMS data to better represent the EESI-TOF data? Or were there factors chemically or temporally distinct enough in the AMS data alone that the EESI factors did not influence this selection? It is important to acknowledge how the EESI factor identification has influenced the factor selection for the AMS and vice

versa.

Figure 2a - y-ticks are overlapping, sticks are feint and their thickness needs to be increased. The region above m/z 120 should be magnified to see clearer patterns in the higher m/z ions which are poorly represented in this figure as is.

Figures 2-4 are grainy and need producing at higher quality for ACP.

Description of the AMS factors should refer to the relevant figures, e.g. diurnal, time series or mass spectra. Descriptions of factor time series and diurnals have various inconsistencies. For HOA, the correlation is not shown in the supplement in Fig. S13 as stated in the manuscript. This should be included. You have stated that the factor peaks between 0600 and 0900 which is not what is shown in Figure 3a. This figure shows elevated concentrations overnight, with the increase between 0600 and 0900 only reflected in the standard deviations, not in the mean. It would be useful to see the diurnal variations split between the seasons as it appears that there is a high degree of variability which means that the average diurnal profile does not show the temporal profile described in the manuscript. I would argue that the morning peak from 0700 to 0900 is also missing in your data, given it is only shown in the standard deviations. The caption of Figure 3 needs to be changed, it is not the "grey area" which represents standard deviations, it is the shaded area. The high contribution of PAHs to the CCOA factor referred to need to be shown more explicitly in the mass spectra by enlarging the MS in the high mass regions in Figure 2a as mentioned previously. The aromatic signatures described for CCOA need to be evidenced, there is no references included or explanation of how they have been identified to be aromatic in origin. The description of OOAs previously classified on page 14 lines 36-39 need to be referenced.

Discussion of EESI factors on page 17 onwards is unclear. The author states that "PMF of the EESI-TOF mass spectral time series was conducted on a 7-factor solution". Surely the PMF was conducted to derive 7 factors? There is no explanation here of how those 7-factors were selected. In addition, there is no detail provided on the influence

and biases introduced from constraining EESI factor time series to the seven non-HOA factors. As highlighted above, factors derived from near-molecular-level techniques typically have more distinct time series than those derived from AMS studies. It should be better justified why this approach to constrain TS was taken and if any effort was made to run the PMF of the EESI factors without constraining the TS to that of the AMS factors.

AMS and EESI factors should be named more clearly. As in Stefenneli and Qi papers, these were named with EESI or AMS subscripts. This would make studying the figures more clear. Numbering of Figures is incorrect after Figure 5. The text here refers to Figure 7 when presumably it should be when the van krevelen plot shows as Figure 4 and according to the sequence should be Figure 6. References to figures should be checked, corrected and improved throughout. Markers on the van krevelen plot are too small and should be changed. It is stated that the sizing of these markers is related to the z-score, however this cannot be interpreted from the figure.

Defining COA on the basis of it having most of the mass at ions with high m/z needs to be better explained. Particularly as ions at high m/z in AMS are attributed to PAHs. Intense ions in the mass spectra shown in Figure 4 should be labelled. CCOA , figure numbers are not included and are blank and should be included. Figure 4 needs improving, the factor time series are unclear and variations cannot be seen in the aspect ratios the figures have been output in. The x-ticks need to be made more regular. The mass spectra y-ticks are overlapping, and in some cases such as CCOA, the mass spectra cannot be clearly seen due to intense ions. The presentation needs to be improved.

Correspondences between the AMS and EESI SOA factors need to be better justified, with comparisons of time series presented. AMS MO-OOA and EESI-TOF MO-OOA being related to one another is not explained and simply assumed in the manuscript. Have the ions listed in paragraph 2 on page 21 been observed elsewhere, these ions formulas should be explained. Various nitro-aromatics in Beijing have been reported

previously. Can you explain why the ions stated on line 46 of page 21 are relevant from aromatics? This should be justified. Page 22 line 18, has this ion been previously observed and if so, where? Bertrand et al reference has no date, this needs to be corrected.

The description of MO-OOA relies on a discussion of haze and non-haze events, which are not depicted on the figures except the final figure (Figure 9) and even in this case they are not clearly shown. Figures need to be improved so the interpretation of factors can be followed. Small acids can also be derived from aromatic oxidation (Zaytsev et al. 2019, Mehra et al. 2020, Wang et al. 2020). Their attribution to aqueous phase chemistry needs to be better explained. Line 44 page 22, "section x" needs to be replaced by a section number. Discussion of AMS and EESI factors is confusing due to the identical naming, these need to be better distinguished in the text.

Atmospheric Implications:

Values are missing on line 15 page 29 of "concentrations between x and y". Page 29 lines 14-48, the discussion of "minor" and "major" haze events is unclear and Figure 9 which shows the values discussed in this section is poorly put together, it is unclear which events each pie chart represents making it impossible to interpret. This needs to be improved. The events discussed in the text have dates, which cannot be established from looking at the figures or the pie charts.

The conclusion that oxidation of aromatics from SFC is responsible for SOA is poorly evidenced (lines 44-48). The evidence for the aqueous factor is stronger, however, certain aspects need improving as discussed above.

Conclusions: Overall, the chemical resolution of the EESI is not discussed in great detail, many of the ions used in the PMF are not shown or discussed in the manuscript. This should be improved.

Supplement: A more comprehensive list of ions should be presented in the supplement. The correlations of AMS nitrate and EESI shown in Figure S1 are poor for the first 2 time periods and should be discussed further.

Labelling of supplement factors needs to be improved. "Splitted" is not correct English and needs changing in all figures.

References: Check referencing style for journal names - in some cases ACP is Atmos Chem Phys and in others it is Atmos. Chem. Phys. This should be consistent with journal guidelines. The same applies to various other abbreviated journal names.

Bryant, D. J., Dixon, W. J., Hopkins, J. R., Dunmore, R. E., Pereira, K. L., Shaw, M., Squires, F. A., Bannan, T. J., Mehra, A., Worrall, S. D., Bacak, A., Coe, H., Percival, C. J., Whalley, L. K., Heard, D. E., Slater, E. J., Ouyang, B., Cui, T., Surratt, J. D., Liu, D., Shi, Z., Harrison, R., Sun, Y., Xu, W., Lewis, A. C., Lee, J. D., Rickard, A. R., and Hamilton, J. F.: Strong anthropogenic control of secondary organic aerosol formation from isoprene in Beijing, Atmos. Chem. Phys., 20, 7531–7552, https://doi.org/10.5194/acp-20-7531-2020, 2020.

Wang, Y., Hu, M., Wang, Y., Zheng, J., Shang, D., Yang, Y., Liu, Y., Li, X., Tang, R., Zhu, W., Du, Z., Wu, Y., Guo, S., Wu, Z., Lou, S., Hallquist, M., and Yu, J. Z.: The formation of nitro-aromatic compounds under high NOxÂăand anthropogenic VOC conditions in urban Beijing, China, Atmos. Chem. Phys., 19, 7649–7665, https://doi.org/10.5194/acp-19-7649-2019, 2019.

Chen, Y., Takeuchi, M., Nah, T., Xu, L., Canagaratna, M. R., Stark, H., Baumann, K., Canonaco, F., Prévôt, A. S. H., Huey, L. G., Weber, R. J., and Ng, N. L.: Chemical characterization of secondary organic aerosol at a rural site in the southeastern US: insights from simultaneous high-resolution time-of-flight aerosol mass spectrometer (HR-ToF-AMS) and FIGAERO chemical ionization mass spectrometer (CIMS) measurements, Atmos. Chem. Phys., 20, 8421–8440, https://doi.org/10.5194/acp-20-8421-2020, 2020.

Massoli, P., Stark, H., Canagaratna, M. R., Krechmer, J. E., Xu, L., Ng, N. L., Mauldin, R. L., Yan, C., Kimmel, J., Misztal, P. K., Jimenez, J. L., Jayne, J. T. and Worsnop, D. R.: Ambient Measurements of Highly Oxidized Gas-Phase Molecules during the Southern Oxidant and Aerosol Study (SOAS) 2013, ACS Earth Sp. Chem., 2, 653–672, doi:10.1021/acsearthspacechem.8b00028, 2018.

Wang, S., Newland, M. J., Deng, W., Rickard, A. R., Hamilton, J. F., Muñoz, A., Ródenas, M., Vázquez, M. M., Wang, L. and Wang, X.: Aromatic Photo-oxidation, A New Source of Atmospheric Acidity, Environ. Sci. Technol., 54(13), 7798–7806, doi:10.1021/acs.est.0c00526, 2020.

Zaytsev, A., Koss, A. R., Breitenlechner, M., Krechmer, J. E., Nihill, K. J., Lim, C. Y., Rowe, J. C., Cox, J. L., Moss, J., Roscioli, J. R., Canagaratna, M. R., Worsnop, D. R., Kroll, J. H. and Keutsch, F. N.: Mechanistic study of the formation of ring-retaining and ring-opening products from the oxidation of aromatic compounds under urban atmospheric conditions, Atmos. Chem. Phys, 19, 15117–15129, doi:10.5194/acp-2019-666, 2019.

Mehra, A., Wang, Y., Krechmer, J. E., Lambe, A., Majluf, F., Morris, M. A., Priestley, M., Bannan, T. J., Bryant, D. J., Pereira, K. L., Hamilton, J. F., Rickard, A. R., Newland, M. J., Stark, H., Croteau, P., Jayne, J. T., Worsnop, D. R., Canagaratna, M. R., Wang, L., and Coe, H.: Evaluation of the chemical composition of gas- and particle-phase products of aromatic oxidation, Atmos. Chem. Phys., 20, 9783–9803,https://doi.org/10.5194/acp-20-9783-2020, 2020.

Yan, C., Nie, W., Äijälä, M., Rissanen, M. P., Canagaratna, M. R., Massoli, P., Junninen, H., Jokinen, T., Sarnela, N., Häme, S., Schobesberger, S., Canonaco, F., Prevot, A. S. H., Petäjä, T., Kulmala, M., Sipilä, M., Worsnop, D. R. and Ehn, M.: Source characterization of Highly Oxidized Multifunctional Compounds in a Boreal Forest Environment using Positive Matrix Factorization, Atmos. Chem. Phys., 16, 12715–12731, doi:10.5194/acp-16-12715-2016, 2016.

Stefenelli, G., Pospisilova, V., Lopez-Hilfiker, F. D., Daellenbach, K. R., Hüglin, C., Tong, Y., Baltensperger, U., Prévôt, A. S. H. and Slowik, J. G.: Organic aerosol source apportionment in Zurich using an extractive electrospray ionization time-of-flight mass spectrometer (EESI-TOF-MS)-Part 1: Biogenic influences and day-night chemistry in summer, Atmos. Chem. Phys, 19, 14825–14848, doi:10.5194/acp-19-14825-2019, 2019.

Qi, L., Chen, M., Stefenelli, G., Pospisilova, V., Tong, Y., Bertrand, A., Hueglin, C., Ge, X., Baltensperger, U., Prévôt, A. S. H. and Slowik, J. G.: Organic aerosol source apportionment in Zurich using an extractive electrospray ionization time-of-flight mass spectrometry (EESI-TOF): Part II, biomass burning influences in winter, Atmos. Chem. Phys., 19, 8037–8062, doi:10.5194/acp-19-8037-2019, 2019.

---

## Referee Comment (RC2) · Anonymous Referee #2 · 7 Nov 2020

This manuscript describes aerosol chemistry measurements that were acquired during a two and a half month period in Beijing in 2017. Aerosol chemistry was obtained using a PM25 inlet Long Time of Flight Mass Spectrometer (LTOFAMS) and an Extractive Electrospray Ionization-Time of Tlight Mass Spectrometer (EESI-ToF-MS). Both of these instruments provide high mass resolution measurements of aerosol chemistry and are complementary in that the EESI-TOF results in low fragmentation, allowing the detection of large molecules making it easier to determine the source and identity of different organic species.

The study presents a combined positive matrix approach that is used to optimize the factor solutions obtained from the combined instrumentation. It equally introduces the use of z-score analysis on the mass spectral profiles of the different PMF factor solutions to determine which m/z contributed to the identification of each factor. Using this combined analysis the authors present 4 different types of primary organic aerosol and 4 types of secondary organic aerosol. The paper is relatively well written and organized. Figures are well labelled and clear, but in some cases need better referencing in the text. I recommend this manuscript for publication after some minor changes. My comments and suggestions are below.

Sample set up:

Page 4, Line 14: Were the aerosols dried prior to sampling? What was the average sample aerosol RH throughout the study? Were particle losses calculated for the PM1 to 2.5 range?

Can the authors add the size range typically measured by the EESITOF to the discussion of particle size sampling efficiency on page 5/6.

How often was the denuder regenerated during the sampling campaign?

What is the make and model of the SMPS used in this study? What is the size range measured by the SMPS. Can the authors show the aerosol size distribution measured by the SMPS and compare with that of the L-TOF (shown in Fig S2).

How did this size range change during the different sampling events "haze events" and the "clean periods" (Heating non-heating)?

What is the difference in mass between the PM25 inlet and the SMPS. How representative of the total PM2.5 mass is the PM25 LTOFAMS measurements? Are auxiliary measurements of total PM available for comparison?

Given that the smaller, locally formed aerosol particles are not efficiently sampled by the PM25 inlet, what impact does this have on the interpretation of the measurements?

Is a standard or capture vaporizer used in conjunction with the PM25 inlet? If a standard vaporizer is used, what is the particle collection efficiency estimated to be, and is the calculated CE dependent on particle diameters (to account for the enhanced effects of particle bounce for larger particles diameters)?

High concentrations were measured during this field campaign, often resulting in the clogging of the EESI-TOF instrument. Was there any evidence to suggest that there was overloading of the LTOFAMS instrument?

The authors show several PMF solution for the LTOFAMS analysis but only show the final solution for the EESI-TOF. Can the authors state if the PMF analysis on the "unconstrained" EESI TOF was performed?, and if so which factors dominated the unconstrained PMF solution?

Aqueous phase SOA: The PMF analysis allowed the extraction of a more oxidized aqueous phase aerosol. In Fig. 3a the MOOA_AQ is shown to have highest concentrations (reaching 20 micrograms) during the haze event on the 4th and 7th, during high NO3 contributions (and high RH and low wind seed). The characteristic enhancement of the CO2 is illustrated in Fig. 7a during this period. However, there are several other periods during the field campaign when this MO-OOAaq species is identified (at concentrations near to 5 micrograms) (under the same conditions of RH and high NO3 Fraction) but the enhancement of the CO2+ signal was not observed.

Is this CO2+ enhancement really associated with these species or is it somehow an artefact during high NO3 concentrations?.

How did the factor mass spectral profile compare with reference mass spectra (oxalic acid, malonic acid and succinic acid (Canagaratna et al., 2015))?

The authors state that the m/z 44 artefact is very low in this instrument (4%), however, could this CO2+ enhancement be somehow related to artefacts linked to mixtures of inorganic and organic species?

Previous studies have shown how aerosol liquid water can promote the formation of water-soluble organic nitrogen (Yu Xu et al., 2020 Environ. Sci. Tech. https://pubs.acs.org/doi/pdf/10.1021/acs.est.9b05849).

What is the role of nitrogen in the formation of these aqueous organic species? Is there evidence of organonitrate species? Has aerosol acidity being evaluated during these measurements? During the intense haze episode, this MOOA species was measured continuously over a period of 3 days. In one of the cited articles (Kuang et al., 2020) it is mentioned that most of the aqSOA was formed during daytime periods with high photochemical activity and that dark aqSOA only contributed negligibly to the total OOA concentrations.

In this work, the increase in aqSOA remains constant over three days with little diurnal variation. Do you consider that this aqSOA is locally formed or influenced by regional processes? Does the aerosol size distribution provide information to determine this?

A non source-specific factor LO-OOA ns ?

Recently it has been shown that the PM25 inlet AMS systems may be capable of measuring airborne bacteria (Wolf et al., 2017 Atmospheric environment, (https://doi.org/10.1016/j.atmosenv.2017.04.001).). In this paper, there are some characteristics of the LO-OOAns species (O:C, diurnal pattern, higher concentrations during warm period than colder periods) as the resolved bacteria-like factor in Wolf et al.,2017.

Is it possible to provide more information on the average diameter as a function of time for each of the resolved factors to help provide more information on their source and atmospheric processing prior to being sampled? At least provide the SMPS size distributions which would help illustrate regionally influenced factors and those from local processes.

Were BC measurements available for correlation with the HOA and BBOA.

Minor remarks:

When discussing the diurnal variations better referencing to the figure is necessary.

Although the information of O/C, H/C are included in and Fig. 5 and 6, it would be useful for comparison to other studies, to have these average values as well as the N/C ratios illustrated on the factor profiles in Fig.3 and 4.

Page 13, Line 11: Were external time series available for comparison, other than CO? Can you provide the value for the "good" correlation.

Page 22, Line 44 (please include correct section no.)

Page 23, Line 7 r2 =of 0.93), remove = or of

Page 21, Line 25: Bertrand et al.. please include the full reference.

Page 5, Line10 What is this diagnostic species?

Is the custom peak fitting algorithm something that could be applied to lower resolution instruments in the future?

---

## Author Comment (AC1) · 14 Mar 2021

Dear Dr. Allan,

Please find enclosed our response to the reviewers for manuscript acp-2020-835 ("Quantification of solid fuel combustion and aqueous chemistry contributions to secondary organic aerosol during wintertime haze events in Beijing").

This file contains a point-by-point response to reviewer #1, with changes to the text
marked in blue. Note that one reference (in this reply to reviewer #1 and also in the manuscript) from Lamkaddam et al. (2021) is going to be published on 24 March, we highlighted this in the reference list, and we will update this once the DOI is available.

Thank you for considering our manuscript for ACP, and we look forward to your response.

Best regards,

Yandong TONG

Please also note the supplement to this comment:
https://acp.copernicus.org/preprints/acp-2020-835/acp-2020-835-AC1-supplement.pdf

**Supplement:**

**Response to RC1**

We thank the reviewer for the helpful comments. Below we provide a detailed point-by-point response to the issues raised by the reviewer. Reviewer comments provided in *italics* and our responses follow in normal text. Changes to the manuscript are denoted in blue font. When our responses reference other comments, we use the formalism R#C#, such that R1C5 would refer to Comment 5 by Reviewer 1.

**Comment #1 --- General Comments:**

*PMF of the EESI-TOF-MS has been constrained by the time series derived from PMF of AMS measurements. This has the potential to introduce significant biases in the results, which should be addressed in the manuscript. EESI-TOF-MS detects a similar range of compounds to those detected by FIGAERO-CIMS and previous studies have shown distinct differences in the time series of CIMS factors compared with those derived from AMS (Chen et al. 2020). The additional chemical composition provided by the EESI-TOF-MS should enable more factors to be derived on the basis of chemistry occurring across the measurement period which the AMS does not have the chemical resolution to identify. Therefore the approach taken to constrain the time series of EESI-TOF-MS PMF factors could have influenced the identification of factors. The PMF of the AMS factors forms the basis of the results presented in the manuscript, and the use if EESI-TOF-MS data is supplementary and used simply to show the composition of the EESI which corresponds to the AMS factors. This is different to a true PMF analysis of the chemically insightful EESI-TOF-MS data and should be discussed accordingly.*

*What are the correlations between the EESI-TOF-MS and AMS factor time series? How sensitive are the time series to the constrains. I understand that carrying out a separate PMF of the EESI-TOF-MS without constraining the time series is a large body of work, however, you should try to discuss how not doing this could have influenced your results and interpretation.*

**Response:**

The reviewer is correct that we have not presented a standalone PMF analysis of the EESI-TOF data. Instead, we use the PMF model to identify EESI-TOF chemical signatures corresponding to temporal trends of factors derived from AMS PMF analysis. We do not argue that this would be the optimal utilisation of co-located AMS and EESI-TOF data, if it were the case that both instruments were performing optimally. However, for the specific situation encountered in this study, where (1) interpretation of the standalone EESI-TOF data is significantly complicated by denuder breakthrough; (2) high EESI-TOF backgrounds may increase the uncertainty of peak fitting; and (3) AMS PMF resolves multiple factors that are temporally distinct but difficult to interpret chemically, we believe the approach taken here maximises the explanatory power of this specific dataset.

We did attempt a preliminary PMF of the standalone EESI-TOF dataset, but found the results to be uninterpretable. This is likely because the PMF model requires detector linearity and static factor profiles. Denuder breakthrough compromises both assumptions, because the volatile and semivolatile contributions to factor profiles depends on the time-dependent state of the denuder. Such effects have recently been characterised in detail by Brown et al. (2021). As discussed in sections 2.2.1 and Text S3, we have tried to reduce these effects by removing ions likely to be dominated by gas phase/denuder breakthrough, but this method does not allow quantitative correction of signals from ions having contributions from both phases. However, by constraining the EESI-TOF PMF solution with AMS factor profiles, the solution becomes weighted towards explaining temporal trends corresponding to the particle phase. Further, by utilising the EESI-TOF for qualitative (factor identification) rather than quantitative (factor resolution) purposes, we minimise the effects of artifacts introduced by gaseous signals.

We now address these issues in section 2.3 as follows (page 7 line 1):

"Note that this strategy would not necessarily be the optimal use of co-located AMS and EESI-TOF data, if both instruments were performing optimally. In particular, it neglects to take advantage of the higher chemical resolution of the EESI-TOF for factor separation. However, for the specific situation encountered in this study, where (1) interpretation of the standalone EESI-TOF data is significantly complicated by denuder breakthrough; (2) high EESI-TOF backgrounds may increase the uncertainty of peak fitting; and (3) AMS PMF resolves multiple factors that are temporally distinct but difficult to interpret chemically, we believe the selected approach maximises the explanatory power of the dataset. As an alternative strategy, a preliminary PMF of standalone EESI-TOF data was attempted, however this did not yield interpretable results. This is likely because the PMF model (discussed in the next section) requires detector linearity and static factor composition. Denuder breakthrough compromises both assumptions, because the volatile and semivolatile contributions to EESI-TOF signal depend on the time-dependent state of the denuder (Brown et al., 2021). The EESI-TOF data processing protocols utilised above reduce but do not eliminate this issue. However, by constraining the EESI-TOF PMF solution with AMS factor profiles, the solution becomes weighted towards explaining temporal trends observed in the particle phase. Further, by utilising the EESI-TOF for qualitative (factor identification) rather than quantitative (factor resolution) purposes, the impact of artifacts introduced by gaseous signals is minimised."

The reviewer also asks about correlations between the EESI-TOF and AMS factor time series, and the sensitivity of the EESI-TOF time series to the constraints. Here the comparison of time series from AMS and EESI-TOF factors are shown below (added to the manuscript as Fig. S17) and the scatter plots of each factor from EESI-TOF vs AMS are shown in R1C2 (added to the manuscript as Fig. S18). Clearly, apart from the time series of MO-OOAaq, which shows discrepancy during the haze strongly influenced by the aqueous-phase chemistry, other factors show great correlation between AMS and EESI-TOF time series.

[Figure]

**Comment #2 --- General Comments:**

*The discussion of ions identified by EESI-TOF-MS is limited and needs to be referenced more extensively. Mass spectra from EESI-TOF-MS should be labelled and a more detailed list of ions provided in the supplement along with their potential sources. Correspondences drawn between AMS and EESI-TOF-MS ions should be better justified through a more thorough evaluation of the ions contributing to each factor. Time series of dominant ions contributing to each factor should be compared with that of the factor profile itself and its corresponding AMS factor in a Figure to show the validity of constraining the EESI-TOF-MS factor profiles by the AMS time series. CIMS and AMS factor time series should also be presented on the same axis for comparison.*

**Response:**

In a complex environment such as Beijing, it is frequently the case that an ion molecular formula alone is not sufficient to establish a clear link to a specific source. There are exceptions to this, which are discussed in the paper. For example, levoglucosan is used as a tracer for biomass burning activities, small acids and diacids are related to aqueous phase chemistry, and fatty acids are related to cooking activities. However, the chemical resolution of the EESI-TOF is also essential for assessing patterns in the factor profiles, such as the differences in carbon number and H:C ratio shown in Figs. 5 and 6. These systematic features are crucial to the interpretation of the results and, we believe, provide explanatory power that is at least equivalent to inspection of individual ions.

The treatment of individual ions has been improved through more systematic referencing, labeling of major ions in the EESI-TOF factor profiles, and inclusion of a complete ion list in the supplement. However, in general we feel that the current balance between pattern-oriented analysis and individual ion inspection is appropriate given their relative information content, and that especially in the absence of structural information, a more comprehensive discussion the set of individual ions will rather obscure the key features of the dataset.

As suggested by the reviewer, key ions in the EESI-TOF factor mass spectra are now labelled, and full peak lists for EESI-TOF and AMS are given in the supplement.

Correspondence between AMS and EESI-TOF factors is given, because the EESI-TOF factors are constrained to AMS. The validity of such constraints are supported by the observation that, for the primary factors where the AMS-only PMF is able to establish a clear identity, the retrieved EESI-TOF chemical composition is consistent with the AMS factor definitions. However, because some differences between the AMS and EESI-TOF factor time series are permitted due to the combination of non-zero $a$-values and bootstrapping, we have added a direct comparison of the AMS and EESI-TOF factor time series to the supplement. This includes a series of plots in which the time series from Figs. 2a and 4a are overlaid on a factor-by-factor basis, as shown in R1C1 and in Fig S17, and also EESI-TOF vs. AMS scatterplots, as shown below and in Fig. S18 (note: the linear fitting uses trust-region Levenberg-Marquardt least orthogonal distance method, because there are uncertainties in the independent variables, namely here factors from AMS and EESI-TOF). From these plots, COA, CCOA, and MO-OOA$_{SFC}$, have better fit, with the $\chi^2$ of linear fit lower than 1000, whereas BBOA, MO-OOA$_{aq}$, LO-OOA$_{SFC}$ and LO-OOA$_{ns}$ are more scattered, with $\chi^2$ higher than 1400, suggesting the method of constraining AMS time series in PMF on EESI-TOF data still has potential to improve. The information about the comparison is added to the text (page 12 line 12):

The EESI-TOF factor time series are compared to their AMS counterparts in Fig. S17, and scatter plots of EESI-TOF vs AMS on a factor-by-factor basis are shown in Fig. S18. These comparisons suggest that allowing some discrepancies, the EESI-TOF factor time series captures the most of the temporal evolution from the AMS factor time series.

[Figure]

[Figure]

[Figure]

[Figure]

[Figure]

[Figure]

[Figure]

**Comment #3 --- General Comments:**

*Presentation, discussion and interpretation of factors is confusing and hard to follow, particularly with respect to comparison between AMS and EESI-TOF-MS factors due to their identical naming. Additionally, many of the figures need to be improved and should be referenced better throughout the manuscript. It would be useful if the haze events of interest and heating/non-heating periods were labelled in each figure. Figure 9, in particular, needs to be overhauled in favour of something much more easy to interpret. Perhaps stacked bars could be used to compare the contributions of different factors during different haze events. Interpretation of figures also requires some attention, with diurnals interpreted based on the time series' of their standard deviations as opposed to their average values. As presented, the figures hinder the interpretation of the manuscript and should be improved to emphasise conclusions.*

**Response:**

We have improved the figures in several ways. In particular, all figures are now high resolution, and periods of severe haze, light haze, and clean air are shaded on every time series. In Fig. 9 (now correctly labelled as Fig. 12), arrows now link pie charts to their respective haze periods, and in Fig. 4, key ions are labelled in the presentation of EESI-TOF factors. Other changes are discussed below in response to specific comments.

Figure referencing has been made more systematic, especially in the discussion of PMF factor identity.

The statement by the reviewer regarding diurnals is incorrect. As stated in the caption, the shaded areas represent the 1$^{st}$ and 3$^{rd}$ quartiles, not standard deviation. Therefore, the original interpretation of this peak is justified, as it reflects a systematically recurring feature of the factor when concentrations are high. This issue is discussed further in response to R1C34.

**Comment #4 --- General Comments:**

*Structure and language throughout the manuscript need to be improved. Placeholders for data values and figure numbers remain throughout. In addition, there are figures referenced which are not included, and those which are included are not referenced in the text thoroughly making the paper challenging to follow. Several sections including the introduction and methods and not well-structured and clear. The introduction needs to be streamlined and made clearer and referencing needs to be improved throughout, particularly in the results section where it would be useful to discuss where ions identified by EESI-TOF-MS have been previously observed.*

**Response:**

The structure of the Introduction is discussed in response to C1R5. Specific modifications are discussed in response to C1R5 through C1R11. Likewise, modifications to the Methods are addressed in C1R12 through C1R26.

Placeholders are corrected throughout the paper.

As discussed in response to the previous comment, figures are now referenced more systematically throughout the manuscript. We have also corrected some numbering errors and omissions that arose during PDF conversion.

**Comment #5 --- Specific Comments, Introduction**

*The introduction is currently very hard to follow and lacks a coherent structure. The author begins by introducing organic aerosol in the first paragraph in which its key sources are outlined, and then in paragraph*

*2 they mix health effects with sources and instrumentation. This structure should be streamlined into a narrative which provides the information required to contextualise the results presented in the manuscript.*

**Response:**

The first paragraph of the introduction provides a general discussion of OA, highlighting the significant contribution of OA to total PM, the adverse health effects of the POA and SOA fractions, and the difficulty in linking SOA to individual sources (in contrast to success for POA). The second paragraph summarises health-relevant air quality/health issues in Beijing (as an example of a megacity where such issues are of particular concern), and summarises the state of knowledge for POA and SOA source apportionment at this site. As such efforts are dominated by AMS/ACSM measurements, it is important to note that the prevailing description of SOA is in terms of non-source-specific OOAs. Such a description is dictated by the nature of the measuring instruments, leading naturally into paragraph 3, where we address the advantages and disadvantages of existing measurement systems (including but not limited to the AMS). This motivates our introduction of the EESI-TOF and the potential of joint AMS/EESI-TOF measurements to provide new insight into SOA sources. The final paragraph is a brief overview of the scope of the study.

We feel that this is a coherent narrative that provides the needed motivation and context for the results presented herein. As a result, we have not performed a major restructuring of the introduction, but have instead attempted to highlight and clarify the existing structure.

**Comment #6--- Specific Comments, Introduction**

*Lines 8-12 on page 2 are not referenced and mention uncertainties in quantification due to instrumentation, however, instrumentation is not introduced until line 48 of the same page. The few sentences about health impacts of ROS seem out-of-place in this paragraph.*

**Response:**

Page 2 lines 8-12 are part of a larger discussion on SOA sources, and refer to instrument limitations specifically in the context of their effects on the "quantification of SOA sources and/or formation pathways." This is relevant to the discussion, and the text is retained. We now reference several reviews highlighting this issue (Hallquist et al., 2009;Fuzzi et al., 2006)

The health risks of OA are a key factor motivating the study of air quality in polluted megacities such as Beijing. ROS is crucial to understanding the health risks of SOA, and thus crucial to the motivation of the paper. Further, it has recently become apparent that SOA from different sources carries different ROS content (and likely different health risks). This information has been added to the paper to better link the health issues/ROS content to SOA source apportionment (page 2 line 3):

"Recent studies indicate that SOA from different sources carries different ROS content (and thus likely different health risks), highlighting the importance of OA source identification and quantification (Zhou et al., 2018; Daellenbach et al., 2020)."

**Comment #7 --- Specific Comments, Introduction**

*Results from the AMS are mentioned in paragraph 2 on page 2, while several sentences describing it are on page 3 (lines 9 -16). I would suggest that the author introduces instrumentation first before talking about observations with those instruments in Beijing. The introduction should be re-written to better contextualise the more detailed aerosol source characterisation which the EESI provides, this is currently disjointed, with the key*

*AMS factors the EESI is expected to better constrain being described on lined 25 - 42 of page 2, while the EESI itself is not described until page 3.*

**Response:**

In the context of the overall narrative (see response to R1C5), the key features of the source apportionment results in paragraph 2 is that they reflect the state-of-the-art, and that the state-of-the-art has some important limitations of direct relevance to public health (i.e., treatment of SOA sources). As these limitations are directly evident from the factor nomenclature and behavior, we find it distracting to engage in a detailed discussion of instrumentation before motivating such a discussion. That is, we believe it is necessary to establish that a problem exists (i.e., lack of clarity in SOA source apportionment results) before discussing how to solve the problem (i.e., instrument capabilities). We have made several modifications to the paper to more clearly present this narrative.

The discussion of Beijing is more clearly linked to the health and source apportionment issues raised in the preceding paragraph (page 2 line 12):

"Fine aerosol pollution is a major public health concern in many megacities, highlighting the need for efficient mitigation strategies informed by a detailed assessment of POA and SOA sources. Beijing is an area of particular interest, due to the frequency of extreme haze events in northern China (An et al., 2019) and a rapidly changing pollution landscape in response to the "Atmospheric Pollution Prevention and Control Action Plan" implemented in 2013 by the Chinese government. This initiative targeted selected anthropogenic emissions sources, reducing annual mean $PM_{2.5}$ concentration by ~30% between 2013 and 2017, although remaining much higher than both national air quality standards and WHO guidelines."

The AMS data is now introduced clearly in the context of its being the main online source apportionment tool in existing Beijing studies (page 2 line 18):

"As a result, numerous studies have investigated the composition and sources of $PM_{2.5}$ in Beijing (Duan et al., 2020; Duan et al., 2019; Xu et al., 2019; Zhao et al., 2019; Äijälä et al., 2017; Elser et al., 2016; Hu et al., 2016; Sun et al., 2016a; Huang et al., 2014; Zhang et al., 2014; Sun et al., 2013), with most online source apportionment studies utilizing aerosol mass spectrometers (AMS). These studies have successfully identified POA sources, with dominant winter sources including coal combustion (10 to 30%), biomass burning (9 to 18%), traffic (9 to 18%), and cooking (12 to 20%). In contrast, although SOA typically comprises 35 to 70% of Beijing OA, far less is known about its sources and formation processes."

Finally, we have reorganised paragraph 3 so that it begins with the AMS, thereby establishing the instrument characteristics leading to the limitations in SOA source apportionment that conclude paragraph 2. Other techniques are then presented and evaluated as alternatives (page 2 line 35):

Limitations in SOA source apportionment are directly tied to limitations of the measuring instruments. For the Aerodyne aerosol mass spectrometer (AMS), a trade-off exists between quantification and time resolution vs. chemical resolution. Quantification and time resolution are facilitated by high temperature vaporisation. which induces significant thermal decomposition and ionisation-induced fragmentation (DeCarlo et al., 2006). This

decreases chemical resolution, particularly for the multifunctional and highly oxygenated molecules of which SOA is comprised (e.g. multifunctional acids, peroxides, organonitrates, organosulphates, oligomers), thereby hindering SOA source apportionment. To avoid thermal decomposition, the CHARON PTR-MS uses a lower temperature vaporisation scheme, but the proton transfer reaction ionisation scheme is sufficiently energetic to cause extensive fragmentation of typical SOA molecules (Muller et al., 2017; Eichler et al., 2015). To reduce ionisation-induced fragmentation, several semi-continuous measurement techniques have also been developed, e.g. Thermal Desorption Aerosol GC/MS-FID (TAG) by Williams et al. (2006), and Filter Inlet for Gases and AEROsols chemical ionisation time-of-flight mass spectrometer (FIGAERO-CIMS) by Lopez-Hilfiker et al. (2014). Although these instruments have lower thermal decomposition and better chemical resolution, like offline filter sampling they are subject to reaction/vaporisation processes on the collection substrate and decreased time resolution. Alternatively, offline filter analysis has some advantages, including 1) possibility to apply a wide variety of analytical techniques, which can maximise the chemical information retrieved for the analysed fraction; and 2) low cost and maintenance requirements for filter sampling, which in turn facilitates 3) practicality of wide spatial and temporal coverage. However, it also has some drawbacks, including 1) low time resolution incapable of capturing characteristic timescales of certain OA sources and/or ageing and formation processes, 2) artefacts due to adsorption, evaporation, and chemical reactions during sample collection, storage, and/or transfer, (Ge et al., 2012; Huang et al., 2010; Hildebrandt et al., 2010; Hallquist et al., 2009), and 3) the analysable OA faction may vary significantly between different techniques.

To better investigate SOA sources and/or formation processes, an instrument that can resolve aerosol chemical composition was recently developed at the Paul Scherrer Institute (PSI). The extractive electrospray ionisation time-of-flight mass spectrometer (EESI-TOF) utilises a soft ionisation technique with minimal thermal energy transfer to the analyte molecules. This yields online, near-molecular-level measurement (i.e., molecular formulae) of organic aerosol composition with high time resolution (seconds) without thermal decomposition or ionisation-induced fragmentation (Lopez-Hilfiker et al., 2019). Operating principles are discussed in detail in section 2.2.1. Two recent source apportionment studies in Zurich using an EESI-TOF, together with an AMS, successfully resolved several SOA factors and quantified the processes governing SOA concentrations for summer and winter (Qi et al., 2019; Stefenelli et al., 2019). These studies confirm that EESI-TOF and AMS are highly complementary, with the AMS providing robust quantification but limited chemical resolution, and the EESI-TOF providing a linear but hard-to-quantify response with high chemical resolution. The combined measurements, therefore, have the potential to provide quantitative, real-time measurement of organic aerosol composition with high chemical resolution.

**Comment #8--- Specific Comments, Introduction**

*Consider splitting paragraph 2 on page 2 up with the discussion of Beijing separate from the broad discussion of AMS and ACSM capabilities. This should be made clearer.*

*Lines 32 - 33 on page 2 are not justified, there are a variety of previous studies in Beijing which study and apportion OA to different sources and formation mechanisms, inc. Bryant et al. (2020) and Wang et al. (2019).*

**Response:**

The overall structure of the Beijing discussion was addressed in detail in response to the previous comment (R1C7) and so is not repeated here. We have incorporated the suggested studies, and several others, into the discussion of the current state of knowledge of SOA sources and processes. These studies provide valuable insight into sources and processes affecting SOA in Beijing, but do not achieve SOA source apportionment. The revised text follows (page 2 line 24):

"In contrast, although SOA typically comprises 35 to 70% of Beijing OA, far less is known about its sources and formation processes. Bryant et al. (2020) found the isoprene-derived SOAs are strongly controlled by anthropogenic $NO_x$ and sulphate aerosols in summer via offline-filter analysis. Wang et al. (2019) discussed the factors that influence the formation of secondary nitro-aromatic compounds under high $NO_x$ concentration and aromatic precursors. Modeling studies also established links between atmospheric oxidising capacity and SOA formation (Feng et al., 2019), and suggested the role of heterogeneous HONO and primary residential emission in SOA formation in winter (Xing et al., 2019). However, apportionment of SOA to specific sources has not yet been achieved, with online source apportionment studies (using AMS) reporting either a single SOA factor (denoted oxygenated organic aerosol, OOA), or two factors distinguished by the extent of oxygenation (less oxygenated OOA, LO-OOA, and more oxygenated OOA, MO-OOA) (Xu et al., 2019; Elser et al., 2016; Sun et al., 2016a; Sun et al., 2013)."

As suggested, we end the paragraph here, and then proceed with a discussion of AMS-specific treatment of SOA factors.

**Comment #9--- Specific Comments, Introduction**

*The introduction of EESI on page 3 should explain better how it is able to capture detail which the AMS cannot, and which components of aerosol measured by the AMS are most poorly understood. Line 25 on page 3, include references.*

**Response:**

The components that are not well-measured by the AMS were identified as the "multifunctional and highly oxygenated molecules of which SOA is comprised", during the AMS instrument discussion (page 2 line 39). We feel this is the appropriate place for this information, as it is a consequence of the particle vaporisation and ionisation scheme.

We clarify that the ability of the EESI-TOF to provide improved chemical resolution are due to the combination of a soft ionisation technique and minimal transfer of thermal energy to the analyte molecules. Further details are reserved for the Methods section. The revised text reads as follows (page 3 line 1):

"To better investigate SOA sources and/or formation processes, an instrument that can resolve aerosol chemical composition was recently developed at the Paul Scherrer Institute (PSI). The extractive electrospray ionisation time-of-flight mass spectrometer (EESI-TOF) utilises a soft ionisation technique with minimal thermal energy transfer to the analyte molecules. This yields online, near-molecular-level measurement (i.e., molecular formulae) of organic aerosol composition with high time resolution (seconds) without thermal decomposition or ionisation-induced fragmentation (Lopez-Hilfiker et al., 2019). Operating principles are discussed in detail in section 2.2.1."

**Comment #10--- Specific Comments, Introduction**

*Discussion of instruments on page 3 lacks accurate chronology, e.g. CHARON, FIGAERO and GC/MS-FID.*

**Response:**

The distinction between continuous and semi-continuous measurement techniques is key to the discussion of instrument capabilities and thus guides the structure of this paragraph. The CHARON is a continuous measurement technique, and thus appears immediately following the AMS (which is likewise continuous). The TAG and FIGAERO are semi-continuous systems, and so are discussed separately. We have switched the order of TAG and FIGAERO in the single sentence naming these instruments so that semi-continuous techniques now appear in chronological order (see page 2 line 41).

**Comment #11--- Specific Comments, Introduction**

*Page 3 lines 35 - 38 are contradictory, you first say that the EESI-TOF provides a hard-to-quantify response, then go on to say it has the potential to provide a quantitative measurement of OA.*

**Response:**

This comment does not accurately describe the original text. We state that the EESI-TOF provides a "linear but hard-to-quantify response", but that the AMS/EESI-TOF combination (not EESI-TOF alone) has the potential to provide quantitative OA measurements with high chemical resolution. We consider this an accurate statement, although the optimal method of combining AMS and EESI-TOF data is not yet clear.

The original text is provided for reference and left unchanged (except for correction of a minor typo denoted in blue, page 3 line 9):

"These studies confirm that EESI-TOF and AMS are highly complementary, with the AMS providing robust quantification but limited chemical resolution, and the EESI-TOF providing a linear but hard-to-quantify response with high chemical resolution. The combined measurements, therefore, have the potential to provide quantitative, real-time measurement of organic aerosol composition with high chemical resolution."

**Comment #12--- Specific Comments, Methodology**

*Again the structure of this needs to be corrected, you discuss the exact co-ordinates of the measurement location and then go on to talk about Beijing in the context of the NCP. This needs to be more focused. A better structure would start with Beijing, move to the location of the site and then talk about its surroundings and potential influences*

**Response:**

The text is revised according to this suggestion (page 3 line 21):

"Beijing is the capital city of China PR and one of the most populated cities in the world, with more than 20 million inhabitants. It is located at the northwestern end of the North China Plain and surrounded by the Yan Mountains from the southwest to the northeast. Measurements were conducted at the National Centre for Nanoscience and Technology in Beijing (40.00º N, 116.38º E) and the measurement site is located on the roof of the South Building of the National Centre for Nanoscience and Technology (~20 m above ground level) surrounded by smaller buildings, with the exception of an 18-floor building approximately 30 m to the north,

which may interfere with and even block the wind from this direction. The northern part of the fourth ring highway is situated about 200 m south of the site. However, buildings between the highway and the site reduce the influence from local highway traffic. This location is not affected by major emissions from industries."

**Comment #13--- Specific Comments, Methodology**

*Page 4 lines 6 - 8, you suggest that highway traffic and industry do not influence measurements at the site. However, given that pollution in Beijing is regional, and it is well known that industrial air masses from the SE affect the air quality in Beijing, this statement needs to be removed or better justified. Surely the site is influenced by both local emissions and regional transport?*

*Page 4 lines 9-10, can you explain why this period from October - December was chosen when measurements were carried out from September. No details are provided on sampling flow rates, these need to be added.*

**Response:**

The statement in the original manuscript specifically referred to the influence of the local fourth ring highway, rather than highway traffic generally. The statement regarding industry was ambiguous and has been clarified. The revised text reads as follows (page 3 line 27):

"The northern part of the fourth ring highway is situated about 200 m south of the site. However, buildings between the highway and the site reduce the influence from local highway traffic. This location is not affected by major emissions from industries."

The campaign lasted from late September to mid-December, 2017. However, operational problems with the AMS and/or EESI-TOF from late September to mid October prevent use of that data. This is clarified in the text as follows (page 3 line 40):

"Here we focus on OA measurements from late October to mid-December 2017, during which period both the AMS and EESI-TOF were mostly operational."

The flow rate through the cyclone is 5 L min$^{-1}$, and the AMS and EESI-TOF flowrates are ~0.1 L min$^{-1}$ and ~ 0.8 L min$^{-1}$, respectively. These numbers are added to the corresponding parts, namely section 2.1 for flowrate through cyclone, section 2.2.2 for AMS flowrate and section 2.2.1 for EESI-TOF flowrate.

**Comment #14--- Specific Comments, Methodology**

*Lines 7-10 page 5 - could the sensitivity differences be related to changes in relative humidity?*

**Response:**

The instrument sensitivity was observed to undergo a step change following physical cleaning/realignment of the capillary. This is not consistent with the timescale of ambient meteorological changes, and we therefore eliminate ambient RH variation as a possible cause.

**Comment #16--- Specific Comments, Methodology**

*Lines 9 -12 page 5 - here you state a " a diagnostic species" is used, and below this you state "inorganic nitrate species" - these need to be clarified in more detail.*

**Response:**

This question was also raised by Reviewer 2 (R2C23) and we repeat the response here. The sentence in question was meant to introduce the need to identify a diagnostic species, the identity (AMS $NO_3^-$) and use of which comprise the rest of the paragraph. To clarify this, the revised sentence reads (page 3 line 11 in supplement, according to R1C19, we put this part in the supplement):

"Therefore, we select a diagnostic species that can be measured with higher time resolution to monitor the sensitivity throughout the campaign."

The sentence reference "inorganic nitrate species" has been revised to clarify the precise quantities measured (page 3 line 12 in supplement, according to R1C19, we put this part in the supplement):

"Intercomparison of inorganic nitrate species (AMS $NO_3^-$ and EESI-TOF $[NaNO_3]Na^+$) yield strong correlations during periods in which the instrument operation was stable (i.e., not affected by major clogging or cleaning/realignment of the electrospray capillary)."

**Comment #17--- Specific Comments, Methodology**

*Lines 29 - 30 page 5 - provide more discussion of the assumption that the time dependant changes in sensitivity are eliminated for organics.*

**Response:**

The phrase "we assume" seems to have caused some confusion. This was intended only as a callback to discussion earlier in the paragraph, where we noted that the normalization to nitrate corrected for effects that "uniformly affect all measured ions (i.e., without compound-dependent effects)". The sentence has been revised, and now reads (page 3 line 28 in supplement, according to R1C19, we put this part in the supplement):

"This correction accounts for time-dependent changes in sensitivity that uniformly affect all measured ions."

**Comment #18--- Specific Comments, Methodology**

*Lines 30 - 33 page 6 - here you state that particle-phase signals tend to be less oxygenated than the background ions, while in supplement section S1 you state the opposite. This needs to be corrected.*

**Response:**

We appreciate the correction. The particle signals indeed are typically more oxygenated than the background, and the identified text is corrected accordingly.

**Comment #19--- Specific Comments, Methodology**

*Some of the details provided in this section may not be of interest to the typical reader and I would suggest moving more of the detailed instrument trouble shooting to the supplement as currently the paper is harder to follow as a result. The sections in the supplement are much clearer are will be easier to access for a technical reader who wants to understand the troubleshooting process. This level of detail in the main body of the manuscript itself shifts the focus from the results.*

**Response:**

[revised manuscript text omitted]

We remove the text about normalisation of time-dependent EESI-TOF sensitivity into SI, see Text S4:

**Text S4: Normalisation of time-dependent EESI-TOF sensitivity**

The EESI-TOF achieved ~ 90 % data coverage during the sampling period, with the 10 % missing data including solution changes, signal loss due to electrospray capillary clogging, interruption by periodic maintenance (e.g. to clean the EESI and capillary into the TOF and to regenerate the denuder) and calibration by nebulising levoglucosan aerosol to quantify the mass concentration with an SMPS after each haze event (typically three to four days). Although the calibration by levoglucosan could indicate the instrument's linear response to mass concentration, the sensitivity to levoglucosan was found to be different in between different haze events because of the interruption. Therefore, we select a diagnostic species that can be measured with higher time resolution is utilised to monitor the sensitivity throughout the campaign. Intercomparison of inorganic nitrate species (AMS $NO_3^-$ and EESI-TOF $[NaNO_3]Na^+$) yield strong correlations during periods in which the instrument operation was stable (i.e. not affected by major clogging or cleaning/realignment of the electrospray capillary). Note that these issues are expected to result in changes in EESI-TOF sensitivity that uniformly affect all measured ions (i.e. without compound-dependent effects). Therefore, we correct for these compound-independent effects by comparing the nitrate signal ($[NaNO_3]Na^+$) from the EESI-TOF and the nitrate concentration ($NO_3^-$) from the AMS (Fig. S1). The whole campaign was divided into different periods, and the slope of linear fit between EESI-TOF nitrate signal and AMS nitrate concentration in each period was taken as the sensitivity of EESI-TOF to nitrate in this period as shown in Eq. (S3a). The time period from 3 to 7 November was selected as a reference period and the sensitivity determined in other periods ($k_q$, with $q$ denoting the individual periods) was normalised to the sensitivity of reference period ($k_{ref}$). Finally, the data collected from EESI-TOF was normalised according to Eq. (S3b)

$$k_q = \left(\frac{I_{[NaNO_3]Na^+,EESI-TOF}}{I_{NO_3^-,AMS}}\right)_q \tag{S3a}$$

$$I'_{i,j,q} = I_{i,j,q} \times \frac{k_{ref}}{k_q} \tag{S3b}$$

Here $I_{[NaNO_3]Na^+,EESI-TOF}$ and $I_{NO_3^-,AMS}$ are the signal of $[NaNO_3]Na^+$ and $NO_3^-$ collected by EESI-TOF and AMS, respectively in Eq. (S3a). In Eq. (S3b), $I'_{i,j,q}$ and $I_{i,j,q}$ indicate the signal of the $i$th ion at time point $j$ in $p$th period after and before normalisation, respectively, and $k_{ref}$ is the reference sensitivity. This correction accounts for time-dependent changes in sensitivity that uniformly affect all measured ions.

Therefore, the new section 2.2.1 has the brief introduction of instrument (paragraph 1), brief summary of operational and analytical challenges and the solutions(paragraph 2), data treatment procedures (paragraph 3) and the preparation of EESI-TOF data into PMF (paragraph 4). Please see the revised text (page 3 line 44):

[revised manuscript text omitted]

**Comment #20--- Specific Comments, Methodology**

*Line 42 page 6 - "as discussed in Sect. 2.3" - this should either be as "will" be discussed or should be removed.*
**Response:**

The suggested change has been implemented.

**Comment #21--- Specific Comments, Methodology**

*Line 41 page 6 to line 9 page 7 - This discussion of PMF and its error seems to be out of place in this section, it should be moved to section 2.3.*

**Response:**

This discussion does not address the PMF model directly, but rather the generation of EESI-TOF data products that are necessary as model inputs. As the preceding discussion of EESI-TOF operation, performance issues, and data analysis protocols are directly relevant to the generation of these data products, we feel it is easier for the reader to follow if the discussion is not broken between sections. (Likewise, generation of AMS model inputs is discussed in the AMS instrument section.) The PMF section then focuses on model theory and operation, which is largely independent of data source. Therefore we retain the identified text in its original location.

**Comment #22--- Specific Comments, Methodology**

*Lines 15 -28 page 8 - explain and justify potential biases in the interpretation of EESITOF data by constraining time series' to those of AMS factors, given that EESI-TOF factors, similar to CIMS factors have previously provided factor TS which are distinctly different those from the AMS, e.g. Massolli et al (2018), Yan et al (2016) and Chen et al. (2020).*

**Response:**

We agree that combining information from multiple instruments in a source apportionment analysis can make interpretation more complex. However, there is a critical difference between combining data from instruments that measure the same gas and/or particle ensemble (e.g. AMS and EESI-TOF) and those measuring different ones (e.g. AMS and gas-phase CIMS). In the latter case, one might indeed bias the results, as the possibility would exist for molecules and/or factors to be orthogonal to each other or in opposition due to, e.g. temperature-induced changes in gas/particle partitioning and reaction of primary gases to form SOA. However, the AMS/EESI-TOF combination used here is fundamentally different, because both systems are measuring the same set of molecules, i.e. OA. The instruments describe OA using different variables, but are fundamentally

measuring the same quantity. Therefore there is no reason to suppose that constraining factors from one instrument into the other would introduce a systematic bias. The exception to this would be a factor such as HOA, which is detected by the AMS but consists almost entirely of compounds to which the EESI-TOF is insensitive. This could indeed generate a bias, and for that reason it is excluded from the EESI-TOF analysis.

Of the studies cited by the reviewer, Massolli et al. (2018) and Yan et al. (2016) measured the gas phase via CIMS, and retrieved factors that indeed might be difficult or inappropriate to integrate with AMS data. In contrast, while the particle-phase FIGAERO factors retrieved by Chen et al. (2020) do not have resolved analogues in the corresponding AMS dataset due to the higher chemical resolution of the FIGAERO, integration of the datasets would be warranted, as the time series of AMS SOA factors can be thought of as linear combinations of FIGAERO SOA factors. This is qualitatively implied by the authors, although not quantitatively attempted.

Rather than biases, the question for the EESI-TOF/AMS combination employed here is whether the AMS factor time series are sufficiently distinctive to allow good separation of corresponding EESI-TOF factors. This is assessed through *a*-value exploration and bootstrapping, as discussed in response to R1C2.

**Comment #23--- Specific Comments, Methodology**

*Check how you wish to refer to EESI through the manuscript, it varies between EESITOF and EESI-TOF-MS. Choose one.*
**Response:**

EESI-TOF is now used throughout the manuscript to describe the instrument and its outputs. EESI is used when specifically discussing the ionisation source.

**Comment #24--- Specific Comments, Methodology**

*Line 41 page 8 - determine should be "determination" Line 42 page 8 - "compare the" should be "comparison of" Line 44 page 8 - z-score is mentioned here, either do not mention or signpost the reader to where it will be discussed.*

**Response:**

The textual errors are corrected as suggested, and the sentence referring to z-score is removed. To better alert the reader to the relevance of z-score, section 2.3 has been retitled "z-score analysis of factor profiles."

**Comment #25--- Specific Comments, Methodology**

*Major comment - AMS factors and EESI-TOF or CIMS factors are not expected to be directly related to one another, by constraining each EESI factor to an AMS factor solution is it not possible that you are introducing a*

*bias which limits the extent to which you are describing chemistry in your interpretation and are instead biasing the results towards primary sources.*

**Response:**

This issue was discussed in detail in response to R1C22. To briefly summarise, there is no reason to expect a bias from the AMS/EESI-TOF combination, because both instruments are measuring the same entity (i.e. OA). This is fundamentally different to the AMS/gas-phase CIMS combination, where the potential for orthogonal or opposing behaviours between gas and particles could indeed introduce a bias. However, in the present case, every factor measured by the AMS is composed of variables that have (possibly complex) analogues in the EESI-TOF, (with the exception of HOA, which is excluded for this very reason). As a result, there is no reason to suspect a bias.

**Comment #26--- Specific Comments, Methodology**

*Throughout "as discussed" refers to past tense, it should be changed to "will be discussed in".*

**Response:**

We now utilise "as was discussed" to refer to information conveyed earlier in the text and "as will be discussed" to describe information conveyed at a later point.

**Comment #27--- Specific Comments, Results**

*Page 10 lines 46-47 - criteria used to define "haze episodes" should be defined here rather than in figure caption for Figure 1.*

**Response:**

The definition of light and severe haze episode has been added to the first paragraph of the section, as follows (page 9 line 7):

During the measurement period, we observed nine haze episodes, classified as light haze (NR-PM$_{2.5}$ concentrations from 20 to 150 µg m$^{-3}$) or severe haze (NR-PM$_{2.5}$ concentrations above 150 µg m$^{-3}$).

**Comment #28--- Specific Comments, Results**

*Page 10 line 20 - 37 % is written twice. Needs to be corrected.*

**Response:**

The latter 37% is deleted.

**Comment #29--- Specific Comments, Results**

*Figure 1 - Several corrections are needed. 1) y-ticks are overlapping between (b) and (c ). 2) y-label for (c )*
*should specify if this is PM2.5 mass, which I assume is the case, and also include units in brackets. 3) y-label*
*for (d) should be changed to fractional factor contribution or something more clear 4) Colorscale of W_dir*
*should be labelled as wind direction, which is what I assume it is? It would also be better not to use rainbow. 5)*
*Legend for light and severe haze should be made clearer.*

**Response:**

Figure 1 is changed accordingly; the revised figure is shown below.

[Figure]

**Comment #30--- Specific Comments, Results**

*Page 12 lines 9-10 - it should be specified which HR ions were used and where UMR sticks were used. A table*
*in the supplement would be a good way to record this.*

**Response:**

The construction of the AMS data matrix was explicitly discussed in the Methods (section 2.2.2) in the original

manuscript as follows: "The data matrix was constructed by including both (1) ions with known molecular

formula for $m/z \leq 120$ and (2) the integrated signal across each integer $m/z$ for $m/z$ 121 to $m/z$ 300." We feel this

explanation is clear and most appropriate to the Methods, so have retained the original text (page 6 line 10).

A list of HR ions used in AMS peak fitting has been added to the supplement as Table S2.

**Comment #31--- Specific Comments, Results**

*Page 12 line 18 - page 13 line 8- Here you discuss that the apportionment of 4 OOA factors is unusual, and point the reader to section 3.3. for interpretation. Is it the case that 4 factors are chosen in the AMS data to better represent the EESI-TOF data? Or were there factors chemically or temporally distinct enough in the AMS data alone that the EESI factors did not influence this selection? It is important to acknowledge how the EESI factor identification has influenced the factor selection for the AMS and vice versa.*

**Response:**

The source apportionment method was described in section 2.3 ("Source Apportionment Technique) as follows (page 6 line 36): "PMF on AMS-only data to determine factor time series, followed by PMF on EESI-TOF-only data with constrained factor time series to facilitate chemical interpretation of the AMS-determined factors." This statement directly answers the questions raised by the reviewer. Specifically, the 4-factor description of OOA arises solely from AMS data, because only at the AMS stage are factor time series defined. The EESI-TOF results influence the source apportionment only in the sense that they facilitate interpretation of the factors and thus support the decision to retain a solution with 4 OOA factors rather than a solution with fewer factors.

**Comment #32-- Specific Comments, Results**

*Figure 2a - y-ticks are overlapping, sticks are feint and their thickness needs to be increased. The region above m/z 120 should be magnified to see clearer patterns in the higher m/z ions which are poorly represented in this figure as is.*

**Response:**

Figure 2a is changed accordingly. The UMR y-axis is magnified by a factor of 5.

a)

$C_x$, $C_xH_y$, $C_xH_yO$, $C_xH_yO_{z>1}$, $C_xH_yN$, $C_xH_yON_z$, $C_xH_yO_{p>1}N_z$, $C_xS_y$, $H_xO_y$, UMR

[Figure]

b)

[Figure]

**Comment #33-- Specific Comments, Results**

*Figures 2-4 are grainy and need producing at higher quality for ACP.*

**Response:**

The resolution of Figs. 2 to 4 have been increased, and substituted by high-resolution figure.

**Comment #34-- Specific Comments, Results**

*Description of the AMS factors should refer to the relevant figures, e.g. diurnal, time series or mass spectra. Descriptions of factor time series and diurnals have various inconsistencies. For HOA, the correlation is not shown in the supplement in Fig. S13 as stated in the manuscript. This should be included. You have stated that the factor peaks between 0600 and 0900 which is not what is shown in Figure 3a. This figure shows elevated concentrations overnight, with the increase between 0600 and 0900 only reflected in the standard deviations, not in the mean. It would be useful to see the diurnal variations split between the seasons as it appears that there is a high degree of variability which means that the average diurnal profile does not show the temporal profile described in the manuscript. I would argue that the morning peak from 0700 to 0900 is also missing in your data, given it is only shown in the standard deviations. The caption of Figure 3 needs to be changed, it is not the "grey area" which represents standard deviations, it is the shaded area. The high contribution of PAHs to the CCOA factor referred to need to be shown more explicitly in the mass spectra by enlarging the MS in the high mass regions in Figure 2a as mentioned previously. The aromatic signatures described for CCOA need to be evidenced, there is no references included or explanation of how they have been identified to be aromatic in origin. The description of OOAs previously classified on page 14 lines 36-39 need to be referenced.*

**Response:**

Additional references to the respective figures have been added to the AMS factor discussion.

The original version of Fig. S13 showed overlaid time series of HOA and CO. We have revised the figure to make the plot clearer (and include comparisons of other factors with external data) and presented in Fig S16. We now also report correlation coefficients, which for HOA and CO yields $r^2 = 0.50$.

Regarding the diurnals, standard deviations were not shown in the original text. The shaded area instead represented the 1st and 3rd quartiles. The figure below shows the mean concentrations of the diurnal cycles of these eight factors, and dashed lines stand for mean ± standard deviation. Clearly, the increase in 0600 to 0900 is indeed a systematic, recurring feature in the data, at least on days when the HOA concentration is high. It is more clear also to see the elevated concentrations overnight for HOA.

[Figure]

(c)

The caption is changed as suggested from "grey area" to "shaded area".

The attribution of $m/z$ 115 ($C_9H_7^+$), 128, 139, 152, 165, 178, 189, 202, 215, 226, 239 and 252 to aromatics and PAHs is based on the work of Dzepina et al. (2007). The revised text reads (page 11 line 17):

"A series of strong signals are found in the factor profile at $m/z$ 115 ($C_9H_7^+$), 128, 139, 152, 165, 178, 189, 202, 215, 226, 239 and 252, which have been shown to be characteristic of aromatics and PAHs (Dzepina et al., 2007)."

The description of OOAs previously classified on page 14 lines 36-39 is referenced.

**Comment #35-- Specific Comments, Results**

*Discussion of EESI factors on page 17 onwards is unclear. The author states that "PMF of the EESI-TOF mass spectral time series was conducted on a 7-factor solution". Surely the PMF was conducted to derive 7 factors? There is no explanation here of how those 7-factors were selected. In addition, there is no detail provided on the influence and biases introduced from constraining EESI factor time series to the seven non-HOA factors. As highlighted above, factors derived from near-molecular-level techniques typically have more distinct time series than those derived from AMS studies. It should be better justified why this approach to constrain TS was taken and if any effort was made to run the PMF of the EESI factors without constraining the TS to that of the AMS factors.*

**Response:**

This issue was addressed in detail in response to R1C1. As noted in the original manuscript, the source apportionment method utilized in this manuscript consisted of PMF on AMS data (retrieving 8 factors), followed by an EESI-TOF PMF in which all factors were constrained by AMS time series (7 factors, with the HOA factor retrieved from AMS PMF excluded because the EESI-TOF cannot detect hydrocarbons). Separately, a standalone EESI-TOF PMF was explored but found to yield unsatisfactory results, likely due to operational issues (i.e., denuder breakthrough) encountered during the campaign. As a result, the original text correctly describes the analysis method used to generate the results presented in the main text, and is retained.

Note that further clarification of the analysis strategy and a more detailed justification is provided in response to R1C1.

**Comment #36-- Specific Comments, Results**

*AMS and EESI factors should be named more clearly. As in Stefenneli and Qi papers, these were named with EESI or AMS subscripts. This would make studying the figures more clear. Numbering of Figures is incorrect after Figure 5. The text here refers to Figure 7 when presumably it should be when the van krevelen plot shows as Figure 4 and according to the sequence should be Figure 6. References to figures should be checked, corrected and improved throughout. Markers on the van krevelen plot are too small and should be changed. It is stated that the sizing of these markers is related to the z-score, however this cannot be interpreted from the figure.*

**Response:**

Our analysis differs from that of Qi et al. (2019) and Stefenelli et al. (2019) studies, in which PMF was conducted on AMS and EESI-TOF datasets separately. In this study, factor resolution depends entirely on AMS PMF, with the EESI-TOF PMF constrained by AMS time series (see R1C35). Therefore, the AMS and EESI-TOF factors are by definition linked, and we do not differentiate with instrument subscripts.

We thank the reviewer for noticing the figure numbering problem. The indexing became corrupted during conversion to PDF and has been corrected.

For the Van Krevelen pots, the big stars indicate the ions with z-score>1.5, whereas the small stars indicate ions with z-score < 1.5. This is now stated in the figure caption.

**Comment #37-- Specific Comments, Results**

*Defining COA on the basis of it having most of the mass at ions with high m/z needs to be better explained. Particularly as ions at high m/z in AMS are attributed to PAHs. Intense ions in the mass spectra shown in Figure 4 should be labelled. CCOA , figure numbers are not included and are blank and should be included. Figure 4 needs improving, the factor time series are unclear and variations cannot be seen in the aspect ratios the figures have been output in. The x-ticks need to be made more regular. The mass spectra y-ticks are overlapping, and in some cases such as CCOA, the mass spectra cannot be clearly seen due to intense ions. The presentation needs to be improved.*

**Response:**

We have revised the discussion of COA to clarify that the ions at high *m/z* are slightly oxygenated and have high H:C. This distinguishes them from the PAHs observed by the AMS for CCOA; in any case, PAHs are not detectable by the EESI-TOF. The revised text follows (page 12 line 36):

"Consistent with Qi et al. (2019) and Stefenelli et al. (2019), the mass spectrum of this factor is characterised by large contributions at high *m/z* from slightly oxygenated ions with high H:C ratios."

We appreciate the helpful suggestions on figure quality, all of which have been incorporated into the manuscript. In addition, the figures have been expanded horizontally to make full use of the available column width.

**Comment #38--- Specific Comments, Results**

*Correspondences between the AMS and EESI SOA factors need to be better justified, with comparisons of time series presented. AMS MO-OOA and EESI-TOF MO-OOA being related to one another is not explained and simply assumed in the manuscript. Have the ions listed in paragraph 2 on page 21 been observed elsewhere, these ions formulas should be explained. Various nitro-aromatics in Beijing have been reported previously. Can you explain why the ions stated on line 46 of page 21 are relevant from aromatics? This should be justified. Page 22 line 18, has this ion been previously observed and if so, where? Bertrand et al reference has no date, this needs to be corrected.*

**Response:**

The issue of correspondence between the AMS and EESI-TOF time series for all factors was addressed in our response to R1C1. A time series comparison has been added to the supplement as Fig. S17. Note that the EESI-TOF and AMS factor time series are not independent, as the AMS time series were used on a factor-by-factor basis to constrain the EESI-TOF solution.

$C_6H_{10}O_5$ is a well-established tracer for primary biomass burning, which is observed in BBOA but not MO-$OOA_{SFC}$ likely due to aging.

$C_{10}H_{16}O_x$ (x = 3,4,5,6…), are also apportioned to SFC-related factors. These ions are observed also in Qi et al. (2019).

Nitrocatechol ($C_6H_5NO_4$) and its homologous series ($C_7H_7NO_4$, $C_8H_9NO_4$) were observed as oxidation products of wood burning emissions by Mohr et al. (2013). However, here these ions don't have strong intensity and are apportioned to different factors. Ion series of $C_6H_{11}NO_4$, $C_7H_{13}NO_4$, $C_8H_{15}NO_4$, $C_9H_{17}NO_4$ and $C_{10}H_{19}NO_4$ are observed in Qi et al. (2019). These ions were not documented from observation before, but judging from the molecular formula, they might be organonitrate resulting from the high $NO_x$ concentration in Beijing.

The ions discussed on page 21/line 46 are $C_{12}H_{10}O_8$ and $C_{16}H_{14}O_6$. To our knowledge, these ions have not been identified as tracers for specific sources. Here, they are tentatively attributed to aromatics based on their low H:C ratio. Further, the observation of higher relative contributions from low H:C ions was observed in aged wood burning emissions (Qi et al., 2019). These points have been clarified in the text as follows (page 13 line 44):

"The temporal evolution of these ions is consistent with EESI-TOF ions having a low H:C ratio and thus tentatively attributed to aromatics e.g. $C_{12}H_{10}O_8$ and $C_{16}H_{14}O_6$ (see Fig. S25). This also suggests an elevated contribution from aromatic oxidation relative to the non-SFC-derived SOA factors. An increased contribution from ions with low H:C was also observed in oxidised wood burning emissions by Qi et al. (2019)."

Bertrand et al is not published and we now reference it as private communication.

**Comment #39--- Specific Comments, Results**

*The description of MO-OOA relies on a discussion of haze and non-haze events, which are not depicted on the figures except the final figure (Figure 9) and even in this case they are not clearly shown. Figures need to be improved so the interpretation of factors can be followed. Small acids can also be derived from aromatic oxidation (Zaytsev et al. 2019, Mehra et al. 2020, Wang et al. 2020). Their attribution to aqueous phase chemistry needs to be better explained. Line 44 page 22, "section x" needs to be replaced by a section number. Discussion of AMS and EESI factors is confusing due to the identical naming, these need to be better distinguished in the text.*

**Response:**

Haze events were identified via shading in Fig. 1, but not used in other figures (including Fig. 9, now Fig. 12). We now apply this shading to all figures in the main text (Figs. 3a, 3b, 4a, 10a, 10c, and 12) to clearly denote haze events.

We agree with the reviewer that both small acids and larger oxygenated molecules can be derived from aromatic oxidation. In the absence of aqueous chemistry (or other heterogeneous processes), the ratio of small acids to larger oxygenates is governed by partitioning. Aqueous chemistry provides an additional pathway for organic molecules to enter the condensed phase that favors highly soluble organics such as small acids. Therefore, although small acids are not a unique tracer for aqueous chemistry, an enhanced fraction of small acids is expected in aqueous chemistry-influenced SOA. In the current dataset, the observation of an increased contribution of small acids coincides with other observations consistent with aqueous chemistry (e.g. high LWC). Taken together, they strongly suggest a large contribution from aqueous reactions. This has been clarified in the text as follows (page 14 line 27):

"These molecules can enter the particle via aqueous-phase chemistry (Tan et al., 2012; Tan et al., 2010; Carlton et al., 2007; Ervens et al., 2004), or as condensation products of gas-phase reactions (Mehra et al., 2020; Wang et al., 2020; Zaytsev et al., 2019); Legrand et al., 2005; Sellegri et al., 2003; Sempere and Kawamura, 1994). For example, Lamkaddam et al. (2021) have shown that up to 70% of isoprene oxidation products can be dissolved in a water film. However, because aqueous reaction pathways under subsaturated conditions favor the uptake of highly soluble molecules such as small acids/diacids, their contribution relative to larger oxygenates is increased, consistent with the lower $CO^+/CO_2^+$ slope observed here."

Line 44 in page 22 is corrected now. "see section S3"

Naming of EESI-TOF and AMS factors (i.e., lack of subscripts) was discussed in R1C36.

**Comment #40--- Specific Comments, Atmospheric Implications**

*Values are missing on line 15 page 29 of "concentrations between x and y". Page 29 lines 14-48, the discussion of "minor" and "major" haze events is unclear and Figure 9 which shows the values discussed in this section is poorly put together, it is unclear which events each pie chart represents making it impossible to interpret. This needs to be improved. The events discussed in the text have dates, which cannot be established from looking at the figures or the pie charts.*

**Response:**

The reviewer identifies three issues that make this section difficult to interpret: (1) undefined terminology for classification of haze events; (2) unclear presentation of Fig. 9 (now Fig. 12); (3) missing values in the text. Regarding terminology, "major" and "minor" haze events refer to the "severe haze" and "light haze" periods, respectively identified in Fig. 1. We now use "severe haze" and "light haze" exclusively throughout the manuscript. Figure 9 (now correctly numbered as Fig. 12) has been clarified by expanding the horizontal scale, adding the severe haze/light haze/clean shading from Fig. 1, and connecting the pie charts to their respective haze periods with arrows.. Finally, the missing values have been added, which state that the three light haze events had maximum concentrations between 29.8 and 42.0 μg m$^{-3}$.

**Comment #41--- Specific Comments, Atmospheric Implications**

*The conclusion that oxidation of aromatics from SFC is responsible for SOA is poorly evidenced (lines 44-48). The evidence for the aqueous factor is stronger, however, certain aspects need improving as discussed above*

**Response:**

We have clarified the statement identified by the reviewer. Indeed we do not present new evidence in this section that bears on the contribution of SFC to SOA; rather this statement is simply highlighting the fraction of total OOA attributable to the sum of MO-OOA$_{SFC}$ and LO-OOA$_{SFC}$. The attribution of these factors to the oxidation of SFC emissions was discussed in detail in sections 3.3.2 and 3.3.3, and above in response to R1C38. The revised text follows (page 16 line 39):

"Two salient features are: 1) in the heating season, SOA formation is driven by oxidation of aromatics from solid fuel combustion, with secondary SFC-related factors (i.e., sum of MO-OOA$_{SFC}$ and LO-OOA$_{SFC}$) contributing 37.2 % to 72.8 % of total SOA, and 2) under high NO$_x$ and RH conditions, aqueous phase chemistry may make a major contribution to SOA formation (with MO-OOA$_{aq}$ comprising 53.7 % of total SOA)."

Similarly, the statement on aqueous factor contributions depends on previous discussion for factor identification, which is found in section 3.3.4 and above in response to R1C39.

**Comment #42--- Specific Comments, Conclusion**

*Overall, the chemical resolution of the EESI is not discussed in great detail, many of the ions used in the PMF are not shown or discussed in the manuscript. This should be improved.*

**Response:**

This issue was discussed in response to R1C2 and the relevant text is repeated here.

In a complex environment such as Beijing, it is frequently the case that an ion molecular formula alone is not sufficient to establish a clear link to a specific source. There are exceptions to this, which are discussed in the paper. For example, levoglucosan is used as a tracer for biomass burning activities, small and diacids are related to aqueous phase chemistry, and fatty acids are related to cooking activities. However, the chemical resolution of the EESI-TOF is also essential for assessing patterns in the factor profiles, such as the differences in carbon number and H:C ratio shown in Figs. 5 and 6 (now Figs. 8 and 9). These systematic features are crucial to the interpretation of the results and, we believe, provide explanatory power that is at least equivalent to inspection of individual ions.

The treatment of individual ions has been improved through more systematic referencing, labeling of major ions in the EESI-TOF factor profiles, and inclusion of a complete ion list in the supplement. However, in general we feel that the current balance between pattern-oriented analysis and individual ion inspection is appropriate given their relative information content, and that especially in the absence of structural information, a more comprehensive discussion the set of individual ions will rather obscure the key features of the dataset.

**Comment #43--- Specific Comments, Supplement**

*A more comprehensive list of ions should be presented in the supplement. The correlations of AMS nitrate and EESI shown in Figure S1 are poor for the first 2 time periods and should be discussed further.*

**Response:**

Two tables are added to the SI. Table S2 shows the AMS HR ion fragments and Table S3 shows the EESI-TOF ions included in the PMF matrix. Note that for the EESI-TOF, this table does not include ions determined to be background-dominated or for which a molecular formula could not be proposed.

We apologise for the confusion. The original figure grouped together several periods over which different fits were conducted. The figure has been revised such that each panel now corresponds to a unique fit, also shown in Fig. S2

[Figure]

[Figure]

[Figure]

EESI NaNO$_3$Na$^+$ vs AMS Nitrate
4 Nov to 7 Nov, subepisode 3

y = 1.06 x + 1.27
r$^2$ = 0.93

EESI NaNO$_3$Na$^+$ (cps)

AMS Nitrate (µg m$^{-3}$)

Date and time

06.11.2017 00:00
18:00
12:00
06:00
05.11.2017 00:00

[Figure]

EESI NaNO$_3$Na$^+$ vs AMS Nitrate
4 Nov to 7 Nov, subepisode 4

y = 1.06 x - 11.72
r$^2$ = 0.82

EESI NaNO$_3$Na$^+$ (cps)

AMS Nitrate (µg m$^{-3}$)

Date and time

09:00
08:00
07:00
06:00
05:00
04:00
06.11.2017 03:00

[Figure]

**Comment #44--- Specific Comments, Supplement**

*Labelling of supplement factors needs to be improved. "Splitted" is not correct English and needs changing in all figures.*

**Response:**

This error has been corrected; "splitted" is replaced with "split".

---

## Author Comment (AC2) · 14 Mar 2021

Dear Dr. Allan,

Please find enclosed our response to the reviewer #2 for manuscript acp-2020-835 ("Quantification of solid fuel combustion and aqueous chemistry contributions to secondary organic aerosol during wintertime haze events in Beijing").

This file contains a point-by-point response to reviewer #2, with changes to the text

marked in blue. Note that one reference (in this reply to reviewer #2) from Lee et al. (2021) is submitted to AMT. In the reference list it is:

Lee, C.P., Surdu, M., Bell, D., Lamkaddam, H., Wang, M., Ataei, F., Hofbauer, V., Lopez, B., Donahue, N., Dommen, J., Prevot, A. S. H., Slowik, J. G., Wang, D., Baltensperger, U., and El-Haddad, I.: Effects of aerosol size and coating thickness on the molecular detection using extractive electrospray ionization, Atmos. Meas.Tech., submitted, 2021.

We highlighted it in the reference list, and we will update this information once the DOI is available.

Thank you for considering our manuscript for ACP, and we look forward to your response.

Best regards,

Yandong TONG

Please also note the supplement to this comment:
https://acp.copernicus.org/preprints/acp-2020-835/acp-2020-835-AC2-supplement.pdf

**Supplement:**

**Response to RC2**

We thank the reviewer for the helpful comments. Below we provide a detailed point-by-point response to the issues raised by the reviewer. Reviewer comments provided in *italics* and our responses follow in normal text. Changes to the manuscript are denoted in blue font. When our responses reference other comments, we use the formalism R#C#, such that R1C5 would refer to Comment 5 by Reviewer 1.

**Comment #1**

*Page 4, Line 14: Were the aerosols dried prior to sampling? What was the average sample aerosol RH throughout the study? Were particle losses calculated for the PM1 to 2.5 range?*

**Response:**

As indicated in Section 2.2.2 in the main text, "a Polytube Dryer Gas Sample Dryer (Perma Pure LLC) was mounted in front of the AMS inlet". The average RH is between 20% to 30% after the dryer. The AMS lens transmission efficiency of particles from 1 to 2.5 µm is approximately unity (Williams et al., 2013), therefore particle losses were not calculated.

**Comment #2**

*Can the authors add the size range typically measured by the EESITOF to the discussion of particle size sampling efficiency on page 5/6.*

**Response:**

The EESI-TOF measures particles up to at least 750 nm diameter with better than 80% efficiency (Lopez-Hilfiker et al., 2019). Recent work in our laboratory shows a significant increase in sensitivity below 100 nm (Lee et al,.2021), but due to the small mass fraction contained in this size range, this size dependence has a negligible effect on the current study. The EESI-TOF has not been characterised in the laboratory for particles larger than 750 nm, but given the absence of systematic differences in EESI-TOF vs. AMS comparisons for clean vs. haze periods, which have mode diameters of 302 and 665 nm (see Fig. S3), respectively, with a significant mass fraction above 1 µm during haze (see response to R2C4, R2C5, and Fig. S3), we expect that the AMS and EESI-TOF measure approximately the same size fraction (roughly $PM_{2.5}$).

**Comment #3**

How often was the denuder regenerated during the sampling campaign?

**Response:**

We utilised two denuders for the campaign, with one in operation while the other was regenerated in an oven at ~200 °C. Each denuder alternated 24 hrs of sampling with 24 hrs regeneration. We now note this in the text as follows (page 4 line 5):

"After sampling for 24 hrs, the denuder was replaced, and regenerated for 24 hrs in an oven at ~200 °C."

**Comment #4**

*What is the make and model of the SMPS used in this study? What is the size range measured by the SMPS. Can the authors show the aerosol size distribution measured by the SMPS and compare with that of the L-TOF (shown in Fig S2).*

**Response:**

We have added the following sentence to section 2.1 (page 3 line 32):

"A scanning mobility particle sizer (SMPS), consisting of a model 3080 DMA and model 3022 CPC (TSI, Inc., Shoreview, MN, USA), an aethalometer (model AE33, Magee Scientific, Ljubljana, Slovenia) and an Xact 625i Ambient Metals Monitor (Cooper Environmental Services LLC, Tigard, Oregon, USA) were additionally deployed at the site to measure the particle size distribution from 15.7 to 850.5 nm, the equivalent black carbon (eBC) concentration and the mass of 35 different elements in $PM_{10}$ and $PM_{2.5}$, respectively."

We have added SMPS distributions to all plots of AMS size distributions. The figures below show the new Fig. S3a (average of all haze periods except 4 to 7 Nov.) and S3b (average of all clean periods). SMPS distributions are shown as mass distributions as a function of mobility diameter, with an effective density of 1.2 g cm$^{-3}$ used to estimate mass.

[Figure]

[Figure]

**Comment #5**

*How did this size range change during the different sampling events "haze events" and the "clean periods" (Heating non-heating)?*

**Response:**

Figure S3 (see response to previous comment) shows a mode vacuum aerodynamic diameter ($d_{va}$) of 302 nm for the clean periods and 665 nm for the haze episodes. Also of note, a significant fraction of mass occurs above 1 µm during haze, consistent with previous studies (Elser et al., 2016)

We have also added Figs. S3c-S3h to the supplement, which show SMPS and AMS size distributions for the following events: 10-11 Nov. (clean, non-heating), 22-24 Nov. (clean, heating), 11-13 Nov. (haze, non-heating), 30 Nov – 3 Dec (haze, heating), 4-7 Nov. (intense haze, non-heating, aqueous chemistry-influenced) and 18-22 Nov (intense haze, heating, strong SFC contribution). For clean periods, the the mode diameter during the non-heating season is larger than during the heating season. For haze periods, the mode diameters in the heating and non-heating seasons are usually comparable, with the exception of two severe haze events. These events, in which larger particles are observed are the 18 to 22 Nov severe haze characterised by biomass burning activity, and the aqueous chemistry-influenced severe haze from 4 to 7 Nov.

[Figure]

[Figure]

[Figure]

Haze period from 11 to 13 November

[Figure]

Haze period from 30 November to 3 December

[Figure]

[Figure]

**Comment #6**

*What is the difference in mass between the PM25 inlet and the SMPS. How representative of the total PM2.5 mass is the PM25 LTOFAMS measurements? Are auxiliary measurements of total PM available for comparison?*

**Response:**

We have added a comparison of AMS NR-PM$_{2.5}$ and estimated SMPS mass (assuming an effective density of 1.2 g cm$^{-3}$) to the supplement as Fig. S4, and show the figure below for convenience.

The time series and scatter plot of particle concentration measured by AMS and SMPS are shown below and in Fig S4. As indicated in R2C4, the size cut of SMPS is 850 nm. The comparison between them shows three

distinct slopes in the scatter plot. The data from 31 Oct to 24 Nov falls mostly on the 1:1 line, except for the intense haze from 4 to 7 Nov, when the AMS concentration is much higher than the SMPS. This is because a large fraction of the mass occurs above the SMPS cutoff of 850 nm. After 25 Nov, we continue to observe a strong correlation between the SMPS and AMS, but is nearly 2 times higher than the AMS. This is believed to be a problem with the SMPS number counts. It is unlikely to be a problem with the AMS, as the correlation of PM$_{2.5}$ elemental sulphur measured by a co-located Xact and AMS sulphate is consistent throughout the study. This comparison is added to the supplement as Fig. S4 and shown below.

[Figure]

[Figure]

[Figure]

**Comment #7**

*Given that the smaller, locally formed aerosol particles are not efficiently sampled by the PM25 inlet, what impact does this have on the interpretation of the measurements?*

**Response:**

As shown by Xu et al. (2017), the $PM_{2.5}$ lens provides similar or better transmission to the standard $PM_1$ lens, except for $d_{va}$ = 100-200 nm, where the $PM_{2.5}$ lens transmission is up to 50% lower. (Below 100 nm, the transmission decreases significantly but similarly for both lenses.) Assuming these particles are spherical and they have an effective density of 1.2 g $cm^{-3}$, this corresponds to a $d_m$ range of 83 – 167 nm. As discussed in response to R2C4 and R2C5 and shown in Figs. S3 and S4, this accounts for a negligible mass fraction for all conditions encountered in the study. Therefore it is highly unlikely that these losses significantly affect the mass-based source apportionment analyses conducted here. However, given the lack of clearly multi-modal size distributions in the AMS, we cannot rule out the likelihood of an underestimation of all factors during clean periods rather than a bias in the fractional apportionment

**Comment #8**

*Is a standard or capture vaporizer used in conjunction with the PM25 inlet? If a standard vaporizer is used, what is the particle collection efficiency estimated to be, and is the calculated CE dependent on particle diameters (to account for the enhanced effects of particle bounce for larger particles diameters)?*

**Response:**

The original manuscript stated that the composition-dependent collection efficiency (CDCE) method was used (Middlebrook et al., 2012). We now also note that the AMS utilises a standard vaporizer, and no correction for large particles was applied. This is now stated in section 2.2 as follows (page 5 line 34):

"The particle beam impacts on a heated tungsten surface (standard AMS vaporiser, ∼ 600 °C, and ∼ $10^{-7}$ Torr)"

"A composition-dependent collection efficiency (CDCE) was applied to correct the measured aerosol mass (Middlebrook et al., 2012), and no size-dependent CE corrections were applied."

**Comment #9**

*High concentrations were measured during this field campaign, often resulting in the clogging of the EESI-TOF instrument. Was there any evidence to suggest that there was overloading of the LTOFAMS instrument?*

**Response:**

We did not find any evidence for clogging or overloading of the LTOF-AMS during the campaign. Throughout the entire campaign, the variation of the flowrate was <1 %. the airbeam < 10 %.

**Comment #10**

*The authors show several PMF solution for the LTOFAMS analysis but only show the final solution for the EESI-TOF. Can the authors state if the PMF analysis on the "unconstrained" EESI TOF was performed? And if so which factors dominated the unconstrained PMF solution?*

**Response:**

This is discussed in detail in response to R1C1, and we partially repeat the response here. Briefly, a preliminary PMF of the standalone EESI-TOF dataset was attempted but found to be uninterpretable, likely due to the suboptimal response of the EESI-TOF to the high concentrations experienced in this campaign, specifically denuder breakthrough. The PMF model requires detector linearity and static factor profiles, both of which are compromised by denuder breakthrough, because the volatile and semivolatile contributions to factor profiles depend on the time-dependent state of the denuder. Denuder breakthrough effects have recently been characterised in detail by Brown et al. (2021). As discussed in sections 2.2.1 and Text S3, we have tried to reduce these effects by removing ions likely to be dominated by gas phase/denuder breakthrough, but this method does not allow quantitative correction of signals from ions having contributions from both phases. However, by constraining the EESI-TOF PMF solution with AMS factor profiles, the solution becomes weighted towards explaining temporal trends observed in the particle phase. Further, by utilising the EESI-TOF for qualitative (factor identification) rather than quantitative (factor resolution) purposes, we minimise the effects of artifacts introduced by gaseous signals.

**Comment #11**

*Aqueous phase SOA: The PMF analysis allowed the extraction of a more oxidized aqueous phase aerosol. In Fig. 3a the MOOA_AQ is shown to have highest concentrations (reaching 20 micrograms) during the haze event on the 4th and 7th, during high NO3 contributions (and high RH and low wind seed). The characteristic enhancement of the CO2 is illustrated in Fig. 7a during this period. However, there are several other periods during the field campaign when this MO-OOAaq species is identified (at concentrations near to 5 micrograms) (under the same conditions of RH and high NO3 Fraction) but the enhancement of the CO2+ signal was not observed. Is this CO2+ enhancement really associated with these species or is it somehow an artefact during high NO3 concentrations?*

**Response:**

The reviewer identifies two important issues: (1) whether the enhanced $CO_2^+/CO^+$ signal can be a measurement artefact due to high $NO_3^+$ concentrations and (2) the reasons for the lack of enhancement in $CO_2^+/CO^+$ during periods of detectable MO-OOA$_{aq}$ outside the 4-7 Nov. event. We address these questions separately below.

Regarding $CO_2^+$ artefacts, it is known that exposure to $NO_3^+$ leads to an artefact of increased $CO_2^+$ signal in the AMS (Pieber et al., 2016). As noted in the original text, we corrected for this effect by characterising the contribution from $CO_2^+$ and $CO^+$ during $NH_4NO_3$ calibrations, in which the concentration of $NH_4NO_3$ ranges from 5 to 80 $\mu g\ cm^{-3}$. The result shows that the artefact of $NH_4NO_3$ induced $CO_2^+$ signal according to Pieber et al. (2016) to total $CO_2^+$ signal is less than 5 %.

Regarding the second question, although MO-OOA$_{aq}$ is detected throughout the rest of the campaign, its percent contribution to OA does not exceed 10 % during clean periods. During these periods, the overall concentrations are low enough that the $CO^+$ signal is rather noisy. Figure S22 shows the time series of reconstructed $CO_2^+$ and $CO^+$, which is the cross product of the time series matrix (G) and $CO_2^+$ and $CO^+$ relative intensity in factor profile matrix (F). It shows high $CO_2^+/CO^+$ ratio during the haze event between 4 to 7 November and nearly 1:1 ratio during the other periods although the MO-OOA$_{aq}$ concentration is not negligible there. Compared to Fig. 10, the lack of significant increase of $CO_2^+/CO^+$ is a consequence of the OA composition.

[Figure]

**Comment #12**

*How did the factor mass spectral profile compare with reference mass spectra (oxalic acid, malonic acid and succinic acid (Canagaratna et al., 2015))?*

**Response:**

The mass spectra of factor MO-OOA$_{aq}$ and MO-OOA$_{SFC}$, and malonic acid ($C_3H_4O_4$), oxalic acid ($C_2H_2O_4$) and succinic acid ($C_4H_6O_4$) are shown below and in Fig S21. The latter three species were measured with ammonium sulphate seeds under the environment of argon gas, therefore, huge peaks at m/z 18, 40 and 64 can be found. The ratio of $CO_2^+$/ $CO^+$ is 1.88, 1.34, 3.96, 1.69 and 0.56 for these five spectra, respectively. The acids are characterised by the $C_xH_yO_{z>1}$ group, and these ions have also although not outstanding, but high signal in the MO-OOA$_{aq}$ and MO-OOA$_{SFC}$ factors. Although MO-OOA$_{SFC}$ has also ions in this group, the ratio of $CO_2^+$/ $CO^+$ is much lower than the one of MO-OOA$_{aq}$.

[Figure]

[Figure]

**Comment #13**

*The authors state that the m/z 44 artefact is very low in this instrument (4%), however, could this CO2+ enhancement be somehow related to artefacts linked to mixtures of inorganic and organic species?*

**Response:**

Pieber et al. (2016) investigated inorganic/organic mixtures, and showed that the artefact depended only on the inorganic fraction (nitrate and, to a lesser degree, sulphate). This has been further supported by the work of Freney et al. (2019) in investigations of laboratory and ambient aerosol.

**Comment #14**

*Previous studies have shown how aerosol liquid water can promote the formation of water-soluble organic nitrogen (Yu Xu et al., 2020 Environ. Sci. Tech. https://pubs.acs.org/doi/pdf/10.1021/acs.est.9b05849). What is the role of nitrogen in the formation of these aqueous organic species? Is there evidence of organonitrate species? Has aerosol acidity being evaluated during these measurements (NOT measured, but can look at the ion balance)? During the intense haze episode, this MOOA species was measured continuously over a period of 3 days. In one of the cited articles (Kuang et al., 2020) it is mentioned that most of the aqSOA was formed during daytime periods with high photochemical activity and that dark aqSOA only contributed negligibly to the total OOA concentrations. In this work, the increase in aqSOA remains constant over three days with little*

*diurnal Variation. Do you consider that this aqSOA is locally formed or influenced by regional processes? Does the aerosol size distribution provide information to determine this?*

**Response:**

As indicated in Fig. 11, the airmass passed through a high $NO_x$ region, which provides an opportunity for inorganic nitrate formation homogeneously and/or heterogeneously, resulting in water uptake from the air and increased LWC concentration. Then this high LWC in turn facilitates further heterogeneous formation of nitrate. The high LWC also provides the environment for other aqueous phase chemistry.

Regarding organonitrates, *z*-score analysis of the MO-OOA$_{aq}$ EESI-TOF profile identifies the nitrogen-containing species $C_9H_{15}NO_6$, $C_{10}H_{17}NO_5$ and $C_9H_{15}NO_5$ as ions characteristic of this factor, suspected to be organonitrates. The stack plots of these three ions in different factors and their contribution to total OA are shown below (also in Fig. S26). However, the organonitrate can be formed via aqueous phase chemistry and via gas phase reaction. From the EESI-TOF data alone, we cannot conclude that these organonitrates are only from aqueous phase chemistry.

From the AMS perspective, the ratio of $NO^+$ to $NO_2^+$ throughout the campaign is about 3.18, as shown in the first figure below (also in Fig. S23), whereas for the particular event from 4 to 7 Nov dominated by aqueous phase chemistry, the ratio is still about 3.13, as shown in the second scatter plot below. This ratio is close to the ratio of ammonium nitrate measured by the AMS from our IE calibration period (3.29) and from another study (3.5) (Sun et al., 2012), and much lower than the ratio of potassium nitrate (~28) (Drewnick et al., 2015). The $NO_3^-$ concentration resulting from $KNO_3$ and $NaNO_3$ in this campaign is rather low, about 0.3 % (if we asume all $K^+$ and $Na^+$ coming from $KNO_3$ and $NaNO_3$), making an insignificant contribution to $NO_3^-$ measured by the AMS. Considering the low ratio of $NO^+$ to $NO_2^+$ and the low contribution from $KNO_3$ and $NaNO_3$ to total $NO_3^-$, it is very likely that the ratio of $NO^+$ to $NO_2^+$ for this particular event from 4 to 7 Nov is governed by inorganic nitrate.

Regarding the time series for MO-OOA$_{aq}$, we calculated the change of the MO-OOA$_{aq}$ concentration over time. The last figure in the reply to this comment (also shown in Fig. S24) shows the time series of MO-OOA$_{aq}$ and the LWC in the upper panel, and the change of the MO-OOA$_{aq}$ concentration over time (2 h time interval) in the lower panel. From the time series, no diurnal cycle for the haze event from 4 to 7 November is observed although there are some short periods when the MO-OOA$_{aq}$ concentration decreases slightly, i.e., $\Delta$MO-OOA$_{aq}/\Delta t < 0$. In addition, different from the MO-OOA$_{aq}$ time series, the LWC concentration has a clear diurnal variation, as shown in Fig. 10. These observations are consistent with irreversible generation (as we suggest in the main text), although evaporation of reversibly-generated MO-OOA$_{aq}$ that happens to be compensated for by a comparable increase in MO-OOA$_{aq}$ production during the day cannot be completely ruled out.

As discussed in the text, the haze event featured by aqueous phase chemistry is considered to be both local and regional, because 1) the airmass passed through the high $NO_x$ region resulting water uptake from the air and LWC concentration and on the way to Beijing, the airmass stayed long enough for aqueous phase chemistry to happen and 2) the stagnant condition in Beijing contributed to the accumulation of pollutants and haze formation. Therefore, we consider this event is considered as both locally and regionally influenced.

[Figure]

[Figure]

[Figure]

[Figure]

Stack Plot of $C_9H_{15}NO_6$

[Figure]

Stack Plot of $C_{10}H_{17}NO_5$

[Figure]

Stack Plot of $C_9H_{15}NO_5$

[Figure]

**Comment #15**

*A non source-specific factor LO-OOA ns ?*

*Recently it has been shown that the PM25 inlet AMS systems may be capable of measuring airborne bacteria (Wolf et al., 2017 Atmospheric environment, (https://doi.org/10.1016/j.atmosenv.2017.04.001).). In this paper, there are some characteristics of the LO-OOAns species (O:C, diurnal pattern, higher concentrations during warm period than colder periods) as the resolved bacteria-like factor in Wolf et al.,2017. Is it possible to provide more information on the average diameter as a function of time for each of the resolved factors to help provide more information on their source and atmospheric processing prior to being sampled? At least provide the SMPS size distributions which would help illustrate regionally influenced factors and those from local processes.*

**Response:**

We cannot conclusively provide time-dependent size distributions for each factor, but some rough estimates are possible.

We checked the explained variation of each ion in LO-OOA$_{ns}$. If the explained variation of one ion in this factor is higher than in any other factor and the unexplained part, we consider this ion to be a surrogate for the LO-OOA$_{ns}$ factor. There are $C_2H_3^+$, $C_3H_3^+$, $C_3H_6^+$, $C_4H_5^+$, $C_3H_9O_2^+$, $C_6H_6^+$ and $C_6H_{12}^+$. These ions have highest intensity and fractional contribution at the corresponding integer *m/z*, therefore, we use the integer *m/z* to represent each corresponding ion here. In addition, a high fraction of *m/z* 44 is also observed in this factor, which is a surrogate for secondary factors. Therefore, we plot the geometric mean diameter of size distribution of the integer *m/z* and *m/z* 44 at UMR as a function of time for all periods. For other factors, we use nitrate as a surrogate for MO-OOA$_{aq}$, and sulphate for other OOA factors, shown in the figures below and in Fig. S27.

In the minor haze events, from Fig 12, the contribution from LO-OOA$_{ns}$ is the highest among these resolved factors, however, the size distributions of these ions are comparable to their size distribution in the other

periods: 587 nm in minor haze events vs 839 nm in the major haze event from 4 to 7 Nov, 550 nm in the major haze event from 18 to 22 Nov and 625 nm in the major haze event from 30 Nov to 3 Dec.

We consider it very unlikely that the LO-OOA$_{ns}$ factor retrieved here is bacteria-related. There are major chemical differences between LO-OOA$_{ns}$ and the bacteria factor of Wolf et al. (2017). That study identified the bacteria OA based on ions in the C$_x$H$_y$N group at $m/z$ 27, 30, 42, whereas in this paper, the surrogates for LO-OOA$_{ns}$ listed previously don't contain nitrogen. In addition, the bacteria factor is a minor faction (1.41 % in PM$_{2.5}$ and 0.52 % in PM$_1$) in Wolf et al. (2017), whereas in this paper, we observed high contribution of this LO-OOA$_{ns}$ factor during the haze events. Therefore, it is very unlikely that the LO-OOA$_{ns}$ is related to bacteria.

[Figure]

**Comment #16**

*Were BC measurements available for correlation with the HOA and BBOA.*

**Response:**

The equivalent black carbon (eBC) was measured by an aethalometer (model AE33, Magee Scientific). Here are the time series comparison and scatter plot (also in Fig. S16) between eBC and 1) HOA and 2) BBOA. Clearly, the correlation of eBC vs HOA is good, with a slope of 6.26 and $r^2 = 0.70$, consistent with the study from Poulain et al. (2021), but this slope is higher than the value typically reported in China (Zhu et al., 2018; Hu et al., 2016; Huang et al., 2010). The scatter plot of eBC vs BBOA is split into two period, 1) 18 to 19 Nov, the first two days of the haze event from 18 to 22 Nov and 2) rest of the campaign. The slope of the period 2) is 1.66, consistent with other studies (Poulain et al., 2021; Zhu et al., 2018) , whereas the slope in 1) gradually increases from 0.07 to 0.59, corresponding to the BBOA concentration decreasing from the beginning to the middle of this haze event.

[Figure]

[Figure]

**Comment #17 --- Minor remarks**

*When discussing the diurnal variations better referencing to the figure is necessary.*

**Response:**

We now directly reference Fig. 3c wherever diurnal variations are in section 3.2

**Comment #18 --- Minor remarks**

*Although the information of O/C, H/C are included in and Fig. 5 and 6, it would be useful for comparison to other studies, to have these average values as well as the N/C ratios illustrated on the factor profiles in Fig.3 and 4.*

**Response:**

To avoid cluttering Fig. 3, we have added these values to the supplement in Table S1, the elemental ratio (H:C, O:C and N:C) for eight factors from AMS and seven factors from the EESI-TOF. The elemental ratio from AMS is calculated according to Canagaratna et al. (2015). For the EESI-TOF, the molecule-dependent sensitivity is not considered in the calculation.

| | AMS | | | EESI | | |
|---|---|---|---|---|---|---|
| | H:C | O:C | N:C | H:C | O:C | N:C |
| HOA | $1.853 \pm 0.009$ | $0.022 \pm 0.0003$ | $0.006 \pm 0.0002$ | | | |
| COA | $1.629 \pm 0.005$ | $0.099 \pm 0.001$ | $0.003 \pm 0.0001$ | $1.755 \pm 0.032$ | $0.254 \pm 0.058$ | $0.008 \pm 0.008$ |
| BBOA | $1.419 \pm 0.021$ | $0.394 \pm 0.023$ | $0.022 \pm 0.002$ | $1.523 \pm 0.061$ | $0.426 \pm 0.060$ | $0.028 \pm 0.007$ |
| CCOA | $1.548 \pm 0.017$ | $0.155 \pm 0.015$ | $0.016 \pm 0.001$ | $1.569 \pm 0.030$ | $0.373 \pm 0.022$ | $0.017 \pm 0.003$ |
| MO-OOA$_{aq}$ | $1.323 \pm 0.018$ | $0.576 \pm 0.028$ | $0.038 \pm 0.005$ | $1.659 \pm 0.015$ | $0.390 \pm 0.009$ | $0.030 \pm 0.002$ |
| MO-OOA$_{SFC}$ | $1.220 \pm 0.016$ | $0.417 \pm 0.013$ | $0.011 \pm 0.002$ | $1.623 \pm 0.026$ | $0.354 \pm 0.008$ | $0.041 \pm 0.005$ |
| LO-OOA$_{SFC}$ | $1.656 \pm 0.031$ | $0.246 \pm 0.056$ | $0.064 \pm 0.013$ | $1.662 \pm 0.090$ | $0.409 \pm 0.107$ | $0.023 \pm 0.012$ |
| LO-OOA$_{ns}$ | $1.565 \pm 0.011$ | $0.134 \pm 0.008$ | $0.008 \pm 0.0005$ | $1.693 \pm 0.022$ | $0.334 \pm 0.008$ | $0.018 \pm 0.003$ |

**Comment #19 --- Minor remarks**

*Page 13, Line 11: Were external time series available for comparison, other than CO? Can you provide the value for the "good" correlation.*

**Response:**

The time series comparison and scatter plot of HOA vs. CO have been added to the supplement as Fig. S16. The slope in the scatter plot is 1332, consistent with the previous studies (Poulain et al., 2021; Zhang et al., 2017).

[Figure]

**Comment #20 --- Minor remarks**

*Page 22, Line 44 (please include correct section no.)*

**Response:**

This information was added to section 3.3.4 as following (page 14 line 41):

"Note that due to the application of the volatility-based filter for distinguishing particle-phase vs. spurious ions (see section Text S3), the contribution of such small, highly oxygenated ions presented here represents a lower limit."

**Comment #21 --- Minor remarks**

*Page 23, Line 7 r2 =of 0.93), remove = or of*

**Response:**

This typo was corrected (page 14 line 50):

"The LWC concentration is presented in Fig. 10, together with the time series of MO-OOA$_{aq}$. The two time series are strongly correlated ($r^2 = 0.93$), and both are dramatically higher during the 4 to 7 November event than for the rest of the study."

**Comment #22 --- Minor remarks**

*Page 21, Line 25: Bertrand et al.. please include the full reference.*

**Response:**

Because it is no longer clear when and in which form this data will be published, and a detailed discussion of this complex environmental campaign is beyond the scope of the current manuscript, the reference has been changed in the reference list:

Bertrand, A., personal communication.

**Comment #23 --- Minor remarks**

*Page 5, Line10 What is this diagnostic species?*

**Response:**

This question was also raised by Reviewer 1 (R1C16) and we repeat the response here. The sentence in question was meant to introduce the need to identify a diagnostic species, the identity (AMS NO$_3^+$) and use of which comprise the rest of the paragraph. To clarify this, the revised sentence reads (page 3 line 11 in supplement, as suggested by R1C19, we move this part to Text S4 in supplement):

"Therefore, we select a diagnostic species that can be measured with higher time resolution is utilised to monitor the sensitivity throughout the campaign."

**Comment #24 --- Minor remarks**

*Is the custom peak fitting algorithm something that could be applied to lower resolution instruments in the future?*

**Response:**

This is an interesting question, and one that we hope to explore in the future. In our view, a prerequisite for its routine use in spectral analysis would be a robust validation on synthetic data, which we have not yet conducted. Therefore we do not recommend it for widespread use (at the present time) and consider its use here as an *ad hoc* adaptation to sub-optimal instrument performance, with the lack of systematic validation increasing the uncertainties and informing our decision to utilise the EESI-TOF data for interpretative (qualitative) analysis rather than quantitative factor resolution (see also response to R1C1).

**Reference**

Brown, W. L., Day, D. A., Stark, H., Pagonis, D., Krechmer, J. E., Liu, X., Price, D. J., Katz, E. F., DeCarlo, P. F., Masoud, C. G., Wang, D. S., Hildebrandt Ruiz, L., Arata, C., Lunderberg, D. M., Goldstein, A. H., Farmer, D. K., Vance, M. E., and Jimenez, J. L.: Real-time organic aerosol chemical speciation in the indoor environment using extractive electrospray ionization mass spectrometry, Indoor Air, 31, 141-155, https://doi.org/10.1111/ina.12721, 2021.

Canagaratna, M. R., Jimenez, J. L., Kroll, J. H., Chen, Q., Kessler, S. H., Massoli, P., Hildebrandt Ruiz, L., Fortner, E., Williams, L. R., Wilson, K. R., Surratt, J. D., Donahue, N. M., Jayne, J. T., and Worsnop, D. R.: Elemental ratio measurements of organic compounds using aerosol mass spectrometry: characterization, improved calibration, and implications, Atmos Chem Phys, 15, 253-272, https://doi.org/10.5194/acp-15-253-2015, 2015.

Drewnick, F., Diesch, J. M., Faber, P., and Borrmann, S.: Aerosol mass spectrometry: particle–vaporizer interactions and their consequences for the measurements, Atmos. Meas. Tech., 8, 3811-3830, https://doi.org/10.5194/amt-8-3811-2015, 2015.

Elser, M., Huang, R. J., Wolf, R., Slowik, J. G., Wang, Q. Y., Canonaco, F., Li, G. H., Bozzetti, C., Daellenbach, K. R., Huang, Y., Zhang, R. J., Li, Z. Q., Cao, J. J., Baltensperger, U., El-Haddad, I., and Prevot, A. S. H.: New insights into PM2.5 chemical composition and sources in two major cities in China during extreme haze events using aerosol mass spectrometry, Atmos Chem Phys, 16, 3207-3225, https://doi.org/10.5194/acp-16-3207-2016, 2016.

Freney, E., Zhang, Y. J., Croteau, P., Amodeo, T., Williams, L., Truong, F., Petit, J. E., Sciare, J., Sarda-Esteve, R., Bonnaire, N., Arumae, T., Aurela, M., Bougiatioti, A., Mihalopoulos, N., Coz, E., Artinano, B., Crenn, V., Elste, T., Heikkinen, L., Poulain, L., Wiedensohler, A., Herrmann, H., Priestman, M., Alastuey, A., Stavroulas, I., Tobler, A., Vasilescu, J., Zanca, N., Canagaratna, M., Carbone, C., Flentje, H., Green, D., Maasikmets, M., Marmureanu, L., Minguillon, M. C., Prevot, A. S. H., Gros, V., Jayne, J., and Favez, O.: The second ACTRIS inter-comparison (2016) for Aerosol Chemical Speciation Monitors (ACSM): Calibration protocols and instrument performance evaluations, Aerosol Sci Tech, 53, 830-842, https://doi.org/10.1080/02786826.2019.1608901, 2019.

Hu, W. W., Hu, M., Hu, W., Jimenez, J. L., Yuan, B., Chen, W. T., Wang, M., Wu, Y. S., Chen, C., Wang, Z. B., Peng, J. F., Zeng, L. M., and Shao, M.: Chemical composition, sources, and aging process of submicron aerosols in Beijing: Contrast between summer and winter, J Geophys Res-Atmos, 121, 1955-1977, https://doi.org/10.1002/2015JD024020, 2016.
Huang, X. F., He, L. Y., Hu, M., Canagaratna, M. R., Sun, Y., Zhang, Q., Zhu, T., Xue, L., Zeng, L. W., Liu, X. G., Zhang, Y. H., Jayne, J. T., Ng, N. L., and Worsnop, D. R.: Highly time-resolved chemical characterization of atmospheric submicron particles during 2008 Beijing Olympic Games using an Aerodyne High-Resolution Aerosol Mass Spectrometer, Atmos Chem Phys, 10, 8933-8945, https://doi.org/10.5194/acp-10-8933-2010, 2010.

Lee, C.P., Surdu, M., Bell, D., Lamkaddam, H., Wang, M., Ataei, F., Hofbauer, V., Lopez, B., Donahue, N., Dommen, J., Prevot, A. S. H., Slowik, J. G., Wang, D., Baltensperger, U., and El-Haddad, I.: Effects of aerosol size and coating thickness on the molecular detection using extractive electrospray ionization, Atmos. Meas.Tech., submitted, 2021.

Pieber, S. M., El Haddad, I., Slowik, J. G., Canagaratna, M. R., Jayne, J. T., Platt, S. M., Bozzetti, C., Daellenbach, K. R., Frohlich, R., Vlachou, A., Klein, F., Dommen, J., Miljevic, B., Jimenez, J. L., Worsnop, D. R., Baltensperger, U., and Prevot, A. S. H.: Inorganic Salt Interference on CO2+ in Aerodyne AMS and ACSM Organic Aerosol Composition Studies, Environ Sci Technol, 50, 10494-10503, https://doi.org/10.1021/acs.est.6b01035, 2016.

Poulain, L., Fahlbusch, B., Spindler, G., Müller, K., van Pinxteren, D., Wu, Z., Iinuma, Y., Birmili, W., Wiedensohler, A., and Herrmann, H.: Source apportionment and impact of long-range transport on

carbonaceous aerosol particles in central Germany during HCCT-2010, Atmos. Chem. Phys., 21, 3667-3684, https://doi.org/10.5194/acp-21-3667-2021, 2021.

Sun, Y. L., Zhang, Q., Schwab, J. J., Yang, T., Ng, N. L., and Demerjian, K. L.: Factor analysis of combined organic and inorganic aerosol mass spectra from high resolution aerosol mass spectrometer measurements, Atmos. Chem. Phys., 12, 8537-8551, https://10.5194/acp-12-8537-2012, 2012.

Williams, L. R., Gonzalez, L. A., Peck, J., Trimborn, D., McInnis, J., Farrar, M. R., Moore, K. D., Jayne, J. T., Robinson, W. A., Lewis, D. K., Onasch, T. B., Canagaratna, M. R., Trimborn, A., Timko, M. T., Magoon, G., Deng, R., Tang, D., Blanco, E. D. L. R., Prevot, A. S. H., Smith, K. A., and Worsnop, D. R.: Characterization of an aerodynamic lens for transmitting particles greater than 1 micrometer in diameter into the Aerodyne aerosol mass spectrometer, Atmos Meas Tech, 6, 3271-3280, https://doi.org/10.5194/amt-6-3271-2013, 2013.

Wolf, R., El-Haddad, I., Slowik, J. G., Dällenbach, K., Bruns, E., Vasilescu, J., Baltensperger, U., and Prévôt, A. S. H.: Contribution of bacteria-like particles to PM2.5 aerosol in urban and rural environments, Atmospheric Environment, 160, 97-106, https://doi.org/10.1016/j.atmosenv.2017.04.001, 2017.

Xu, W., Croteau, P., Williams, L., Canagaratna, M., Onasch, T., Cross, E., Zhang, X., Robinson, W., Worsnop, D., and Jayne, J.: Laboratory characterization of an aerosol chemical speciation monitor with PM2.5 measurement capability, Aerosol Sci Tech, 51, 69-83, https://doi.org/10.1080/02786826.2016.1241859, 2017.

Zhang, X., Zhang, Y., Sun, J., Yu, Y., Canonaco, F., Prévôt, A. S. H., and Li, G.: Chemical characterization of submicron aerosol particles during wintertime in a northwest city of China using an Aerodyne aerosol mass spectrometry, Environmental Pollution, 222, 567-582, https://doi.org/10.1016/j.envpol.2016.11.012, 2017.

Zhu, Q., Huang, X. F., Cao, L. M., Wei, L. T., Zhang, B., He, L. Y., Elser, M., Canonaco, F., Slowik, J. G., Bozzetti, C., El-Haddad, I., and Prévôt, A. S. H.: Improved source apportionment of organic aerosols in complex urban air pollution using the multilinear engine (ME-2), Atmos. Meas. Tech., 11, 1049-1060, https://doi.org/10.5194/amt-11-1049-2018, 2018.